# Aqueous system-level processes and prokaryote assemblages in the ferruginous and sulfate-rich bottom waters of a post-mining lake

Daniel A. Petrash[1,2], Ingrid M. Steenbergen[1,3], Astolfo Valero[1,3], Travis B. Meador [1,3], Tomáš Pačes[2], Christophe Thomazo[4,5]

[1]Biology Centre of the Czech Academy of Sciences, České Budějovice, 370 05, Czechia
[2]Czech Geological Survey, Prague, 152 00, Czechia
[3]University of South Bohemia, České Budějovice, 370 05, Czechia
[4]University of Burgundy, Dijon, 21000, France
[5]Institut Universitaire de France, Paris, 75000, France

*Correspondence to*: Daniel A. Petrash (daniel.petras@geology.cz)

**Abstract.** In the low nutrient, redox stratified Lake Medard (Czechia), reductive Fe(III) dissolution outpaces sulfide generation from microbial sulfate reduction (MSR) and ferruginous conditions occur without quantitative sulfate depletion. The lake currently has marked overlapping C, N, S, Mn, and Fe cycles occurring in the anoxic portion of the water column. This feature is unusual in stable, natural redox stratified lacustrine systems where at least one of these biogeochemical cycles is functionally diminished or undergoes minimal transformations because of the dominance of (an)other component(s). Therefore, this post-mining lake has scientific value for (i) testing emerging hypotheses on how such interlinked biogeochemical cycles operate during transitional redox states; and (ii) to acquire insight on redox proxy signals of ferruginous sediments underlying a sulfatic and ferruginous water column. An isotopically constrained estimate of the rates of sulfate reduction (SRR) suggests that despite a high genetic potential, this respiration pathway may be limited by the rather low amounts of metabolizable organic carbon. This points to substrate competition exerted by iron and nitrogen respiring prokaryotes. Yet, the planktonic microbial succession across the nitrogenous and ferruginous zones also indicates genetic potential for chemolithotrophic sulfur oxidation. Therefore, our SRR estimates could be rather portraying high rates of anoxic sulfide oxidation to sulfate, probably accompanied by microbially induced disproportionation of S intermediates. Near and at the anoxic sediment−water interface, vigorous sulfur cycling can be fuelled by ferric and manganic particulate matter and redeposited siderite stocks. Sulfur oxidation and disproportionation then appear to prevent substantial stabilization of iron monosulfides as pyrite but enable the interstitial precipitation of microcrystalline equant gypsum. This latter mineral isotopically fingerprints sulfur oxidation proceeding at near equilibrium with the ambient anoxic waters, whilst authigenic pyrite-sulfur displays a 38 to 27 ‰ isotopic offset from ambient sulfate, suggestive of incomplete MSR and open sulfur cycling. Pyrite-sulfur fractionation decreases with increased reducible reactive iron in the sediment. In the absence of ferruginous coastal zones today affected by post-depositional sulfate fluxes, the current water column redox stratification in the post-mining Lake Medard is thought relevant for refining interpretations pertaining the onset of widespread redox stratified states across ancient nearshore depositional systems.

# 1 Introduction

The biogeochemical reactions governing the distinctive redox structure of modern permanently stratified lakes have been studied, for the most part, in natural settings featuring relatively high dissolved iron but low sulfate concentrations (Swanner et al., 2020). Improved by insights from laboratory experiments (e.g., Konhauser et al., 2007; Rasmussen et al., 2015; Jiang and Tosca, 2019), geochemical and microbiological analyses made in such lacustrine systems have provided us with an empirical framework to interpret modern iron biomineralization mechanisms and, by analogy, similar processes that would have allocated widespread, punctual deposition of ancient iron formations in the Precambrian.

Lakes that display permanent stagnation and marked redox gradients in their water column are termed meromictic. Meromictic lakes featuring ferruginous conditions in their water columns (i.e., $[Fe^{2+}] > [H_2S/HS^-]$ and $[Fe^{2+}] > [NO_3^-/NO_2^-]$) are relevant to decipher the environmental significance of specific chemical and isotopic signals recorded in iron-rich deposits, and to advance paleoenvironmental interpretations of redox stratified oceans, such as those prevalent during the Precambrian (Canfield et al., 2018), or intermittently developed during the Phanerozoic (Crowe et al., 2008; Walter et al., 2014; Posth et al., 2014; Lambrecht et al., 2018; Canfield et al., 2018; Swanner et al., 2020 Reershemiusand Planavsky, 2021).

Ferruginous water columns that also contain elevated dissolved sulfate concentrations are not uncommon in acidic shallow pit lakes (e.g., Denimal et al., 2005; Trettin et al., 2007), and have also been reported in pH neutralized post-mining lakes (McCullough and Schultze, 2018). Lake Medard, in NW Czechia (Fig. 1), belongs to this latter group. The newly formed lake features low nutrient contents (i.e., it is oligotrophic), and its temperature, redox and salinity stratified water column (Fig. 2a) remains unmixed throughout the year. Given its recent water filling history—completed in 2016, and the fact that its ferruginous bottom waters contain up to 21 mM of dissolved sulfate (Petrash et al., 2018), this oligotrophic lake can be considered as a large-scale incubation experiment featuring an imbalanced sulfatic transition between aqueous ferruginous and euxinic redox states. The later redox state is defined by an abundance of dissolved sulfide able to titrate dissolved $Fe^{2+}$ out from solution (Scholz, 2018; van de Velde et al., 2021).

Here we combined spectroscopic analyses of the hypoxic (i.e., 2.0 to 0.2 mg·L$^{-1}$ $O_2$), nitrogenous and ferruginous, and ultimately anoxic (< 0.03 mg·L$^{-1}$ $O_2$) ferruginous and sulfatic bottom water column of Lake Medard. System-level processes that can be linked to specific planktonic prokaryote functionalities were interpreted. For this aim, isotope ratios of carbon and oxygen in dissolved inorganic carbon, sulfur and oxygen in dissolved sulfate, and concentration profiles of bioactive ions and volatile fatty acids (VFAs) were measured together with a 16S rRNA gene amplicon sequence profile. Amplicon gene sequencing informed our ecological and biogeochemical interpretations despite quantitative biases that are inherent in this type of data (Salcher, 2014; Piwosz et al., 2020). To complement our interpretations, we also conducted mineralogical analyses and a mineral-calibrated wet chemical speciation study of reactive Fe and Mn pools in the upper anoxic sediments. Using these data, we developed a mechanistic model that assesses the potential regulatory roles of prokaryotes over the geochemical gradients detected in the water column, and their influence over interlinked biogeochemical cycling involving reactive minerals. Consumption and replenishment of iron, sulfur (S), carbon (C), nitrogen (N) and manganese (Mn) across the

redoxcline and near the anoxic sediment−water interface (SWI) are presented as a set of geochemical reactions. These reactions differentiate distinctive niches where a phylogenetically and metabolically diverse planktonic microbial community induce vigorous elemental recycling.

Our observations in this unique lake are thought relevant since analogue aqueous-level system processes would have also operated in some ancient ferruginous coastal settings. Lake Medard could therefore offer valuable information to further understand early diagenetic signals resulting from analogue microbial ecosystem dynamics. When preserved in the rock record, such signals could be elusive, and reflective, for instance, of ferruginous nearshore facies affected by continental sulfate delivery during shallow burial. In this regard, our research furthers understanding of the cryptic S cycle under ferruginous conditions unaccompanied by quantitative dissolved sulfate exhaustion.

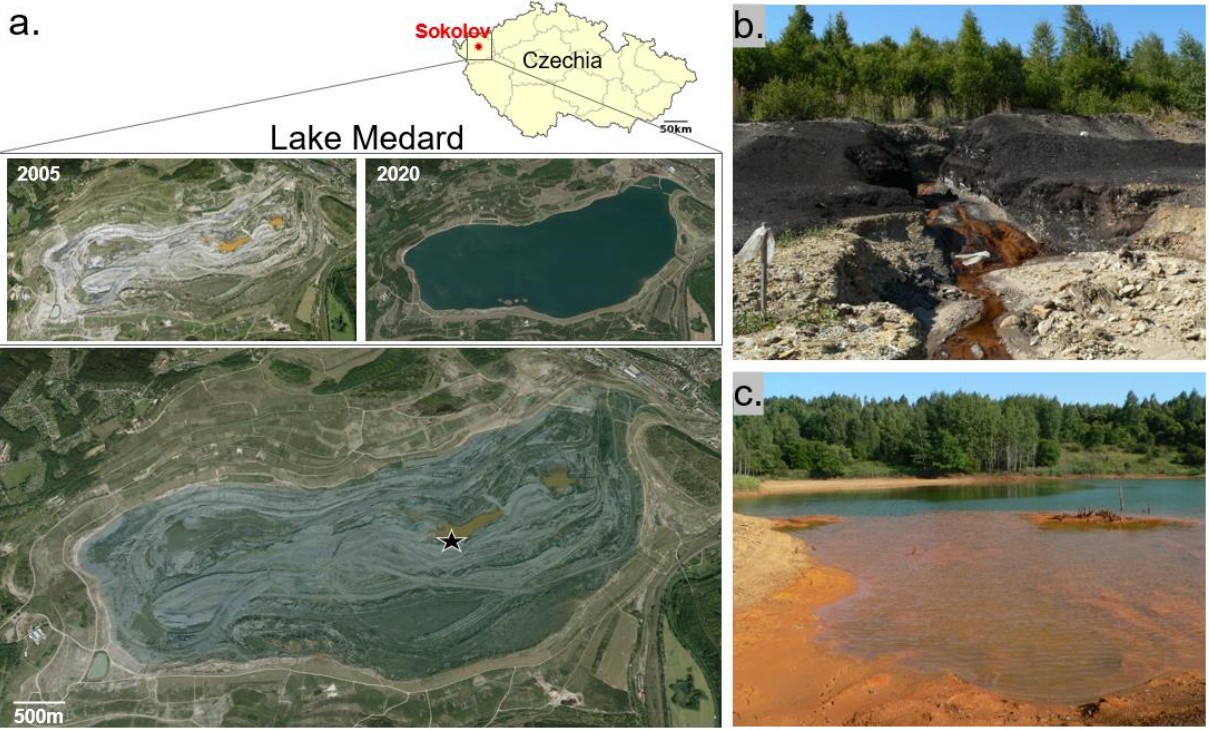

**Figure 1. The area now occupied by the post-mining Lake Medard was previously an open cast coal mine near Sokolov, NW Czechia. Upon mine abandonment, the deepest parts of the open cast mine became shallow acidic pit lakes and are now the lake depocenters. The deeper zone of the lake now features ferruginous and sulfate-rich aqueous conditions but the pH is circumneutral. The star marks the central sampling location in a recent lake imagery superimposed on the 2005 mine-pit imagery (a). The mine-pit had important fluxes of solutes linked to pyrite oxidation in exploited coal seams and their associated pyrite-bearing lithologies (b-c). These fluxes may still affect the hydrochemistry of the present-day lacustrine system, i.e., solutes are currently sourced from now submerged lithologies that also bear pH-neutralizing carbonates (Appendix A). Imagery dates 5/19/2020 (©CNES/Airbus) and 1/1/2004 (©GEODIS Brno). Historical photographic record by courtesy of The Czech Geological Survey.**

## 2 Study site

Reclamation (flooding) of land occupied by the decommissioned Medard open-cast lignite mine in the Sokolov mining district of Karlovy Vary, northwest Czechia, led to the ca. 4.9 km$^2$ (~60 m max. depth) post-mining Lake Medard (Fig. 1a; 50°10'41" N, 12°35'46" E). The lake was filled with waters diverted for reclamation purposes from the nearby Eger (Ohře) River. The filling of the former open-cast mine pit with river water started in 2010 and was reportedly completed by 2016 (Kovar et al., 2016). During closure and abandonment of the former mine pit, dissolved iron, and sulfate—derived from pyrite oxidation, leached towards initially shallow ephemeral and acidic pit-lakes formed as surficial and groundwater filled the mine pit (Fig. 1b-c). In these mining-impacted brines, metastable Fe(III)-oxyhydroxides and -oxyhydroxysulfates precipitated (Murad and Rojík, 2005). Runoff also affected the hydrochemistry of the ensuing shallow pit lake (Fig. 1b-c) by carrying solutes sourced from weathered, Miocene tuffaceous and carbonate-rich lacustrine claystones associated with the mined coal seam. These lithological units were described by Kříbek et al. (2017).

At present, Lake Medard exhibits density, temperature, and marked redox stratification in its hypolimnion that is hypoxic (0.2 to 0.03 mg·L$^{-1}$ O$_2$) to anoxic (Fig. 2a), and ferruginous (Petrash et al., 2018). Water-rock interactions down to the underlying granitic basement also influences the hydrochemistry of the modern lake. Percolation and subsurface flow of meteoric water causes dissolution of fault-related thernadite (Na$_2$SO$_4$) accumulations. Thernadite dissolution and groundwater reflux introduce significant loads of isotopically heavy sulfate into the present-day hydrological system (Pačes and Šmejkal, 2004). Additional details on the geological framework of the area and its influence over the hydrochemistry of the post-mining lake are in Appendix A.

Water column stratification was already observed in 2009, when environmental monitoring of the shallow pit-lake formed after decommissioning of the dewatering-wells took place (e.g., Medová et al., 2015). In the current deep post-mining lake, both, abiotic and microbially mediated precipitation of poorly crystalline iron minerals—i.e., amorphous ferric hydroxide (Fe(OH)$_3$) and metastable nanocrystalline ferrihydrite (Fe$_2$O$_3$·(H$_2$O)$_n$)—occurs near the pelagic redoxcline (i.e., the redox transition between low dissolved oxygen and anoxic waters, Fig. 2a), from where these solid phases are exported to the SWI (Petrash et al., 2018). Mineral equilibrium reactions at the SWI proceed mostly within the nitrogenous to ferruginous redox potentials (Eh), and at a circumneutral to moderately alkaline pH. Stability diagrams showcasing the predicted stability of S and Fe species in the bottom waters of Lake Medard are shown in Fig. B1 (Appendix B). The stability diagrams show that the current physicochemical conditions of the bottom sulfatic waters favour colloidal Fe(III)-oxyhydroxides formation, but ferruginous monimolimnial waters also occur.

## 3 Methods

### 3.1 Water sampling and analyses

### 3.1.1 Physicochemical parameter measurements and water column sampling

A water quality monitoring and profiling probe (YSI 6600 V2-2) was used—prior to sampling—to measure conductivity, temperature, $O_2$ concentrations, pH, and Eh in the stratified portion of the water column of Lake Medard (from 47 to 55 m depth) in its central location (Fig. 1a, star). The probing resolution was 1 m above and below the $O_2$ minimum zone and 0.5 m at the redoxcline. Based on the profiles, water column samples (n = 8; 4 replicates) were collected (in November 2019) using a Ruttner sampler with a capacity of 1.7 L. Flushing/rinsing of the sampling device with distilled water (dH$_2$O) was performed between samples. A total of eight samples were taken at depths 47, 48, 48.5, 49, 50, 52, 54 and 55 m. Replicate samples were taken at depths 47, 48.5, 50 and 54 m below the lake water surface. On aliquots of our water samples, we performed (i) prokaryote DNA extraction followed by MiSeq Illumina 16S rRNA gene amplicon sequencing; (ii) mass determinations of cations (iron, manganese, potassium, sodium, magnesium and calcium); (iii) high pressure liquid chromatography for concentrations of chlorine, sulfate, nitrate, ammonium and phosphate anions, and VFA abundances; (iv) measurement of dissolved inorganic carbon and methane concentrations, and (v) isotope ratio analyses of $\delta^{13}C$ in total dissolved inorganic carbon and methane; and (vi) isotope ratio analyses of $\delta^{34}S$ and $\delta^{18}O$ values in dissolved sulfate. Details on these analyses follow.

### 3.1.2 Environmental microbial DNA sampling

For each DNA sampling depth, an aliquot of 1 L was transferred to polyethylene (PET) bottles using a hand pump connected to sterile a Sterifil® Aseptic System loaded with sterile cellulose nitrate Whatman® Microplus-21 ST filters (0.45 µm cutoff, 47 mm diameter). The filters were separated from the filtrating apparatus using a pair of sterilized tweezers (70% ethanol and Bunsen burner) and transferred into sterile 2 mL CryoTube vials (Thermo Scientific). These were stored in liquid N$_2$ for transport to the lab, where DNA extraction from the biomass collected on the filters took place. After each sample collection, the filtration apparatus was rinsed 3 times with dH$_2$O, and a new filter was carefully placed onto the apparatus. Samples for 16 S rRNA gene analyses were collected from the two redox compartments of the lake: i.e., the hypoxic hypolimnion and anoxic monimolimnion. The rinsing water (1 L) prior to second-last sampling (52 m) was used as a control.

### 3.1.3 Microbiome profile

DNA was extracted from the water filters described above using Quick DNA Soil Microbe Kit (Zymo Research) according to the manufacturer's instructions. A total of 11 water replicates (i.e., 47 to 54 m depth and replicates) were evaluated. The DNA extracted from these samples was ≥ 6 ng as per Qubit dsDNA BR fluorometric assays (Life Technologies), and below limits of quantification (<L.Q.) for the control (i.e., nucleic acids <0.2 ng). DNA integrity was assessed by agarose gel (2%) electrophoresis.

A two-step PCR protocol targeting the small subunit 16S rRNA gene in bacteria and archaea was conducted using the universal primer combinations 341F/806R (CCTAYGGGRBGCASCAG and GGACTACNNGGGTATCTAAT) and 519F/915R (CAGCCGCCGCGGTAA and GTGCTCCCCCGCCAATTCCT), respectively. The samples were sequenced on the MiSeq Illumina platform. The 16Ss rRNA gene amplicon datasets were analyzed with a pipeline consisting of an initial step where

all reads passing the standard Illumina chastity filter (PF reads) were demultiplexed according to their index sequences. This was followed by a primer clipping step, in which the target forward and reverse primer sequences for bacteria and archaea were identified and clipped from the starts of the raw forward and reverse reads. Only read pairs exhibiting forward and reverse primer overlaps were kept for merging by using FLASH 2.2.00 (Magoč and Salzberg, 2011). This yielded a total of 1,799,339

high-quality sequence reads, with an average length—after processing—of 412 bp.

Sequence features (herein described as representative operational taxonomic units, OTU) were clustered using QIIME2 (VSEARCH cluster-features-de-novo option; Rognes et al., 2016). To assign taxonomic information to each OTU, we performed DC−MEGABLAST alignments of cluster-representative sequences regarding the NCBI sequence database (Release 2019−10−10). A taxonomic assignment for each OTU was then transferred from the set of best-matching reference sequences

(lowest common taxonomic unit of all best hits). Hereby, a sequence identity of >70% across at least ≥ 80% of the representative sequence was a minimal requirement when considering reference sequences. We assigned significant tentative correspondence of OTUs to reference species provided that an identity threshold ≥ 97 % of the V3−V4 hypervariable region for bacteria and V4-V5 for archaea were meet. Further processing of OTU and taxonomic assignments (75.8% of the sequences after chimera detection and filtering; Edgar et al., 2011) and read abundance estimation for all detected OTU was performed

using the QIIME2 software package (version 1.9.1, http://qiime.org/). Abundances of bacterial and archaeal taxonomic units were normalized using lineage-specific copy numbers of the relevant marker genes to improve estimates (Angly, 2014). The microbial sequence data for this study (lengths ≥ 402 bp) were deposited in the European Nucleotide Archive (ENA) at EMBL−EBI under accession number PRJEB47217.

### 3.1.4 Cation concentration analyses

For cation concentration analyses, aliquots of 15 mL were filtered using sterile high flow, 28 mm diameter, polyethersulfone (PES) filters to remove particles >0.22 µm and then placed in acid-cleaned, PET centrifuge tubes. The aliquots were acidified using concentrated trace metal grade $HNO_3$. At the lab these water aliquots were digested with trace metal grade $HNO_3$ (8 N) and were sent for analyses at the Pôle Spectrométrie Océan at IUEM in Brest, France. A Thermo Element2 high resolution inductively coupled plasma mass spectrometer set on solution mode was used. The data were calibrated against multi-element

standards at concentrations that were measured repeatedly throughout the session. Multi-element solutions were measured at the beginning, end, and twice in the middle of the sequence and a 5 µg·L$^{-1}$ standard was further repeated after every five samples throughout the sequence. Additionally, 5 ppb indium (In) was added directly to the 2% $HNO_3$ diluant employed to prepare all standard solutions and was used to monitor signal stability and correct for instrumental drift across the session. Each sample and standard were bracketed by a rinse composed of the same diluant (i.e., the 2% $HNO_3$ with In) for which data

was also acquired to determine the method detection limit. Relative standard deviations (2σ level) were better than 0.01 wt. % for Fe and Mn, and between 0.001 and 0.002 % for other analyzed elements, e.g., K, Na, Mg, Ca, concentrations of which were used for aqueous-mineral equilibrium modeling (Appendix A, also Supplement 1: Phreeqc modeling input/results).

### 3.1.5 Ions, ammonia, and VFAs concentration analyses

Alkalinity (i.e., the capacity of water to neutralize free hydrogen ions, $H^+$) was measured as $HCO_3^-$ via acidometric titration of filtered water samples. The titrations were conducted on board immediately upon sample collection by using 0.16 N sulfuric acid cartridges on a digital titrator (Hach).

Ions, ammonia and VFAs concentrations were measured in filtered, unacidified water sample aliquots via high pressure liquid chromatography (HP-LC) at BC-CAS, České Budějovice. For these analyses we used an ICS5000 + Eluent Generator (Dionex), with conductivity detection application, and suppression. Analytes were separated using Dionex IonPac AS11-HC-4 µm (anions, VFAs) and IonPac CS16-4 µm (ammonium) columns (2x250 mm in size). The flow rate was 0.36 mL/min; run time was 65 min (anions, VFAs) and 17 min for ammonium. Potassium hydroxide was the eluent for inorganic anions and monovalent organic acids; methanesulfonic acid was the eluent for ammonium ion detection/quantification. A combined stock calibration standard solution featuring environmentally relevant anions ratios was used for determining concentrations and was prepared from corresponding analytical-reagent grade salts. To optimize and calibrate the method for VFA analyses and determine the limits of detection, we used stock mixtures of IC grade formate, oxalate, acetate, lactate, pyruvate, and butyrate standards for preparing our working saline stock solutions. Detection limits were better than 60 ppb for lactate and oxalate, and 200 ppb for pyruvate, formate, and acetate. Recoveries, based on standards, exceed 80 % for all analytes reported. The ion concentration measurements have an error $(2\sigma) < 20$ % based on replicate analyses.

### 3.1.6 Dissolved (in)organic carbon and methane

Aliquots of the lake water collected were immediately transferred from the sampler to pre-cleaned—i.e., three-times rinsed with ddH$_2$O and oven-dried at 550 ºC, 12 mL glass exetainer septum capped vials (Labco), pre-filled with He(g) and 1mL NaCl oversaturated solution (40%) for CH$_4$, or 1 mL 85% phosphoric acid for $\Sigma CO_2$. On board, the vials were filled with ~11 mL water samples using a syringe connected to 15 cm PES tube that was introduced from below into the sampler to prevent diffusion of atmospheric gases into the exetainer vials.

A dissolved inorganic carbon ($\Sigma CO_2$) concentration profile was produced using a peak area calibration curve obtained on a MAT253 Plus isotope ratio mass spectrometer (IR-MS; Thermo Scientific). The same instrument was used for determining isotope ratios of $\Sigma CO_2$ ($\delta^{13}C_{\Sigma CO2}$, $\delta^{18}O_{\Sigma CO2}$) and methane ($\delta^{13}C_{CH4}$), and for a rough estimation of the CH$_4$ concentrations at the monimolimnion. In brief, CO$_2$ (or CH$_4$) is purged from the headspace of the exetainer vials, then the gas passes through a Nafion water trap and into a sample loop PoraPlot-Q column (0.32 mm ID) cooled in liquid N$_2$; with He as the carrier gas. The sample gases are then separated via a Carboxen PLOT 1010 (0.53 mm ID; Supelco) held at 90°C with a flow rate of 2.2 mL·min$^{-1}$ and transferred via a Conflo IV interface to the instrument. For methane, prior to transfer to the IR-MS, the sample is transferred via a multi-channel device to a nickel oxide conversion reactor tube with copper oxide as catalyst (1,000°C). The $\delta^{13}C$ values obtained relative to CO$_2$ working gas are then corrected for linearity and normalized to laboratory working standards calibrated against CO$_2$ evolved from the international standard IAEA-603.

The concentration measurements have an error (1σ) < 4 % for $\Sigma CO_2$ and < 25 % for $CH_4$. Isotope data are expressed in delta notation, $\delta = R_{sample}/R_{standard} - 1$, where R is the mole ratio of $^{13}C/^{12}C$ or $^{18}O/^{16}O$ and reported in units per mil (‰). The $\delta^{13}C$ data are reported vs. the Vienna Pee Dee Belemnite (V-PDB) standard. The $\delta^{18}O$ data are reported vs. the international Vienna Standard Mean Ocean Water (V-SMOW) standard. The reproducibility of the $\delta^{13}C_{DIC}$ and $\delta^{13}C_{CH4}$ measurements was better than ±0.05 ‰ and ±0.3 ‰ (1σ), respectively, based on replicates for reported values of the standard materials and the samples.

Reproducibility of $\delta^{18}O_{\Sigma CO2}$ measurements is better than 0.4 ‰. DOC was analyzed in untreated samples by catalytic combustion at 680 °C (Shimadzu 5000A) with a detection limit of ~0.05 mg·L$^{-1}$.

### 3.1.7 Dissolved sulfur analyses

    For measuring dissolved acid-volatile sulfur (AVS) in the monimolimnion (i.e., HS$^-$, intermediate sulfur species, H$_2$S and the aqueous FeS clusters; Rickard & Morse, 2005), 500 mL aliquots of water samples collected at the 52-54 m depth interval were

220 transferred to PET sample bottles pre-filled with 2 mL of 1 M Zn acetate, then 50 mL of 5 M NaOH were added. The combined concentrations of AVS bound into the ZnS precipitates were spectrophotometrically determined in an acidified solution of phenylenediamine and ferric chloride by using a Specord 210UV/Vis (Analitik). The detection limit of the method is ≥ 0.25 μM.

    As for cation analyses, 1L aliquots of the filtered water samples were intended for sulfate S and O isotope analyses. These

225 samples were acidified to a pH ~3 with 6N reagent grade HCl. Also, to oxidize and degas dissolved organic matter, we added 6 ml of hydrogen peroxide (H$_2$O$_2$) 6 % and heated the samples (90 °C) until clear (i.e., 1 to 3 h). Dissolved sulfate was then precipitated as purified barite (BaSO$_4$) by using a saturated BaCl$_2$ solution. Accordingly, after heating, ~5 ml of 10 % BaCl$_2$ was added to the water samples that were then allowed to cool down overnight. An additional 1mL of BaCl$_2$ solution was added the next day to ensure that all possible BaSO$_4$ precipitated. The precipitates were then collected on pre-weighed

membrane filters, rinsed thoroughly using deionized water, stored in plastic petri dishes, and dried in a desiccator using a sulfate-free desiccant, the dry BaSO$_4$ powder was scraped into clean vials, weighted, and stored until shipped to the Biogéosciences Laboratory, Dijon, France, for isotope analysis.

    Each purified BaSO$_4$ sample was analyzed for $\delta^{34}S_{SO4}$ and $\delta^{18}O_{SO4}$. Samples were measured on a Vario PYRO cube elemental analyzer (Elementar) in-line with a 100 IR-MS (IsoPrime) in continuous flow mode. The SO$_4^{2-}$ isotope data are expressed in

in the $\delta$-notation, $\delta \equiv R_{sample}/R_{standard} - 1$, where R is the mole ratio reported in units per mil (‰) vs. the Vienna Canyon Diablo Troilite (V-CDT) and V-SMOW standards for $^{34}S/^{32}S$ and $^{18}O/^{16}O$, respectively. Analytical errors are better than ± 0.4 ‰ (2σ) based on replicate analyses of the international barite standard NBS-127, which was used for data correction via standard-sample-standard bracketing. International standards IAEA-S-1, IAEA-S-2 and IAEA-S-3 were used for calibration with a cumulative reproducibility better than 0.3 ‰ (1σ).

**3.2 Sediment samples**

We also sampled the upper anoxic sediment column to a depth of ~8 cm. The mineralogy of these fine-grained sediments (silt to clay in size) was qualitatively and semi-quantitatively assessed via X-ray diffraction (XRD). The $\delta^{34}$S and $\delta^{18}$O of gypsum ($CaSO_4 \cdot 2H_2O$), $\delta^{13}$C of siderite ($FeCO_3$), and $\delta^{34}$S isotope values of pyrite ($FeS_2$) from these sediments were also measured and reported as described above using the delta notation, $\delta = R_{sample}/R_{standard} - 1$, where R is the mole ratio. Scanning electron microscopy aided by electron dispersive spectrometry (SEM-EDS) was used for textural analyses focused on the S- and/or Fe-bearing phases. In addition, a sequential extraction scheme (after Poulton et al., 2004; Goldberg et al., 2012) was conducted to characterize the sedimentary partitioning of reactive Fe and Mn fractions. Details on these analyses follow.

### 3.2.1 Sampling

Replicate sediment cores (~16 cm in length) were collected with a messenger-activated gravity corer attached to 20 cm-long polycarbonate tubes (5 cm in diameter). The cores were immediately sealed upon retrieval with butyl rubber stoppers, preserving about 3 cm of anoxic lake water. The head water showed no signs of oxidation (i.e., no reddish hue observed) upon transport—within about 6 h from collection—to the lab. The sediment pile was extruded and sectioned at 2 cm intervals. Surfaces of the silty clayey sediment in contact with the core liner were scrapped to remove potential contamination from the lake water and to minimize smearing effects. The sediment subsamples were rapidly frozen using liquid $N_2$ and then stored at $-18\ ℃$ until freeze-dried. We interrogated the upper part of the sediment pile to a depth of 8 cm (i.e., 2 replicate samples per depth; 2 cores).

### 3.2.2 Mineralogy

The mineralogy of the sediment was determined, semi-quantitatively, via X-ray diffraction (XRD). Powder XRD data were collected on a D8 Advance powder diffractometer (Bruker) with a Lynx Eye XE detector, under a Bragg-Brentano geometry and Cu K$_1$ radiation ($\lambda=1.5405$ Å). Collection in the 2Θ range 4$-$80° was performed using 0.015° step-size increments and 0.8 s collection time per step size. Qualitative phase analyses were performed by comparison with diffraction patterns from the PDF-2 database. A semi-quantitative phase analysis was performed by the Rietveld refinement method (Post & Bish, 1989), as implemented in the computer code Topas 5 (Bruker). The crystal structure of the mineral phases used for refinement were obtained from the Inorganic Crystal Structure Database (ICSD) database. During Rietveld refinement, only the scale factors, unit-cell parameters, and size of coherent-diffracting domains were refined. A correction for preferred orientation was applied for selected mineral phases (i.e., K-feldspar, mica, gypsum).

The abundance of sedimentary Fe- and Mn-bearing phases was established by applying a sequential extraction scheme aiming to quantify the contribution of the operationally defined reactive pool capable of reacting after reductive dissolution with sulfide (after Poulton and Canfield, 2005). A wet chemical extraction scheme was applied to liberate (i) the fraction of total acid volatile sulfur (AVS) in the sediment, which might consist of mackinawite, a portion of greigite, and an (usually) unknown, yet typically negligible fraction of pyrite (Rickard and Morse, 2005); and (ii) chromium reducible sulfur (CRS), consisting primarily in pyrite but also in the sediment intermediate sulfur compounds (Canfield et al., 1986). AVS was

extracted with cold concentrated HCl for 2 h. Then, the resulting hydrogen sulfide concentration (i.e., between 0.004 and 0.036 wt. %) was precipitated as $Ag_2S$ by using a 0.3 M $AgNO_3$ solution. Subsequently, CRS was liberated using a hot and acidic

1.0 M $CrCl_2$ solution (Canfield et al., 1986). The resulting $H_2S$ was trapped as $Ag_2S$. Mass balance after gravimetric quantification was used to calculate the amount of AVS and CRS. Concentration analyses of Fe and Mn dissolved in each of these extracts were conducted via ICP-MS measurements (Xseries II, Thermo Scientific) at the Department of Environmental Geosciences, Czech University of Life Sciences, Prague.

### 3.2.3 Sedimentary geochemistry and stable S, O and C isotope analyses

Aliquots of the sediment samples were analyzed for total S ($S_{tot}$) concentration using a CS analyzer (ELTRA GmbH). The detection limit was 0.01 wt. % for $S_{tot}$. The relative errors using the reference material (CRM 7001) was ± 2 % for $S_{tot}$.

Total S for $\delta^{34}S$ determination was extracted in the form of $BaSO_4$ from the sediments. To evaluate the S and sulfate-O isotope ratios of gypsum ($\delta^{34}S_{gy}$), first the heavy mineral fraction of the samples, which includes pyrite, was excluded by using 1,1,2,2-tetrabromethane ($\rho= 2.95$). The gypsum was then dissolved in $ddH_2O$ to extract sulfate. The free sulfate obtained was

precipitated as $BaSO_4$ as described above (Sect. 3.1.7). The $BaSO_4$ was then converted to $SO_2$ by direct decomposition mixed with $V_2O_5$ and $SiO_2$ powder and combusted at 1000 °C under vacuum ($10^{-2}$-$10^{-3}$ mbar); mass spectroscopic measurements of the evolved $SO_2$ were conducted on a Finnigan MAT 251 IR-MS dedicated to S isotope determinations. The results are expressed in delta notation and reported against the V-CDT and V-SMOW standards. The accuracy of the measurements was determined via international standards, with reproducibility better than 0.2 ‰.

The IR-MS instrument used to evaluate the isotope ratios of dissolved sulfate was also used for determining the $\delta^{34}S$ of pyrite in the upper anoxic sediments. Prior to analyses, an AVS/CRS wet chemical extraction scheme similar to the one described above was applied. After centrifugation, the $Ag_2S$ precipitate was washed several times with $ddH_2O$ and oven-dried at 50 °C for 48 h. The pyrite $\delta^{34}S$ measurements were performed on $SO_2$ molecules via combustion of ~500 mg of silver sulfide homogeneously mixed with an equal amount of $WO_3$ using a Vario PYRO cube (Elementar GmbH) connected online via an

open split device to the IR-MS. International standards (IAEA-S-1, IAEA-S-2, IAEA-S-3) were used for calibration. Isotope results are reported in the delta notation against the V-CDT standard. Analytical reproducibility was better than 0.5 ‰ based on replicates for standard materials and samples.

The isotope ratios of carbonate in the sediment fraction were evaluated—after removal of organic carbon with $H_2O_2$, by implementing the method described by Rosenbaum and Sheppard (1986). These were measured using a Delta V mass

spectrometer (Thermo Fisher Scientific) coupled with an EA-1108 elemental analyzer (Fisons). The same instrument was used for measuring the sediment $\delta^{13}C_{org}$. For this purpose, the samples where finely milled, place in tin (Sn) capsules, and oxidized to $CO_2$ at 1,040°C in the elemental analyzer. The reproducibility of the isotope measurements for organic C was better than ±0.12 ‰, and better than ± 0.1 ‰ for both carbon and oxygen isotopes of siderite. For siderite, the accuracy of the measurement was monitored by analyses of the IAEA NBS-18 ($\delta^{13}C= -5.014$ ‰, $\delta^{18}O= -23.2$ ‰) and two in-house standards; the long-

term reproducibility is better than 0.05 ‰ for $\delta^{13}C$ and 0.1 ‰ for $\delta^{18}O$.

### 3.2.4 Textural features

For SEM of the sediments, we used a Mira 3GMU scanning electron microscope (TESCAN) combined with a NordlysNano electron back-scattering diffraction (EBSD) system for semi-quantitative chemical petrography, and a Magellan 400 (FEI) for higher resolution imaging in secondary electron mode.

**4 Results and discussion**

### 4.1 Bottom water column stratification and dissolved oxygen levels

Physicochemical parameters measured in the dysoxic to anoxic waters at the time of sampling are shown in Fig. 2a. Profiling of these parameters was consistent with several previous and subsequent probe monitoring measurements in the meromictic post-mining lake (e.g., Petrash et al., 2018). The pH in the hypolimnion was ~8.2 and decreased moderately downwards, reaching 7.4 ± 0.2 units near the anoxic SWI. Simultaneous reactions involving dissolution, anoxic re-oxidation and (re)precipitation of reactive minerals could be responsible of this moderate pH decrease (see Soetaert et al., 2007). These reactions are considered in subsequent sections of this work.

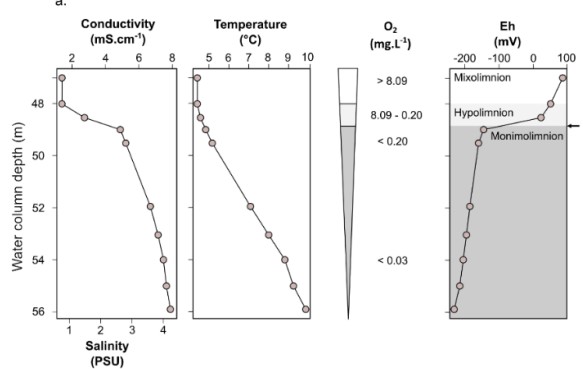

Conductivity exhibited a steep gradient at ca. 48 m depth that flattens with increasing depth. Temperature increased gradually towards the bottom. The zone in the water column where these gradients concur is referred to as the hypolimnion. Increased conductivities within the hypolimnion of post-mining lakes, such as examined here, could result from the legacy of the former mine drainage and/or from groundwater inflow (e.g., Denimal et al., 2005; Schultze et al., 2010).

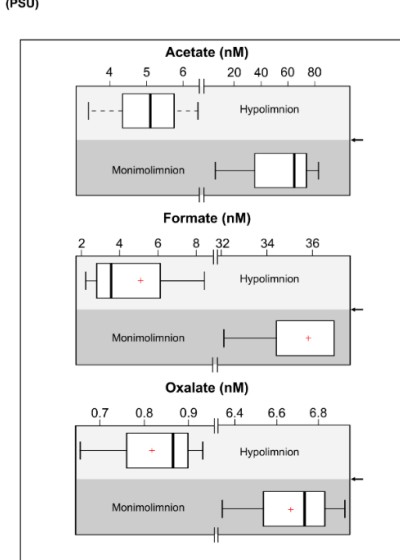

**Figure 2. Physicochemical parameters in the dysoxic to anoxic waters of Lake Medard in its central sampling location, which has a maximum depth of 56 m (a), and concentrations range of acetate, formate and oxalate quantified in the dysoxic (n= 4; <48 depth) and anoxic (n= 3; 54-55 m depth) waters of Lake Medard (b). The arrow shows the redoxcline's depth at the time of sampling.**

Salinity was directly derived by using the measured conductivity values (after Hambright et al., 1994). It increased three-fold from the hypolimnion downwards (Fig. 2a). This could result from recharge of groundwater carrying high loads of dissolved salts, and/or from the lack of mixing of the legacy mine-impacted pit lake waters with those now comprising the mixolimnion.

The temperature gradient, on the other hand, is a consequence of limited seasonal vertical heat exchange between the density stratified water column and the mixolimnion (Boehrer and Schultze, 2008).

Molecular oxygen ($O_2$) from the mixolimnion cannot be replenished below the density and thermally stratified bottom waters, and $O_2$ dropped rapidly within the 48 to 49 m depth-interval of the water column from about 8.1 to ~0.2 mg·$L^{-1}$. The deepest part of the lake is anoxic (Fig. 2a). At this level, the Eh shifts from >100 mV at the lower mixolimnion to negative values down to ≤ −230 mV near the SWI. The dysoxic, nitrogenous zone of the water column is referred to as the hypolimnion; it contains a sharp redox boundary zone referred to as the redoxcline. Below the redoxcline lies the monimolimnion which becomes anoxic (ferruginous) towards the SWI (Fig. 2a).

The hydrochemically different monimolimnion persists in the deepest depressions of the lakebed throughout the year; although with slight variations in the monitored Eh and pH ranges that could be accompanied by minor (±1 m) shifts in the vertical position of the redoxcline. In this study, we focused on the central part of the lake as it exhibited the broadest Eh range in its bottom water column (Fig. 2a). Details on the eastern and western sampling locations are available in a descriptive study by Petrash et al. (2018). Short-lived changes in redox potential of about 150 mV in the bottom water column were recently considered by Umbría-Salinas et al. (2021). These changes have effects on water column speciation (Fig. B1, Appendix B), and affect the partitioning of several redox sensitive metals that bind to reactive iron phases in the upper sediments (Umbría-Salinas et al. 2021, for details).

## 4.2 Dissolved carbon concentrations and $\delta^{13}C$ isotope values

### 4.2.1 Dissolved organic carbon (DOC)

The average measured DOC concentration in the sampled waters is 1,050 ± 500 µM. This range of values was higher than observed in the bottom waters of meromictic lakes such as Matano (< 100 µM; Crowe et al., 2008), or Pavin (300 ± 100 µM; Viollier et al., 1995). DOC is generally comprised of relatively high molecular weight organic compounds (not quantified here), such as cellular exudates from living and senescent planktonic microorganisms (e.g., algae, protists, bacteria) and their degradation products. Probably also present in solution are soluble humic substances (HSs) derived from the biological breakdown of refractory organic matter (e.g., lignite particles) in the sediment (Petrash et al., 2018). VFAs are linear short-chain aliphatic mono-carboxylate compounds produced during anaerobic degradation of the organic compounds referred above. They serve as C sources and electron donors for planktonic microbial heterotrophy and were therefore quantified here. VFAs in the bottom waters were at nanomolar concentrations that are reflective of the general scarcity of labile organic substrates. A six- to ten-fold increase in concentrations of acetate, oxalate, and formate occurred towards the increasingly saline and $O_2$-depleted waters. Concentrations of lactate, propionate, and butyrate could be detected at similar nanomolar magnitudes in the mixolimnion (not shown), but in the monimolimnion these VFAs were exhausted, i.e., below <L.Q.

The concentrations of total dissolved inorganic carbon (i.e., $\Sigma CO_2 = H_2CO_3 + HCO_3^- + CO_3^{2-}$) ranged from 1.9 to 9.8 mM and increased downwards (Fig 3a). This parameter positively correlated with alkalinity, which ranged from 1.8 to 2.9 meq·L$^{-1}$. Total dissolved inorganic carbon exhibited lower $\delta^{13}C$ values at the anoxic monimolimnion and $[\Sigma CO_2]$ were inversely correlated with the $\delta^{13}C$ values (Table 1, cf. Figs. 3a-b). The $\delta^{13}C$ values are in the range +0.2 to −4.1 and were inversely correlated with the dissolved sulfate concentrations $[SO_4^{2-}]$ too (Table 1), whilst $[SO_4^{2-}]$ and $[\Sigma CO_2]$ were directly correlated

(Fig. 3b-c). From these observations, an increased $\Sigma CO_2$ to alkalinity ratio is consistent with heterotrophy exceeding gross primary production (for example from chemo- and photo-autotrophy). But admixture of the lake's monimolimnion with groundwater carrying geogenic $CO_2$ could also alter the $\Sigma CO_2$ / alkalinity balance. A contribution of organically derived $CO_2$ is evident—as per $\delta^{13}C$ data, yet it could be argued that in the monimolimnion, sulfate reduction has only a moderate impact on alkalinity generation. Although speculative, it is possible that microbial sulfate reduction (MSR) is responsible for the

observed lactate depletion. Therefore, the complete (to $CO_2$) and incomplete (to acetate) oxidation of lactate by MSR could be a factor contributing to the slight decrease in pH in the monimolimnion (see Gallagher et al., 2012).

**Table 1. Measured concentrations and isotopic ratios in the $O_2$ depleted bottom water column of the central sampling location (from 47 to 55 m depth below the surface), Lake Medard.**

| Depth (m) | pH | Eh (mV) | $O_2$ [mg·L$^{-1}$] | Cond. (µS·cm$^{-1}$) | $\Sigma CO_2$# mM | $\delta^{13}C^{(a)}$ (‰)V-PDB | $\delta^{18}O^{(b)}$ (‰)V-PDB | $[CH_4]$ µM | $\delta^{13}C^{(c)}$ (‰)V-PDB | $[NO_3^-]$ | $[NH_4^+]$ | $[Fe^{2+}]$ [µM] | $PO_4^{-3}$ | $[Mn^{2+}]$ | $[SO_4^{2-}]$ mM | $\delta^{34}S^{(d)}$ (‰)V-CDT | $\delta^{18}O^{(e)}$ (‰)V-SMOW |
|---|---|---|---|---|---|---|---|---|---|---|---|---|---|---|---|---|---|
| **47** | 8.1 | 85.9 | 8.0 | 1394 | n.d. | n.d. | n.d. | n.d. | n.d. | 23.8 ± 0.5 | 3.4 ± 0.4 | <0.07 | <1.78 | 0.3 ± 0.01 | 6.0 ± 0.8 | 10.9 ± 0.1 | 2.4 ± 0.1 |
| **48** | 8.1 | 88.4 | 8.0 | 1409 | 1.9 ±0.1 | +0.2 ± 0.05 | 13.2 ± 0.2 | n.d. | n.d. | 24.5 ± 0.5 | 5.2 ± 0.5 | <0.07 | 9.1 ± 1.8 | 0.4 ± 0.01 | 5.9 ± 0.8 | 13.5 ± 0.07 | 2.6 ± 0.1 |
| **48.5** | 7.8 | -36.4 | 3.7 | 3143 | 3.5 ±0.2 | −0.1 | 13.1 | n.d. | n.d. | 26.0 ± 0.5 | 15.4 ± 0.5 | <0.07 | 1.0 ± 0.2 | 19.9 ± 0.4 | 8.3 ± 0.8 | 11.3 ± 0.03 | 2.4 ± 0.4 |
| **49** | 7.8 | −145.1 | 0.9 | 4871 | 7.5 ±0.1 | −2.1 ± 0.03 | 14.2 ± 0.1 | n.d. | n.d. | 19.8 ± 1.2 | 34.5 ± 9.3 | <0.07 | 3.8 ± 0.8 | 20.1 ± 0.4 | 9.6 ± 1.6 | 11.5 ± 0.1 | 3.9 ± 0.1 |
| **50** | 7.7 | −159.9 | 0.1 | 5197 | 5.9 ±0.1 | −2.7 ± 0.1 | 12.7 ± 0.4 | 3.0 ± 0.6 | −68.0 | 17.5 ± 1.2 | 68.7 ± 9.3 | 22.8 ± 0.4 | 10.1 ± 1.0 | 30.6 ± 0.5 | 12.8 ± 1.6 | 12.1 ± 0.1 | 3.5 ± 0.3 |
| **52** | 7.7 | −185.3 | 0.07 | 6661 | 9.8 ±0.2 | −2.5 ± 0.1 | 13.8 ± 0.2 | 7.0 ± 1.3 | −68.2 | 18.7 ± 0.2 | 87.3 ± 14.7 | 20.5 ± 2.3 | 11.7 ± 2.3 | 14.6 ± 0.3 | 14.4 ± 0.7 | 12.5 ± 0.1 | 3.7 ± 0.2 |
| **54** | 7.7 | −204.6 | 0.06 | 7440 | 9.0 ±0.2 | −3.9 ± 0.4 | 13.2 ± 1.1 | 1.9 ± 0.3 | −66.4 | 18.3 ± 0.2 | 134.5 ± 14.7 | 32.7 ± 6.5 | 19.9 ± 4.0 | 13.7 ± 0.2 | 16.8 ± 0.7 | 13.3 ± 0.1 | 4.0 ±0.4 |
| **55** | 7.7 | −214.8 | 0.03 | 7618 | n.d. | n.d. | n.d. | 6.8 ± 0.7 | −67.2 | 17.9 ± 0.2 | 127.7 ± 14.7 | 28.1 ± 5.6 | 28.1 ± 5.6 | 11.1 ± 0.2 | 16.0 ± 0.7 | n.d. | n.d. |

#$\Sigma CO_2 = H_2CO_3 + HCO_3^- + CO_3^{2-.}$
Precision of the isotopic values reported—based on repeated measurements of analytical standards
(better than; ‰ 2σ): (a) 0.05; (b) 0.4; (c) 0.9; (d) 0.1; (e) 0.4


The $CO_2$ source flux at the lake floor was estimated using a two-component mixing model that considers the $\delta^{13}C$ values in sedimentary carbonates and organic matter. An input to our model is the isotope values of the sedimentary organic matter ($\delta^{13}C$

= $-27.9 \pm 0.1$ ‰, n=6), and those of (bi)carbonate ions derived from the dissolution of carbonate phases near the SWI and below (Table 1). For the latter, a minor contribution of $\Sigma CO_2$ evolved from the oxidation of methane (mean $\delta^{13}C_{CH4} \approx -67$ ‰; Table 1) might also be possible and was considered. This methane diffuses throughout the anoxic sediments to the bottom water column. To account for the reactive C of the sedimentary carbonates, we used the $\delta^{13}C$ mean values in the anoxic sediments ($+6.4 \pm 0.3$ ‰), which is within the range reported for carbonates in the lignite-associated lithologies ($\delta^{13}C$ range: $+1.7$ to $+13.4$ ‰; median = $+9.8$ ‰; Šmejkal, 1978, 1984). Our sediment's $\delta^{13}C$ mean value likely fingerprints siderite, which was the only carbonate phase detected via XRD. Yet, other relatively more soluble carbonate phases, such as dolomite and calcite, might be present in small proportions at the lake floor because they occur with siderite in the claystone sediment source. These would account for only $\leq 0.2$ wt. % (i.e., the L.Q. of our semi-quantitative XRD analyses). The range of estimated isotopic C values of the $CO_2$ flux from the sediments to the water column is between $-3.0$ and $-4.2$ ‰ (Fig. 3d). The contributions of $CO_2$ derived from OM degradation, carbonate mineral dissolution and any plausible methanotrophic activity thus produces isotopic C values in the lake bottom water's $\Sigma CO_2$ that match those of the magmatic-derived $CO_2$ emissions (Weinlich et al., 1999; Dupalová et al., 2012).

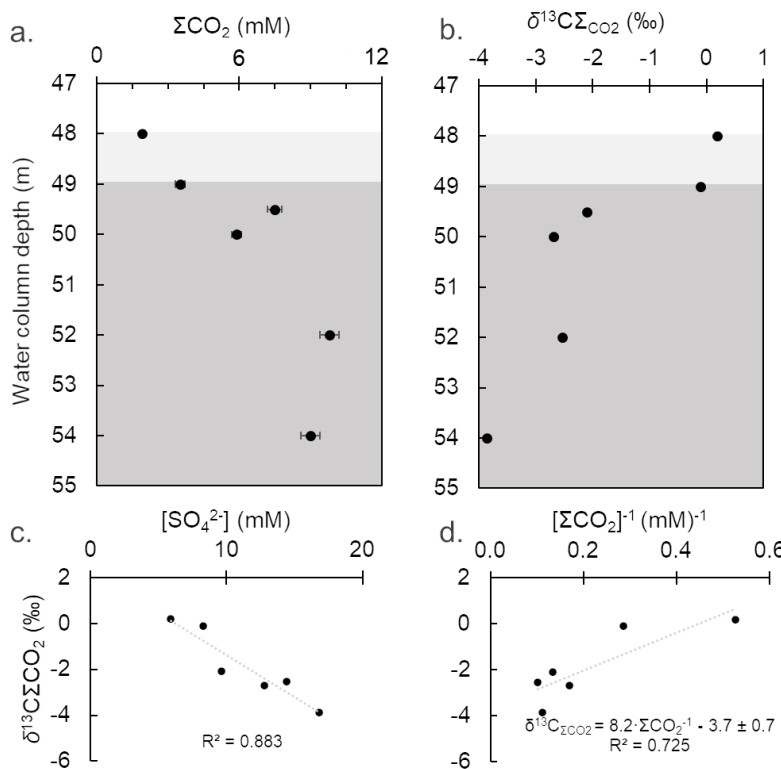

Figure 3. Depth−dependent variation in total dissolved inorganic carbon ($\Sigma CO_2$) (a) and its $\delta^{13}C$ (b) in the oxygen depleted bottom water column of Lake Medard (centre). Background grey colour code as in Fig. 2. There is negative correlation ($R^2 = 0.883$) between the $\delta^{13}C$ values and dissolved $SO_4^{2-}$ concentrations (c). A Keeling-style plot ($\Sigma CO_2$ vs. $\delta^{13}C_{\Sigma CO2}$) was used to deduce the isotopic C signature of the combined $CO_2$ flux at the sediment water interface, i.e., the intercept (d).

The mixing factor in a simple linear mixing model was calculated after Phillips and Gregg (2001). Accordingly, it could be established that dissolution of sedimentary carbonates contributes $70 \pm 5$ % of the dissolved inorganic carbon, with the remaining fraction being $CO_2$ from organic matter heterotrophy (35 to 25 %). The influence of isotopically light $CO_2$ derived from the oxidation of diffused methane is negligible, and any contribution of $CO_2$ from the magmatic source cannot be estimated because of the similar isotopic values. The implication for environmental/early diagenetic interpretations of this approach is that if siderite is formed in the lake sediments, it displays a significant $\delta^{13}C$ offset (i.e., between $+9.1$ and $+10.9$ ‰) from the values of the $\Sigma CO_2$

reservoir of the lake's floor. Alternatively, siderite could rather be a re-deposited mineral sourced from the Miocene claystone

lithology that provided detrital material to the mine spoils and modern lake system. We will revisit siderite under Sect. 4.6.1.

## 4.3 Nitrogen, iron and sulfur species in water column with functional annotations on the planktonic prokaryote community

### 4.3.1 Nitrogen species transformations and the N-utilizing prokaryotes

Dissolved nitrate ($NO_3^-$) concentrations across the dysoxic hypolimnion were approximately 25 µM and decrease about 28 %

towards the anoxic monimolimnion. This decrease is accompanied by an increase in ammonium from 16 µM to up to 142 µM

(Table 1; Fig. 4a). Similar behaviour of reactive N species were described in other ferruginous water columns (e.g., Michiels

et al., 2017; Lambrecht et al., 2018).

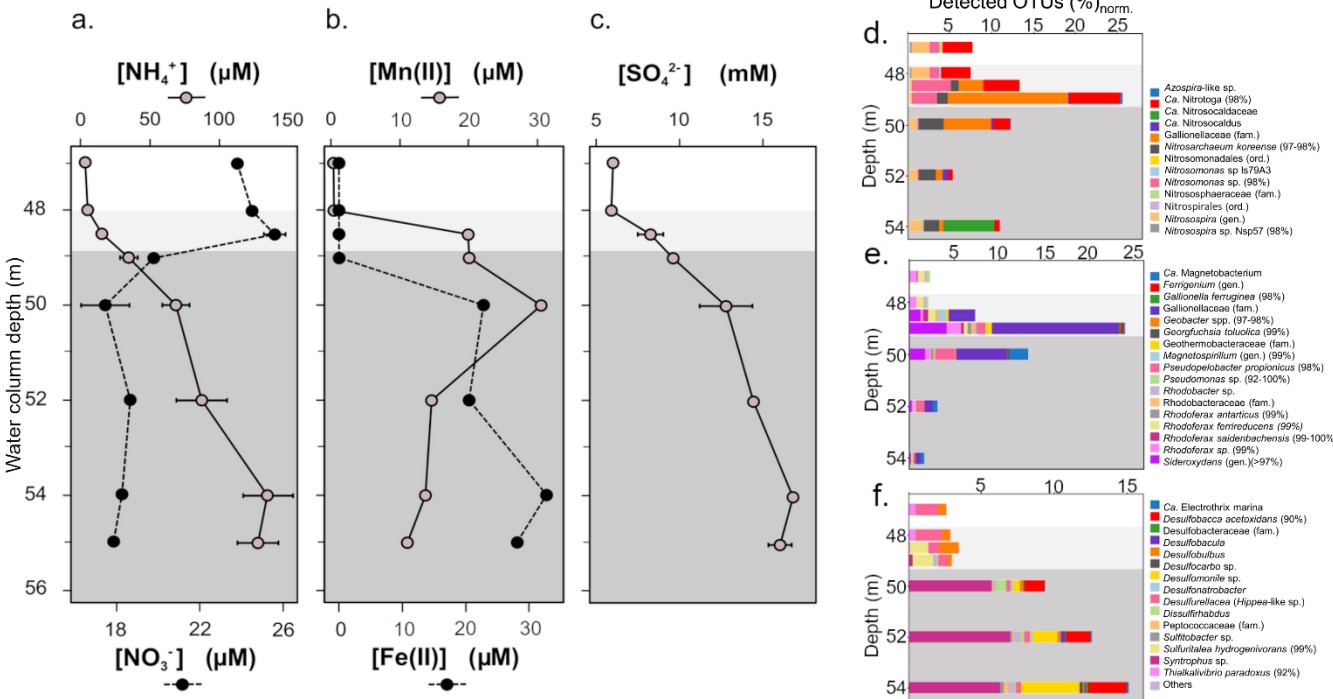

**Figure 4. Measured dissolved concentrations of nitrate and ammonia (a), manganous manganese and ferrous iron (b), and (c) sulfate**
**in the bottom water column of Lake Medard (centre). The right-side panels show the corresponding, normalized abundance of putative planktonic nitrogen (d), iron (e), and sulfur-utilizing (f) prokaryotes. Sequences were classified based on best BLAST hit results, and bacteria/archaea were identified based on phylogenetic affiliations. Normalization was with regard to total amplicon reads in each sample. Grey background colours are based on the Eh profile (Fig. 2), and here also indicate redox stratified niches. The sequences were deposited in the European Nucleotide Archive EMBL-EBI (PRJEB47217).**

The relative abundance of 16S rRNA gene sequences that can be ascribed to N-utilizing planktonic prokaryotes (Fig. 4d)

indicates that *Nitrosomonas*-like species (95 to 98 % gene similarity) are in the dysoxic hypolimnion at a low normalized

abundance which increases at the redoxcline. Here *Nitrosomonas*-like species may conduct the first and rate-limiting step in

nitrification, i.e., $NH_3$ oxidation (Lehtovirta-Morley, 2018). The second nitrification step, nitrite oxidation to $NO_3^-$, could be

exerted predominantly by species exhibiting similarity (98% gene sequence) to *Ca.* Nitrotoga (98 % gene sequence similarity).

*Ca.* Nitrotoga was detected in all our samples but exhibited a higher normalized abundance (up to 9 %) at the redoxcline (Fig. 4d).

Among the relatively abundant, $NH_3$-oxidizing microbes detected is an archaeon related to *Nitrosarchaeum koreense* (97-98 % gene similarity). This archaeon has higher normalized abundances in the ferruginous waters below the redoxcline (Fig. 4d, also Supplement 2: Krona chart). Its distribution across the redox gradient is at odds with the fact that *N. koreense* has been

previously suggested to be an aerobe (Jung et al., 2018). Similarly, members of the Candidatus Nitrosocaldaceae family (similarity 78-82 % in 387 bp) appeared to be present in the anoxic zone of the water column, despite the best studied member of this family, *Ca.* Nitrosocaldus, being reported as displaying an aerobic lifestyle (de la Torre et al., 2008). The archaeal family has heterogeneous metabolic capabilities and is capable of oxidizing ammonia to nitrite (Luo et al., 2021). Our observation could make the case for niche differentiation linked to high loads of dissolved metal concentrations conferring a

competitive advantage to these archaea (e.g., Gwak et al., 2019). Alternatively, the $NH_3$-oxidizing archaea detected predominantly in the ferruginous waters possess a yet to be explored tolerance to anoxia (see Mußmann et al., 2011). For instance, *Ca.* Nitrosocaldus encodes a pyruvate:ferredoxin oxidoreductase that is rather uncommon among aerobic ammonia oxidizers (Daebeler et al., 2018), but it is encoded by most anaerobes able to catalyze the decarboxylation of pyruvate to form acetyl-coenzyme A (Chabrière et al., 1999).

The maximal relative abundance of an *Azospira*-like microorganism (95 % similarity) coincides with the peak of relative abundance of members of the Gallionellaceae family at 49 to 50 m depth (Fig. 4d, Supplement 2). Like *Gallionella* spp., *Azospira* also possess dissimilatory N and Fe-based metabolisms capable of yielding dinitrogen ($N_2$)(Mattes et al., 2013). $N_2$ production probably accounts for a fraction of the apparent nitrogen loss observed when the dissolved reactive $NH_4^+$ and $NO_3^-$ levels are compared across their counter gradients (Table 1; Fig. 4a). Nitrite ($NO_2^-$), an intermediate between $NO_3^-$ and $NH_4^+$,

can also accumulate. Yet, concentration profiles of such intermediate remain to be accurately resolved in the increasingly saline (high chlorine) bottom water column of Lake Medard.

When contrasted, the counter gradients of reactive nitrogen species and those of other dissolved bioactive chemical species suggest that while metabolizing nitrogen, the planktonic prokaryote community could also impact the cycles of Fe and S (e.g., Jewell et al., 2016, 2017; Starke et al., 2017). These cycles in the aqueous system under consideration are likely interlinked

through microbial mediation in the generalized Reactions (1−3), but note that intermediate $NO_2^-$ may also act as a relevant Fe(II) oxidant in this $O_2$-depleted system (Klueglein et al., 2014):

$$10Fe^{2+} + 2NO_3^- + 24H_2O \rightarrow 10Fe(OH)_3 + N_2 + 18H^+ \quad (1)$$

$$NO_3^- + 8Fe^{2+} + 21H_2O \rightarrow NH_4^+ + 8Fe(OH)_3 + 14H^+ \quad (2)$$

$$5HS^- + 8NO_3^- \rightarrow 5SO_4^{2-} + 4N_2 + 3OH^- + H_2O \quad (3)$$

Reaction 1 proceeds mixotrophically, usually requiring a favourable organic co-substrate, whereas reactions 2 and 3 likely proceed under the influence of chemolithotrophic Fe(II) and/or S oxidizing nitrate reducers. Due to energetic considerations,

these microorganisms are known for having metabolic advantages under ferruginous conditions over solely denitrifying organisms (see Robertson and Thamdrup, 2017). Reaction 3 is known to proceed at rather low sulfide levels (Brunet and Garcia-Gil, 1996; Barnard and Russo, 2009), such as those characterizing the monimolimnion of our study site ($\leq 0.3$ μM).

In the following section, to further investigate details on the microbial ecology of the bottom ferruginous waters of Lake Medard, we consider the concentration profiles of dissolved Fe and Mn along the redoxcline. Concentrations of these dissolved metals are operationally defined as the combined ionic and colloidal fractions that passed the 0.22 μm cut-off of membrane filters. By co-evaluating the dissolved Fe and Mn concentration trends we pursue further insight on the mechanism procuring and/or consuming these metals in the stratified water column (Davidson, 1993). A 16S rRNA gene abundance profile of known

iron-utilizing prokaryotes also permitted inferences on what members of the microbial community could be exerting a direct dissimilatory (catabolic), or indirect (via electron transfer) control over the concentration trends of these metals across the redox gradient.

### 4.3.2 Dissolved divalent manganese and iron and the Fe-utilizing prokaryotes

Dissolved manganese concentrations ([Mn]) peaked at about 50 m-depth (Table 1). Below this depth, [Mn] showed a steady

decrease (Fig. 4b). This trend indicates that in the water column the 50-m depth acts as a point source of Mn(II) (Davison, 1993). Divalent iron is also present at a similar concentration magnitude at this depth (Fig. 4b, Table 1), and it can readily act as a reductant of most particulate Mn(IV) settling down from the mixolimnion (Lovley and Phillips, 1988; Myers and Nealson, 1988), Reaction (4):

$$2Fe^{2+} + MnO_2 + 1.5H_2O \rightarrow Fe_2O_3 \cdot 0.5H_2O + Mn^{2+} + 2H^+ \tag{4}$$

Accordingly, a substantial fraction of the Fe(II) diffusing upwards from the monimolimnion could be re-oxidized or cycled back to Fe(III) within the peak zone of Mn(IV)-reduction at 50 m-depth (Fig. 4b). Mn(II) yielded during iron oxidation can then be transported both upwards and downwards away from the 50 m-depth source point by eddy diffusion (Fig. 4b; Davidson, 1993). The internal bottom water column cycling of iron also reflects on the concentration gradient of dissolved phosphate (Table 1). Solubilization of this oxyanion is thought to be regulated by reduction of its particulate Fe(III) sinks. Upward

diffusion, however, allows for dissolved phosphate to be re-complexed back onto ferrihydrite-like phases that precipitate above the redoxcline, where its concentrations decrease (Table 1).

Contrary to Mn, dissolved Fe concentration ([Fe]) increased steadily downwards, and its global maximum is reached at about 54 m-depth in the monimolimnion (Table 1). Immediately below this depth, [Fe] decreases by about 14 %. This decrease can be consistently observed in other anoxic zones of the lake (Petrash et al., 2018), and hints to Fe(II) and reduced S co-

precipitation as metastable acid volatile monosulfide (FeS; e.g., mackinawite). The dissimilar distribution of divalent Fe and Mn in the bottom water column (Fig 4b) reflected reductive dissolution being much more effective for the sinking manganic particulate than for ferric particulate matter.

Our planktonic prokaryote analysis showed that above the redoxcline the relative abundance and taxonomic richness of known iron-respiring prokaryotes were low and dominated by species closely related to the β-Protebacterium *Rhodoferax* (99-100 % gene similarity) (Fig. 4e, Supplement 2). Other sequences that can be functionally affiliated to Fe(III)-reduction in the dysoxic hypolimnion included a bacterium with between 92 and 100 % gene similarity to unclassified *Pseudomonas* spp. (Fig. 4e). *Pseudomonas* could grow by coupling the oxidation of dihydrogen ($H_2$) to the reduction of Fe(III) (Lovley et al., 2004). Bioutilization of manganese by *Pseudomonas* species—both in oxidation and reduction reactions—has been also reported (e.g., Tebo et al., 2005; Geszvain et al., 2011; Lovley, 2013; Wright et al., 2018). Other bacteria that may influence the aqueous manganese cycling to indirectly affect that of dissolved iron belong to the family Hyphomicrobiaceae (e.g., Northup et al., 2003; Spilde, et al., 2005). Three OTUs with significant homology to purportedly Mn(II)-oxidizing members of the family (*Hyphomicrobium hollandicum*, *H.* sp. KC-IT-W2, and *Devosia* sp.) exhibited maximal relative abundances above the redoxcline, but were notably absent from deeper monimolimnial waters (Supplement 2).

As previously mentioned, we detected a sharp increase in the relative number of microaerophilic Fe(II)-oxidizing *Gallionella* species at the redoxcline and immediately below it. They accounted for up to ~24 % of the total normalized gene reads (Fig. 5b). The increase in relative abundance of *Gallionella* spp. coincided with an increase in sequences related to *Sideroxydans* spp. (Fig. 4e). These latter microaerophiles can also use Fe(II) as an energy source for chemolithotrophic growth with $CO_2$ as the sole carbon source (Emerson and Moyer, 1997). Other different physiological groups of putative Fe(II)-oxidizing microorganisms detected above and near-redoxcline samples included anoxygenic phototrophic and nitrate-reducing species (*Magnetospirillum* and *Ferrigenium*; Fig. 4e, Supplement 2), and *Azospira*-like species (Khalifa et al., 2018; Mattes et al., 2013; Dziuba et al., 2016).

Prokaryotes that can adapt their metabolic strategies to the less pronounced geochemical gradients prevailing at the monimolimnion became predominant below the redoxcline. Among them is a bacterium distantly related (89 % identity in 399 bp) to *Candidatus* Magnetobacterium (Lin et al., 2014)*,* which relative abundance substantially increases at the 50-m depth (Fig 4e). At this level, our gene sequence reads also included an OTU closely related to *Georgfuchsia toluolica*, a strictly anaerobic β-Proteobacterium capable of degrading aromatic compounds with either Fe(III) or $NO_3^-$ as electron acceptors (Weelink et al., 2009). HSs derived from lignite degradation contain abundant aromatic compounds (Wang et al., 2017).

Towards the SWI, important members of the Fe−respiring community were those from the family Geobacteraceae, which can use insoluble Fe(III) and/or Mn(IV) as electron acceptors, and acetate, formate, alcohols, aromatics, and $H_2$ as electron donors (Weber et al., 2006; Lovely and Holmes, 2021). The abundance of *Geobacter* species peaked around the maximum of Fe(III) reduction within the monimolimnion, at about 54 m depth. Here, acetate availability is also relatively high (Fig. 2b). The relative proportion of *Geobacter* spp. increased in parallel with that of their phylogenetically associated *Pseudopelobacter propionicus*, which is a fermentative acetogen that can only indirectly mediate Fe(III) reduction. A possible ecological interaction between *P. propionicus* and *Geobacter* species at the interface of redox boundaries in sedimentary environments has been already reported by Holmes et al. (2007) and Butler et al. (2009).

### 4.3.3 Dissolved sulfate and the S-utilizing prokaryotes

The dissolved sulfate concentration ($[SO_4^{2-}]$) changed at the redoxcline, where it increased from 6.0 to 16.8 mM (Fig. 4c). At the lower monimolimnion, a decrease in $[SO_4^{2-}]$ coincided with a decrease of [Fe(II)] (Table 1, Fig. 4b-c). In the lower monimolimnion, we detected an increase in the number of taxonomic groups and relative abundances of known sulfate reducers (Fig. 4f). Their by-product sulfide, however, does not accumulate in the ambient waters ($[H_2S + HS^-] \leq 0.30$ µM). The lack of substantial dissolved sulfide towards the SWI and the similar hydrochemical responses of both Fe(II) and $[SO_4^{2-}]$ could be considered circumstantial evidence for FeS precipitation, with another being $\delta^{56}Fe$ values that increased across the redoxcline and towards the SWI (Petrash et al., 2022). Additional insight on this and other mechanisms of sulfate turnover operating in the water column was sought by evaluating the distribution of S-utilizing prokaryotes.

Our 16S rRNA gene analyses revealed a rather low number of taxonomic groups of sulfur-respiring bacteria at the dysoxic hypolimnion (Fig. 4f; also Supplement 2). Here OTU assignments show mostly a few uncultured members of the newly proposed order Desulfobulbales of the phylum Desulfobacterota (previously δ-Proteobacteria, Waite et al., 2020; Ward et al., 2021))(Fig. 4f). Some species within Desulfobulbales require intermediate S or thiosulfate for heterotrophic growth but can also gain energy from pyruvate fermentation (Flores et al., 2012). *Desulfobulbus* spp. can perform dissimilatory sulfate reduction via the incomplete oxidation of lactate, but *D. propionicus* is known for efficiently conducting disproportionation of elemental sulfur (Lovley and Phillips, 1994). Pyruvate, as lactate, was found below our detection limits across the bottom water column; where sequences distantly related to *D. propionicus* (91 % similarity in 428bp) appeared to be particularly abundant (Fig. 4f; Supplement 2). Probably important for the microbial sulfur cycling at this level of the water column is also a γ-Proteobacterium from the order Chromatiales that has 92 % gene identity in 424 bp to *Thioalkalivibrio paradoxus* (Fig. 4f). *T. paradoxus* is a chemolithoautotrophic sulfur-oxidizing bacterium that can use both reduced and intermediate S compounds for C fixation (Berben et al., 2015).

There were gene sequences that could be confidently ascribed to the facultative S-utilizing autotroph *Sulfuritalea hydrogenivorans* (3 OTUs with $\geq$ 97 % identity in 424 bp) at the redoxcline. The abundance of *S. hydrogenivorans* increased in parallel to a decrease in the *T. paradoxus*-like bacterium, which suggests that the latter may be at a disadvantage and limited by organic C fixation under the specific hydrochemical conditions prevailing at the redoxcline. Such conditions may include, for instance, an abundance of aqueous intermediate S species. Under such conditions, *S. hydrogenivorans* can outcompete the *T. paradoxus*-like bacterium by oxidizing, under denitrifying conditions, either thiosulfate, $S^0$ and/or $H_2$ for C fixation (Kojima and Fukui, 2011; Kojima et al., 2014).

At the redoxcline, the relative abundance of species distantly related to fully sequenced Desulfobulbales also increased to ~1.7 % (Fig. 4f). Below the redoxcline, our genomic data revealed a progressive development of a more diverse sulfur-respiring bacterial population (Fig. 4f). This was dominated by many relatively rare taxa and a few abundant lineages (Supplement 2), and with a punctuated dominance of species distantly related to *Desulfobacca acetoxidans* (90 % identity in 432 bp). *D. acetoxidans* oxidizes acetate using either sulfate, sulfite ($SO_3^{2-}$) or thiosulfate ($S_2O_3^{2-}$) as electron acceptors, but not $S^0$ (Oude

Elferink et al., 1999). The *D. acetoxidans*-like prokaryote first appeared at 49 m depth but became dominant towards the SWI,
together with *Desulfomonile*-related species (96% identity in 432 bp). *Desulfomonile*-related species could be also responsible for the previously noticed pyruvate depletion, but here they may be also thriving chemolithoautotrophically with $S_2O_3^{2-}$ as terminal electron acceptor (DeWeerd et al., 1990; Sun et al., 2001). Other prokaryotes probably gain energy out of intermediate S disproportionation in the anoxic monimolimnion. These may include uncultured species distantly related to *Desulfatibacillum* and *Dissulfurirhabdus* (2 OTUs with 87 % identity in 428 bp). The presence of the genus *Sulfitobacter*
across the aqueous redox gradient and into the monimolimnion (Fig. 4f) points to a continuous genetic potential for chemolithotrophic sulfur oxidation across the entire bottom water column.

## 4.4 $\delta^{34}$S and $\delta^{18}$O isotope values of dissolved sulfate

### 4.4.1 A proxy for disproportionation

Water column $\delta^{18}O_{SO4}$ values ranged from +2.0 to +4.0 ‰, with corresponding $\delta^{34}S_{SO4}$ values ranging between +10.9 and +13.4
585 ‰ (Table 1, Fig. 5a-b). The depth profiles of these isotopes in the water column reveal that dissolved sulfate in the anoxic monimolimnion is enriched in $^{18}$O (Fig. 5a-b) relative to the dysoxic waters. Despite the moderate decrease in $[SO_4^{2-}]$ towards the SWI (Fig. 4c) no significant sulfur isotope fractionation was registered. The $\delta^{34}S_{SO4}$ values were only weakly correlated with $[SO_4^{2-}]$ ($R^2 = 0.16$).

The ambient bottom waters had a narrow $\delta^{18}O_{H2O}$ range of values: −6.1 to −6.7 ‰. This is consistent with ongoing meteoric
water-rock interactions and rather limited evaporation effects (cf. Noseck et al., 2004; Pačes and Šmejkal, 2004; Dupalová et al., 2012). By applying the expression first proposed by Taylor et al. (1984) to relate the $\delta^{18}$O values of dissolved $SO_4^{2-}$ and those of ambient waters, we deduced that the oxygen isotope effect ($^{18}\varepsilon_{SO4\text{-amb. wat.}}$) in our bottom waters ranged between +9.3 and +10.7 ‰. This range was calculated under the assumption that equilibrium of oxygen isotope exchange between cell-internal sulfur compounds and ambient water dominates over kinetic oxygen isotope fractionation (Fritz et al., 1989; Brunner
et al., 2005). The estimated $^{18}\varepsilon_{SO4\text{-amb. wat.}}$ is within the range experimentally derived by Brunner et al. (2005) while using similarly $^{18}$O-depleted ambient waters. It is also within the range observed in studies of S disproportionation reactions generally proceeding under anoxic conditions (e.g., Böttcher et al., 2001; 2005). Yet, it is lower than $^{18}\varepsilon_{SO4\text{-amb. wat.}}$ values reported by Bottrell and Newton (2006) in biotic experiments with excess reactive Fe(III) species—i.e., +16.1 to +17.5 ‰. Therefore, our $^{18}\varepsilon_{SO4\text{-amb. wat.}}$ could result from the superimposition of the isotope signals of sulfate reduction, sulfide re-oxidation and
intermediate sulfur disproportionation. It follows that the sulfur disproportionation in the bottom waters of Lake Medard most likely results from multiple biologically mediated reactions involving not only reactive iron, but also reducible Mn stocks in the sediments (Böttcher et al., 2001). As further discussed below, the anoxic sediments contain a low—i.e., compared with Fe(III)-counterparts—yet still measurable abundance of Mn(IV) (Table 2).

A microbially mediated/induced sulfur disproportionation mechanism that considers reactive iron forms present in the sediments, also involves Mn(IV,III) reduction, and is consistent with formation of FeS in the monimolimnion. It can then be described by the following reactions (Reactions 5−7, after Thamdrup et al., 1993; Böttcher and Thamdrup, 2001):

$$3S^0 + 2FeOOH \rightarrow SO_4^{2-} + 2FeS + 2H^+ \tag{5}$$

$$4S^0 + 4H_2O + 3FeCO_3 \rightarrow SO_4^{2-} + 3FeS + 2H^+ + 3H_2CO_3 \tag{6}$$

$$3S^0 + Mn_3O_4 + 2H^+ \rightarrow SO_4^{2-} + 2HS^- + 3Mn^{2+} \tag{7}$$

Although not shown in the rather simplified reaction set listed above, $S^0$ may well be a different intermediate sulfur species such as $S_2O_3^{2-}$ and/or $SO_3^{2-}$ (e.g., Holmkvist et al., 2011). The intracellular isotope exchange of sulfite with anoxic ambient waters has been proven to produce an oxidized $SO_4^{2-}$ product that is enriched in $^{18}O$ relative to precursory thiosulfate and/or sulfite. This enrichment displays only a minor change, if any, in its corresponding S isotope composition (e.g., Böttcher et al., 2005; Johnston et al., 2014; Bertran et al., 2020; see Table 1). In line with this assertion, at the monimolimnion there is negligible sulfur isotope fractionation accompanying the recorded fractionation of oxygen isotope. Yet, our data recorded a small, but significant reverse sulfur isotope effect (+2.2 ‰) at the upper hypolimnion (Fig. 5a: 48 m depth). This isotope effect could be ascribed either to abiotic or biotic oxidation processes of intermediate S species occurring at that level of the water column (see Zerkle et al., 2016, their table 1).

### 4.4.2 Insights on intermediate sulfur oxidation

A cross-plot of the $\delta^{34}S_{SO4}$ vs. $\delta^{18}O_{SO4}$ values along the redoxcline as well as those of all the possible geogenic sources of sulfate entering the lake system (see also Appendix B: Fig. B2) is shown in Fig. 5c. Analysis shows that the $\delta^{34}S_{SO4}$ values of the redox stratified Lake Medard fingerprint a mixed geogenic-sulfate source. Fig. 5d offers further detail and linear regressions of the covariation in the $\delta^{34}S_{SO4}$ vs. $\delta^{18}O_{SO4}$ cross-plot. The slopes of such linear regressions can be used to roughly estimate sulfate reduction rates (SRR; after Böttcher et al., 2001; Brunner et al., 2005, among others). For assessing our SRR, it is reasonable to assume that the initial S and O isotope composition linked to dissolved sulfate was within the range of the modern nearby acidic drainage (i.e., +2.9 ± 0.1 ‰ for $\delta^{34}S_{SO4}$; 0.0 ± 0.5 ‰ for $\delta^{18}O_{SO4}$), and similar to the initial composition of sulfate in the pit-lake prior to reclamation/flooding (Fig. B2, Appendix B). The residual isotope composition would then be that of dissolved sulfate in the bottom anoxic waters.

In agreement with the lack of accumulation of sulfide in the monimolimnion, our SRR estimation is consistent with slow gross but not net $SO_4^{2-}$ reduction (see Böttcher et al., 2004). The SRR is apparently slower at the monimolimnion (i.e., higher slope) than in the hypolimnion. This is at odds, however, with the higher taxonomic abundance of sulfate reducers that we detected near the SWI (Fig. 4f). The decrease in dissolved sulfate concentration (Table 1) does not lower the slope of the linear regression. It means that the sulfur isotope ratio of dissolved sulfate evolves slower relative to corresponding change in oxygen isotope ratio. This result is likely due to sulfate regeneration through microbial sulfide oxidation; with oxygen isotope exchange

with ambient water occurring via an intracellular oxidation step of intermediate sulfur (Böttcher et al., 2005; Bertran et al., 2020). Under the low organic substrate availability characterizing the bottom waters examined here (Fig. 2b), sulfate reducers capable of disproportionation (e.g., bacteria from the order Desulfobulbales) can maintain intracellular concentrations of sulfite. This manifested geochemically as the rapid change in water column $\delta^{18}O_{SO4}$ (Böttcher et al., 2005; Antler et al., 2013).

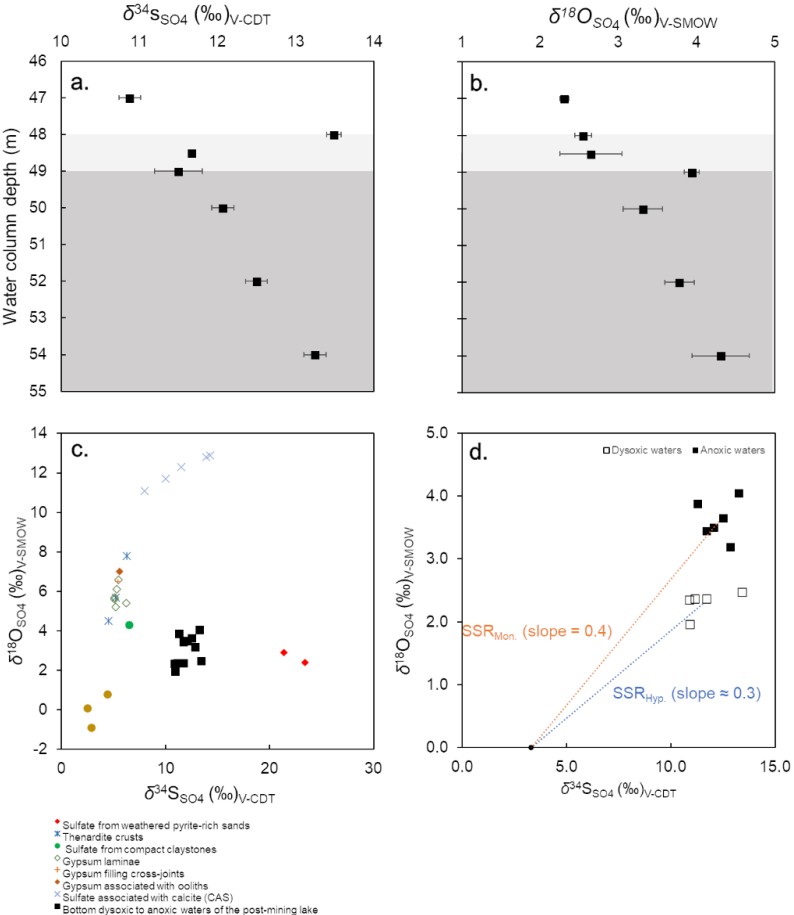

**Figure 5.** The bottom water column $\delta^{18}O_{SO4}$ and $\delta^{34}S_{SO4}$ values (a-b). Grey background colour code as in Fig. 2. Also, a cross-plot of these values in the water column vs. those of all possible sources of dissolved $SO_4^{2-}$ to the modern lacustrine system (c). The coupled sulfur and oxygen isotope-constrained slopes of the linear regressions provide a rough estimation of the SRR. The regressions considered the $\delta^{18}O_{SO4}$ and $\delta^{34}S_{SO4}$ of the acidic drainage as the initial isotope composition of dissolved sulfate immediately after flooding (see text for details).

### 4.5 Insights from solid phase analyses

#### 4.5.1 Semi-quantitative X-ray diffraction

XRD analyses of the anoxic sediments show that most detrital minerals were sourced from the Miocene claystone lithology (Appendix A). These detrital phases include kaolinite, quartz, K-feldspar, the $TiO_2$ polymorphs rutile and anatase, and analcime ($NaAlSi_2O_6 \cdot H_2O$). Minor constituents of the anoxic lake sediments that can also be quantified include gypsum, siderite, and pyrite. Gypsum and siderite were in similar abundances in the upper anoxic sediments (~3 to 4 wt. %), whereas pyrite accounts for a maximum

of 0.5 wt. % of their total mineralogy (Fig. 6a). Given that the diffraction peaks of major and minor mineral sediment constituents mask those of Fe(III)- and Mn(IV)-oxyhydroxides, the abundances of these reactive phases were determined 665 through a sequential extraction scheme that also targets Fe(II)- and Mn(II)-bearing carbonates.

#### 4.5.2 Sequential extractions of reactive iron

The relative concentrations of highly reactive Fe-bearing species (Fe$_{HR}$) in the upper anoxic sediment pile are displayed in Figure 6b. The Fe$_{HR}$ sediment pool is defined as that capable of reacting (upon reductive dissolution) with dissolved sulfide to precipitate metastable FeS, which can later be stabilized to pyrite (Canfield and Berner, 1987; Canfield, 1989). We also report 670 here Fe(II) bound to the pyrite fraction (Fe$_{py}$), and the total iron (Fe$_T$) in the sediments (Poulton and Canfield, 2005).

Our $Fe_{HR}$ was dominated by poorly crystalline phases (Feh), such as ferrihydrite and/or lepidocrocite (γ-FeOOH). These $Fe_{HR}$ mineral fractions were followed in abundance by that of Fe(II)-bearing carbonates (FeC) (Fig. 6b, Table 2). A significant increase in the FeC is observed with increasing depth (Fig. 6b). This may be indicative of partial dissolution of some Fe(II)-bearing carbonates at the SWI, or the result of soluble Fe(II) binding reactive Fe-carbonates deeper into the sedimentary pile. To clarify on this matter, we discuss the petrographic features and C isotope values of siderite in Sect. 4.6.1

Absolute Fe(III) concentrations ascribed to Feh phases increase towards the bottom of our 8 cm depth core, but their abundance, relative to total iron, decreases downwards (Table 2). The extraction step for Feh also extracts Fe(II) bound to monosulfides (Kostka et al., 1995; Scholz and Neumann, 2007). These metastable phases yielded ≤ 0.04 wt. % according to our acid volatile sulfur (AVS) extraction. However, possible rapid oxidation of AVS particles during sampling of the sediments makes it challenging to assess their actual abundance and mineralogy (Schoonen, 2004). It thus appears that the Feh abundance at the top of the sediments (Fig. 6b) is mostly comprised of poorly crystalline oxyhydroxide.

The iron extracted from crystalline Fe(III)-bearing phases (such as goethite) increased from $2.7 \pm 0.4$ % in the first 6 cm to up to 17.8 % of the $Fe_T$ at the 6 to 8 cm interval (Table 2). Fe concentrations bound to pyrite (Table 2, Fig. 6b) constituted up to ~21 % of the $Fe_T$ in the upper sediments (i.e., ~0.8 bulk wt. %), and showed a general downwards decreasing trend contrasting with that of crystalline Fe(III)-bearing phases. From these observations, the 0 to 6 cm depth interval is confidently considered as recent anoxic lake deposition, whilst below 6 cm are sediments that were deposited in the shallow pit lake now undergoing alteration under the redox dynamics of the present-day lacustrine system.

The $Fe_{Py}/Fe_{HR}$ ratio in the 8 cm long sediment profile accounts for the extent to which the Fe pool was pyritized. The ratio is < 0.35 and decreases downwards (Table 2). When considering that the corresponding $Fe_{HR}/Fe_T$ ratios were consistently ≥ 0.71, the results from our sequential extraction scheme applied to iron are consistent with a persistent ferruginous but not euxinic redox state of the now anoxic sediments (Poulton and Canfield, 2011). Variability of $Fe_{py}/Fe_{HR}$ and $Fe_{HR}/Fe_T$ with depth of the sediments reflects the redox dynamics after flooding and establishment of a chemically distinct monimolimnion.

From combining results from $Fe_{HR}$ partitioning in the sediments (Table 2) and the dissolved Mn(II) and Fe(II) concentration trends (Fig. 4b), we can now strengthen an earlier deduction that Fe(II) sourced from reductive dissolution processes in the upper sediments diffuses upwards, where it rapidly reacts with residual $O_2$ in the vicinity of the redoxcline to form metastable Fe(III)-bearing particulate phases. Most of the iron in such amorphous to nanocrystalline ferrihydrite-like aggregates are deposited on the lake's anoxic floor. From the anoxic floor, iron is resolubilized back into the monimolimnion. Yet a fraction of it stabilizes upon burial as goethite (α-FeOOH) or is bound to the surfaces of reactive carbonates. Another fraction is pyritized through reactions involving elemental sulfur and/or polysulfide near the SWI (Fig. 6b) (Shoonen, 2004 for details). Indeed, we observed that in the upper sediment the partitioning of the reactive iron into these minerals can be swiftly altered by short-lived variations (± 150 mV) in the redox potential of the bottom water column. Variations in the relative proportions of reactive iron minerals also control the distribution of siderophile redox sensitive elements in the sediment pile (Umbria Salinas et al., 2021).

**Table 2. Partitioning of reactive iron and manganese species in the lacustrine sediments (0−8 cm depth).**

| Depth (cm) | $Fe_{HR}$ | | | | | $Fe_T$ | $Fe_{HR}/Fe_T$ | $Fe_{py}/Fe_{HR}$ | $Mn_{HR}$ | | | | | $Mn_T$ |
|---|---|---|---|---|---|---|---|---|---|---|---|---|---|---|
| | Exch. | $Fe(II)_{CO3}$ | Poorly cryst. $Fe_{ox}$ | Cryst. $Fe_{ox}$ | $Fe(II)_{py}$ | | | | Exch. | $Mn(II)_{CO3}$ | Poorly cryst. $Mn_{ox}$ | Cryst. $Mn_{ox}$ | $Mn(II)_{py}$ | |
| 0−2 | 14.7 ±2.4 | 283.2 ±45.3 | 302.8 ±48.3 | 29.6 ±4.7 | 189.9 ±3.6 | 1024.4 ±6.2 | 0.80 | 0.23 | 1.0 ±0.1 | 18.9 ±1.5 | 3.6 ±0.3 | ≤ 0.15 | ≤ 0.02 | 23.7 ±0.4 |
| 2−4 | 20.3 ±3.2 | 294.7 ±47.2 | 308.6 ±49.4 | 33.7 ±5.4 | 224.6 ±2.4 | 1067 ±10.5 | 0.83 | 0.25 | 3.8 ±0.3 | 9.4 ±0.8 | 3.3 ±0.3 | ≤ 0.16 | ≤ 0.03 | 13.2 ±0.2 |
| 4−6 | 28.8 ±4.6 | 365.7 ±58.5 | 263.6 ±42.2 | 21.5 ±21.5 | 128.9 ±2.9 | 1142 ±10.9 | 0.71 | 0.16 | 2.7 ±0.2 | 11.6 ±0.9 | 2.8 ±0.2 | ≤ 0.13 | ≤ 0.03 | 14.8 ±0.2 |
| 6−8 | 24.4 ±3.9 | 689.3 ±110.3 | 335.7 ±53.7 | 373.7 ±59.8 | 117.9 ±4.2 | 2097 ±22.8 | 0.73 | 0.08 | 1.2 ±0.3 | 7.3 ±0.6 | 1.9 ±0.2 | ≤ 0.17 | ≤ 0.02 | 9.5 ±0.1 |

[#] Sediment density is estimated in 2.71 g·L$^{-1}$ with a porosity of 40 %; error of the measurement (n=4) is ± 16% [Fe] and ± 8% [Mn]; in µmol·cm$^{-3}$.

### 4.5.3 Sequential extractions of reactive Mn−bearing phases

Results from our extraction scheme applied to Mn (i.e., after Slomp et al., 1997; Van Der Zee and Van Raaphorst, 2004) show that the $Mn_{HR}$ pool in the anoxic sediment was dominated by Mn(II)-bearing carbonates (MnC) (Fig. 6c, Table 2). The carbonates were relatively more abundant at the SWI but in contrast to FeC, showed no clearly defined concentration trend in the upper sediments (Table 2, Fig. 6b-c). A declining trend downwards is clear for the proportions of easily reducible Mn(IV)-bound to poorly crystalline phases, such as δ-$MnO_2$. These were extracted by diluted HCl (Fig. 6c, Table 2) (Slomp et al., 1997). Reducible Mn associated with more crystalline oxyhydroxide forms are extracted by dithionite (Canfield et al., 1993), but concentrations of this fraction might be sourced from crystalline Fe(III)-oxyhydroxides that can either sorb Mn(II) or structurally incorporate Mn(III) (Namgung et al., 2020). Irrespective of its source, the highly crystalline Mn-bearing fraction in our sediment comprises ≤ 0.2 wt. % of $Mn_T$ (Table 2). The concentrations of Mn(II) bound to sulfides accounted for ≤ 0.03 wt. % of the total Mn extracted (Fig. 6c, Table 2). From the analyses of the partitioning of reactive Mn species, we can thus confirm that under the anoxic conditions currently prevailing in the bottom waters and anoxic SWI of Lake Medard, a minor, yet still important fraction of reducible $Mn_{HR}$ can be exported from the water column, and can participate, together with the reactive forms of iron, into the internal cycle of S (e.g., Reaction 7).

### 4.6 Insights from siderite, gypsum, and pyrite analyses

### 4.6.1 Siderite

Siderite accounts for up to 3.5 wt. % of the total mineralogy of the anoxic lacustrine sediment where it occurs as dispersed fine crystalline rhombohedra. Siderite displays corroded surfaces towards the SWI. This textural feature cannot be observed in crystals at the 4 to 8 cm depth interval (Fig. 6d-e). This is consistent with results from the sequential iron extraction scheme (see above) indicative of Fe-carbonate likely undergoing recrystallization and/or growth in the deeper part of the examined sediment pile, but partial dissolution towards the SWI, despite its low supersaturation in the monimolimnion ($\Omega_{sid.}$ = log IAP·(log $K_{SP}$)$^{-1}$ = 1.1; Supplement 1).

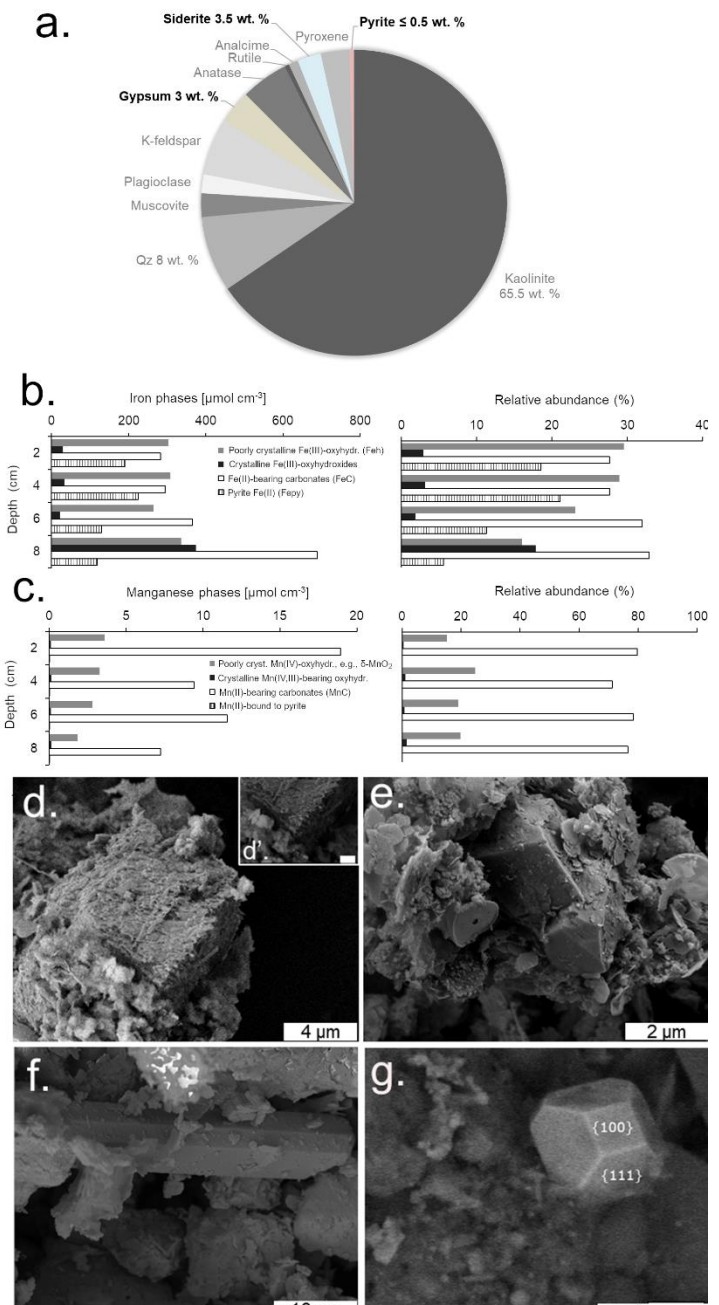

**Figure 6. Representative semi-quantitative mineralogical analysis of the upper sediment (0 to 8 cm depth), the XRD data (a) shows that the sediments are dominated by aluminosilicates and contain gypsum, siderite and pyrite. Results from sequential extraction of iron (b) and manganese (c) portray changes in partitioning of these metals in reactive oxyhydroxide, carbonate and sulfide solid phases with increasing sediment depth. SEM-EDX of rhombohedral siderite in the 0−4 (d) and 4−8 cm sediment depth intervals (e). This carbonate mineral displayed corroded surfaces near the SWI. The texture of microcrystalline equant gypsum (f) and truncated octahedral microcrystalline pyrite (g) are also shown (see text for details).**

The siderite is enriched in $^{13}C$ by around +9 ‰ (mean $\delta^{13}C$ value of siderite is +6.4 ± 0.3 ‰) relative to $\Sigma CO_2$ of the bottom water column (Table 1). The mean $\delta^{13}C$ value of the mineral is, however, within the range of $\delta^{13}C$ isotope values reported by Šmejkal (1978) for carbonates of the Cypris claystone. Also, the mean $\delta^{18}O$ values (+25.7 ± 1.7 ‰) of siderite are within the range observed in Miocene claystone carbonates which, in addition to siderite are comprised also of dolomite and calcite (Šmejkal, 1978, 1984). From combining the average isotopic values and textural features of siderite in our anoxic sediments, the mineral can then be considered a seeded (detrital) phase also sourced from the claystone. Siderite seeds were probably redeposited first in the mine spoils and then in the floor of the post-mining lake, together with aluminosilicates and other major and minor mineral phases, during the lake's flooding stage (2008-2016), or thereafter.

### 4.6.2 Gypsum

Gypsum has a relative abundance of ca. 3 wt. %. It displays a microcrystalline {010}-dominated platy shape (Fig. 6f). This is an equilibrium morphology corresponding to a rather low supersaturation (e.g., Simon et al., 1965; van der Voort and Hartman, 1991; Massaro et al., 2010; Rodríguez-Ruiz et al., 2011). This soluble mineral is not thermodynamically predicted by the aqueous-mineral equilibrium modeling of the monimolimnion water (i.e., $\Omega_{gy}$ = -2.3; Supplement 1). However, a low saturation state ($0 < \Omega_{gy} < 1$) that would allow for

gypsum formation must exist in the upper sediment pore spaces, for instance, where $Ca^{2+}$ ion activities are locally increased by carbonate dissolution.

Gypsum precipitation under low saturation states can probably occur as the result of short-lived, climatically constrained changes in the precipitation-dissolution environment of the upper sediment pile (see Umbria-Salinas et al., 2021). The isotope values of the sulfate moiety in the authigenic gypsum ($\delta^{34}S_{gy}$ and $\delta^{18}O_{gy}$) provide further insight on the significance of this phase within the internal sulfur cycle and early diagenetic context of the system under consideration. The $\delta^{34}S_{gy}$ isotope values ranged from −13.9 and −9.6 ‰. Accordingly, gypsum shows $^{34}S$-depletion of −17.8 to −11.6 ‰ relative to dissolved $SO_4^{2-}$ in the ambient anoxic waters (Table 1). The $\delta^{18}O_{gy}$ values range from +5.1 to +6.3 ‰ (V-SMOW). In consequence, the sulfate in gypsum is $^{18}O$ enriched by +1.4 to +2.6 ‰ as compared with the mean $\delta^{18}O_{SO4}$ of the monimolimnion (Table 1). This magnitude of isotope $^{18}O$ enrichment of gypsum-sulfate appears consistent with the range observed when sulfate is derived from pyrite that is oxidized by ferric iron in aqueous anaerobic experiments (e.g., Taylor et al., 1984b; Toran and Harris, 1989; Balci et al., 2007).

A net $O_2$ neutral reaction that also accounts for (i) significant iron sulfide oxidation, (ii) the localized presence of corroded siderite in the upper sediment, (iii) involves chemolithoautotrophic fixation of $CO_2$, and (iv) produces an isotopically light gypsum-sulfate could therefore be written (Reaction 8):

$$3Fe^{32}S + 3CaCO_3 + FeCO_3 + 14H_2O \rightarrow 4FeOOH + 3Ca^{32}SO_4 \cdot 2H_2O + 4CH_2O \tag{8}$$

Reaction 7 assumes that the acidity produced by the oxidation of pyrite and its precursors is neutralized by a 3:1 dissolution of calcium to iron carbonate phases in the upper anoxic sediments. The $Ca^{2+}$ ions released by carbonate dissolution can then co-precipitate with the porewater $SO_4^{2-}$ ions to form gypsum. The mineral is $^{34}S$-depleted compared to sulfate dissolved in the monimolimnion, but it reflects the $\delta^{18}O$ signature of the ambient anoxic water.

### 4.6.3 Pyrite

Pyrite accounted for $\leq 0.5$ wt. % of the total XRD-estimated mineralogy of the sediments and occurs as finely dispersed single octahedral crystals that are up to 2 µm in size, and exhibit {111} and {100} truncations (Fig. 6g). This morphology is often seen to develop under sulfide-limited conditions in synthetic experiments (e.g., Barnard and Russo, 2009). From the morphology of pyrite and because its $\delta^{34}S$ isotope values differ considerably from those of weathered pyrite in the coal seams-associated lithology (Bouška *et al.*, 1997; Appendix B: Fig. B2), this mineral is more probably authigenic in origin. It must have formed locally within the anoxic sediments at low supersaturation, and with nucleation itself depleting the availability of reactants (i.e., $S^{2-}$ species) required for further nuclei formation (Rickard and Morse, 2005). Hence, its dispersed, fine crystalline occurrence.

The $\delta^{34}S$ isotope values of the finely dispersed pyrite crystals are operationally defined as those of the bulk sediment chromium-reducible sulfide (CRS) pool (Canfield et al., 1986). In the upper anoxic sediments, this CRS pool became $^{34}S$-enriched with

depth. Accordingly, in the 0 to 4 cm-depth pyrite has $\delta^{34}S_{CRS}$ isotope values of $-34.7 \pm 0.4$ ‰. At 4 to 8 cm sediment depth, however, it is relatively $^{34}S$-enriched ($\delta^{34}S_{CRS} = -23.9 \pm 0.9$ ‰).

Pyrite captures the isotopic signature of dissolved sulfide in its local precipitation environment, and near the SWI this mineral appears to have recorded an isotopic offset ($^{34}\varepsilon_{CRS-SO4}$) of around 38 ‰ relative to the $\delta^{34}S_{SO4}$ of the monimolimnion. This magnitude of apparent fractionation could be ascribed to incomplete microbial sulfate reduction, within open system oxidative sulfur cycling (Johnston et al., 2005, Zerkle et al., 2016). It may well point to our biogenic pyrite resulting from the activity of bacteria capable of fully oxidizing the organic substrates scarcely available (Canfield, 2001; Brüchert, 2004), which could explain the observed depletion of lactate and pyruvate in the bottom water column. Limited microbial sulfate reduction is consistent with the fact that pyrite in the modern lacustrine sediments precipitates without triggering sulfate nor divalent iron exhaustion (Scholz, 2018; Canfield, 2001).

Approximately 10 ‰ $^{34}S$ isotope enrichment in authigenic pyrite at the bottom of our section hints to an additional heavy CRS formation mechanism being more active deeper within the anoxic sediment pile. It could also be the case that the $\delta^{34}S_{SO4}$ values in porewaters in equilibrium with the heavier pyrite are evolved because of variable fractionations associated with MSR (Canfield, 2001; Brüchert, 2004). The $\delta^{34}S$ values of pyrite from the lower part of the cores also exhibit a narrower difference when compared with those coexisting with authigenic gypsum as shown in Fig. B2 (Appendix B). We can attribute these results to a greater abundance of highly reactive Feh phases capable of oxidizing monosulfide (Table 2) in the lower part of the cores investigated.

The CRS pool also includes the sediment's $S^0$ fraction (Canfield et al., 1986), and given that $S^0$ derived from the chemolithotrophic oxidation of sulfide is relatively $^{34}S$-enriched (e.g., Zerkle et al., 2016; Pellerin et al., 2019), we suggest that $^{34}S$ enrichment in gypsum in the bottom sediments fingerprints isotopically heavier $S^0$ comprising an evolved CRS pool. This interpretation is not only consistent with the decreased proportions of $Fe_{py}$ in the lower part of the sediment pile (Table 2), but also with microbial disproportionation-induced fractionations (e.g., Canfield, 2001, 2003; Böttcher et al., 2005; Pellerin et al., 2019).

## 4.7 The imbalanced aqueous redox system in Lake Medard: synthesis

The newly formed Lake Medard has overlapping S, N, Fe, and C cycles occurring in the anoxic portion of the water column. This is unusual in natural, redox stabilized meromictic lakes where at least one of these cycles is functionally diminished or undergoes minimal redox transformations. Alternation of two bistable states could be the case in natural aqueous systems that can be rendered ferruginous, and this alternation is largely controlled by shifts in the prevailing trophic state. Accordingly, ferruginous conditions occur in low productivity, organic-poor systems; whilst euxinic conditions would dominate in high productivity, organic-rich systems where production of sulfide depletes dissolved sulfate and may out titrate dissolved iron (van de Velde et al., 2021; Antler et al., 2019).

The redox stratified Lake Medard demonstrates that ferruginous conditions can develop without substantial sulfate consumption (see Scholz, 2018, and references therein). Our geochemical model on this imbalanced redox system confers a major role to a planktonic prokaryote community that is, to some extent, compartmentalized in the bottom water column, where it mediates in the interlinked C, N, S, and Fe and Mn species transformations occurring across the redoxcline (Fig. 7). These transformations involve a cryptic sulfur cycle with generation and consumption of sulfur intermediates and exert an influence on the concentration gradients of other dissolved bioactive species, e.g., phosphate. The internal P cycling occurring below the redoxcline (Fig. 7) can in fact render the entire water column oligotrophic (Petrash et al., 2018).

Towards the hypolimnion, particulate matter formation involves a microaerophilic iron oxidizers/ nitrate reducer community (e.g., Gallionellaceae). These members of the community could be hypothesized as promoters of a continuous amorphous iron aggregate precipitation and export down to the ferruginous SWI, where these aggregates stabilize and/or are reductively dissolved by iron reducers (e.g., *Geobacter* spp.). In the sediment, stocks of pre-existing siderite, and recently stabilized oxyhydroxides fuel anaerobic oxidation and disproportionation of by-product sulfide from MSR. In consequence, the coupled stable oxygen and sulfur isotope-based SRR estimate indicates no net sulfate reduction, despite an increased genetic potential for this pathway, as deducted from analysis, and concomitant evidence for dissolved $SO_4^{2-}$ consumption likely involving metastable FeS formation in the monimolimnion. We are furthering the study of the interplay between Fe and S cycles in the $O_2$-depleted water column by bridging our $\delta^{34}S$ data with $\delta^{56}Fe$ measurements. The combined results support an active vigorous co-recycling of these elements below the redoxcline (Petrash et al., 2022). Accordingly, an increase in the relative proportion of dissolved $^{56}Fe$ near the lakebed ($\delta^{56}Fe = +0.12 \pm 0.05$ ‰) can be ascribed to precipitation of monosulfides, whilst precipitation of oxyhydroxides at the redoxcline leads to depletion of $^{56}Fe$ ($\delta^{56}Fe = -1.77 \pm 0.03$) in the residual dissolved Fe(II) (cf. Busigny et al., 2014).

The $\delta^{34}S_{CRS}$ values in the upper part of the sediment pile were consistent with incipient and incomplete MSR-induced fractionation, yet MSR is not accompanied by dissolved sulfate depletion because of low organic substrate availability and due to bioenergetic considerations given by the presence of dissimilatory iron reducers and an abundance of Fe(III) substrates. Importantly, the $\delta^{34}S_{CRS}$ of the CRS pool at the lower sediment pile likely incorporates $^{34}S$ from intermediate sulfur. Finally, acidity generated by anaerobic S oxidation reactions proceeding near and at the SWI is neutralized by partial carbonate dissolution, which in turn provides $Ca^{2+}$ ions for interstitial microcrystalline gypsum precipitation. This gypsum's $\delta^{34}S$ values fingerprint intermediate sulfur disproportionation. Redeposited siderite, although experiencing dissolution at the SWI, may be undergoing recrystallization and growth below ~4 cm sediment−depth, such as evidenced by increase FeC contents and the absence of corroded siderite crystal surfaces in the lower part of the sedimentary section examined here.

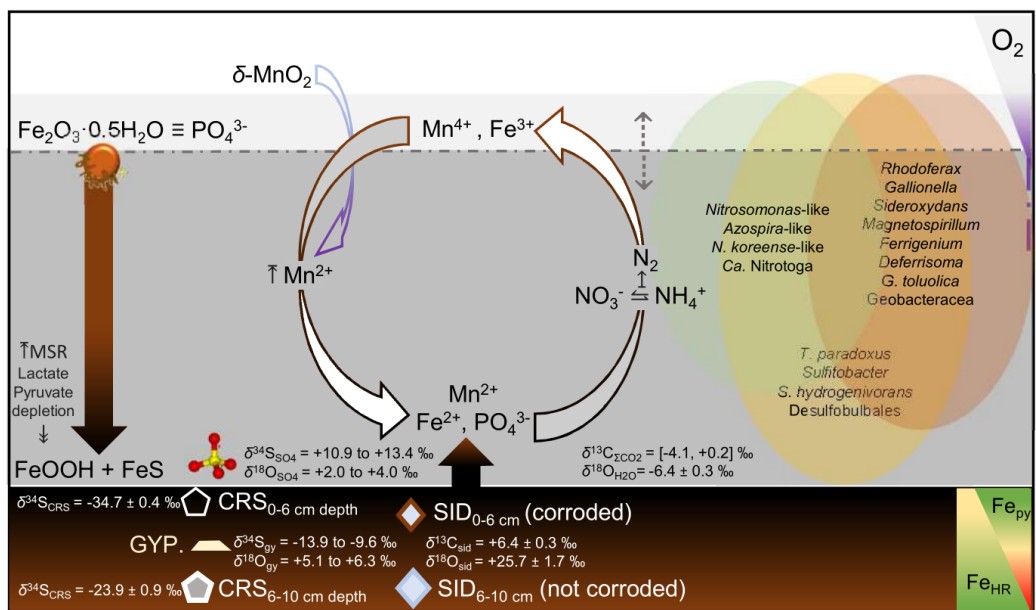

**Figure 7. Scheme summarizing the speciation and stable isotopes ranges of sulfur-bearing phases (pyrite, $S^0$: CRS; gypsum: GYP) and siderite: SID) and the biogeochemical cycling mechanisms likely operating in the redox stratified Lake Medard and its SWI. (Background colours as in Fig. 2) The prokaryote groups depicted represent nitrate-, iron- and sulfur-utilizing species identified via 16S rRNA gene amplicon sequencing (see text for details).**

## 4.8 Relevance for deep time palaeoceanographic and/or diagenetic interpretations

The current lake system provides the opportunity to investigate biogeochemical controls active under a transitional state between nitrogenous and sulfidic conditions. This state cannot be observed in the scarce examples of redox stratified euxinic marine basins existing today (i.e., Black Sea, Cariaco Basin; Meyer and Kump, 2008) nor in the few natural mesotrophic to eutrophic ferruginous lakes presumedly analogues to ancient redox stratified oceans (see Koeksoy et al., 2015). Similar transitional redox states would have been more prevalent at times with decreased Phanerozoic seawater sulfate concentrations and diminished shuttling of Fe(II) to sediments. Together these factors would have enabled more widespread ferruginous conditions (Reershemius and Planavsky, 2021) that transiently encompassed the water column of Mesozoic epicontinental seas (Petrash et al., 2016; Bauer et al., 2022). Therefore, the link between the biogeochemical controls operating in the water column of our study site and the mineral equilibrium conditions prevailing near and at its anoxic SWI may be relevant for studying elusive shallow burial diagenetic signals developed in fluid-buffered sediments. Also, to unravel overprinting of redox proxies in carbonates altered in movable redox stratified coastal aquifers (Petrash et al., 2021).

In deeper geological time, the increased delivery of continental sulfate to Precambrian sediments containing not only iron oxyhydroxides but also siderite, which probably triggered early diagenetic reactions similar to those reported here (e.g., Bachan and Kump, 2015). Comparable diagenetic hydrochemical conditions would have arisen also when transgressions of basinal ferruginous seawater affected evaporitic facies buried by coastal progradation. In this scenario, the low preservation potential

of gypsum would have hindered direct interpretations of any possible isotopic offset recorded by its more stable replacive phases (e.g., silicified Fe-dolomite).

Although gypsum is rarely preserved in Proterozoic shallow-marine successions (but see Blättler et al., 2018), pseudomorphic carbonates after this mineral are volumetrically important in many Precambrian peritidal facies. In such facies, primary gypsum was often replaced by a metastable early diagenetic phase (e.g., Philippot et al., 2009). In a modern thrombolite-forming environment, Petrash et al. (2012) describes an early replacement process of gypsum that involves initial replacement by metastable aragonite. This produces Sr carbonate signals in pseudomorphic calcite replacing aragonite that depart from the Sr content of the ambient water, and, by analogy, can disguise an ancient primary gypsum mineralogy. Similarly to Sr, the structurally substituted sulfate in the carbonate lattice (CAS) of peritidal carbonates (i.e., as a putative proxy for contemporary Proterozoic seawater sulfate) can also be altered early during diagenesis, and now exhibit isotope signals incompatible with those of coexisting pyrite (Blättler et al., 2020). The $\delta^{34}S$ values of these phases—if formed contemporaneously—would be expected to be similar as per the low dissolved sulfate levels generally ascribed to Proterozoic open oceans (e.g., < 400 μM, Fakhraee et al., 2019). An explanation for such a discrepancy is that the CAS and pyrite S isotope proxies recorded the pore fluid signal of diagenetically evolved sulfate in Precambrian (e.g., Rennie and Turchyn, 2014; Li et al., 2015), and some Phanerozoic evaporitic/ stromatolitic facies (e.g., Thomazo et al., 2019). Conversely, a similar inconsistency could arise when early diagenesis ensued transient out-of-equilibrium water column conditions equivalent to those currently prevailing in Lake Medard, i.e.: (i) dissolved $Fe^{2+}$ is amongst the dominant redox species; (ii) substantial dissolved and solid phase sulfate are present; (iii) the oxidized $Fe_{HR}$ sediment stocks buffer dissolved sulfide accumulation; and (iv) dissolution of redeposited carbonates buffers the system with regard to acidity generated by anoxygenic oxidative reactions.

## 5. Conclusions

We investigated biomineralization reactions occurring, and prokaryotes thriving in the ferruginous and sulfate-rich water column of a post-mining lake. For this purpose, we considered the pools and fluxes of iron, manganese, carbon, nitrogen, and sulfur in the bottom redox stratified water column and upper reactive sediments (Fig. 7). Discrete spectroscopic datasets were combined with a 16S rRNA gene-aided inference of the planktonic prokaryote community structure to unravel the mechanisms procuring and/or consuming bioactive nitrogen, iron, and sulfur species in the redox stratified ecosystem. Integration of these datasets provides evidence for niche differentiation, but despite marked redox gradients in the water column, we observed a sustained genetic potential for anoxygenic sulfide oxidation and intermediate sulfur disproportionation. The processes were further substantiated by using sulfate S and O isotope systematics. Microbe-mineral interactions near the anoxic sediment− water interface modulate the aqueous equilibrium of both reactive authigenic and redeposited Fe- and Mn-bearing phases. A vigorous anoxic sulfide oxidation pathway is coupled to the reduction and solubilization of the ferric and manganic particulate stocks of the lacustrine sediment (Fig. 7).

Dissolved sulfate need not to be quantitatively depleted for the establishment of ferruginous conditions in the water column. The aqueous system-scale reactions currently proceeding in the redox stratified water column and upper anoxic sediments of Lake Medard are relevant for describing transient redox imbalanced stages between nitrogenous and ferruginous conditions that developed in low productivity water columns of ancient nearshore marine settings featuring decreased but not exhausted sulfate levels. These could have produced some of the conflicting isotope signatures often described for coexisting phases of

interest as paleoredox proxies, e.g., carbonates and sulfides. The effects in the geochemical record of analogue imbalanced states are yet to be fully accounted for. This research effort has implications for untangling the deep time palaeoceanographic redox structure of continental margins. We anticipate that further studies in the ferruginous artificial lacustrine system targeted here can provide a more complete picture depicting processes recorded by conflicting proxies in several key, well-preserved Precambrian and Phanerozoic shallow-marine facies.

**Appendix A. Geological background**

The northwest Bohemia region (Czechia) was an intracontinental basin comprised of peatlands, isolated ephemeral lakes and peat bogs by the Late Eocene. This lowland landscape developed and expanded in association with subsidence in the Eger rift (Dèzes et al., 2004). By the Oligocene, the lowlands extended over an area >1,000 km$^2$ along the Sokolov and Most basins (Matys Grygar et al., 2014). Thus, organic-rich peatlands now encompass lignite seams that correlate across the Czech-

Germany boundary and towards Polish Silesia. The extended wetlands along the Eger continental rift turned, by the beginning of the Miocene, into a large playa lake affected by exhalative hydrothermal inputs (Pačes and Šmejkal, 2004) and episodically by alkaline volcanism (Ulrych et al., 2011). The paleolake deposits recorded the last interval of the syn-rift sedimentation and consist of 70–120 m thick carbonate-rich, kaolinitic coal-bearing claystone with several horizons of tuff material. These deposits are lithostratigraphically referred to as the Cypris Formation (Kříbek et al., 1998, 2017), and now outcrop in elevated

areas of the Sokolov Mining District, where they overlie the coal seams that were exploited to exhaustion in the former Medard open-cast mine. Percolation of waters from the Miocene paleolake produced epithermal mineral salt deposits. Efflorescences of thernadite ($Na_2SO_4$) are associated with fluid flow along faults and fractures (Šmejkal, 1978). Modern hydrological processes, including groundwater infiltration (Rapantova et al., 2012, Kovar et al., 2016), introduce dissolved sulfate (and iron) into the modern hydrological system (Pačes and Šmejkal, 2004). A 3-year monitoring survey (2007-2010) of dissolved

sulfate and iron concentrations in the watershed now occupied by the post-mining lake (Supplement 3: hydrochemical contours) explains the spatial (and temporal) concentration variabilities seen as bottom waters concentrations of these ions were measured and compared across the lake's central W-E axis. For example, western Medard has consistently higher Fe(II) contents matching the dissolved iron gradients observed in the watershed. Conversely, dissolved $SO_4^{2+}$ increases towards the east (Petrash et al., 2018; Supplement 3).

The Miocene Cypris claystone and quaternary alluvions comprised of material derived from this unit, function as the main source of sediments to the modern post-mining lacustrine system. The mineral assemblage of the stratigraphic unit includes

kaolinite, K-feldspar, quartz, rutile and anatase, and gypsum. It also contains analcime ($NaAlSi_2O_6$), weathered pyrite, carbonates (calcite, Fe-dolomite, and siderite), and greigite ($Fe_3S_4$) (Murad and Rojík, 2003, 2005).

Organic matter content in the Cypris claystone exhibits variability that recorded discontinuous development of widespread

anoxia across the paleolake, accompanied also by shifts in salinity and alkalinity. This paleoenvironmental setting promoted lacustrine authigenic carbonate deposition (Kříbek et al., 2017). Overall, the authigenic mineral assemblage, elemental concentration trends, and the heavy O and S isotopic signatures of secondary sulfate minerals of the Cypris claystone (Fig. B2, Appendix B) indicate precipitation in a large saline playa paleolake in which the oxidative weathering of sulfides, volcanic exhalations, and meteoric water-rock interactions imparted a major geochemical imprint that is superimposed on that of the

episodic changes in the paleolake's redox conditions (Šmejkal, 1978; Pačes & Šmejkal, 2004). A compilation of the $\delta^{34}S$ of the sulfate sourced largely from the Miocene claystone is shown in Figure B2 (Appendix B). As discussed in the main text, dissolved sulfate of the modern redox stratified Lake Medard's waters fingerprint these sources.

## Appendix B.

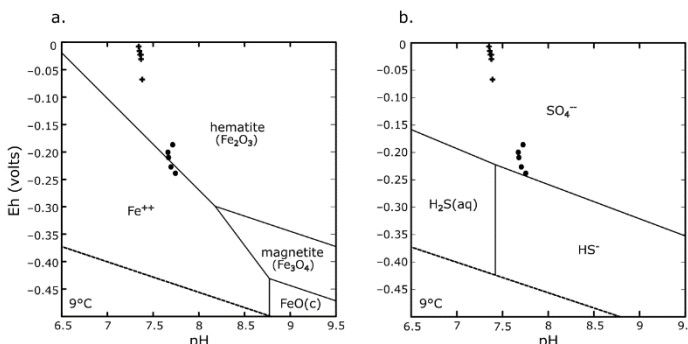

**Figure B1. Pourbaix diagrams of the thermodynamically stable Fe and S phases in the bottom waters of Lake Medard at the time of sampling (dots). Also shown are modeling results of Eh-pH parameters measured when the redoxcline shifted downwards and mean monimolimnion Eh transiently changed from < −200 mV to −80 mV (crosses). Variation of these physicochemical parameters coincide with seasonal hydrological dynamics of the local watershed, and its effects over groundwater influx. Seasonal, short-lived shifts of conditions at the monimolimnic ferruginous waters favour Fe(III)-oxyhydroxide precipitation.**

**Figure B2. A comparison of the ranges of reported $\delta^{34}S$ values of potential sources of oxidized sulfur to Lake**
**Medard (after Šmejkal 1978; Krs et al., 1990 for greigite), the ranges of sulfate-rich bottom water column and authigenic gypsum and pyrite in the upper anoxic sediments (this work) are also shown (filled boxes).**

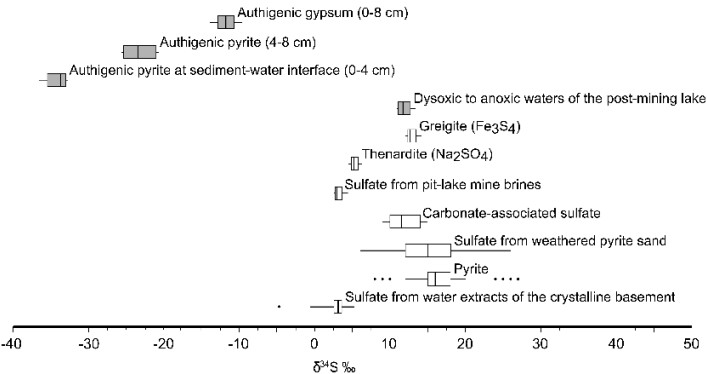


**Author contribution:** DAP: Funding acquisition, conceptualization, methodology, data curation, formal analyses, investigation, resources, writing—original draft preparation and editions. IMS: methodology, data curation, formal analyses, original draft preparation. AV: methodology, data curation, formal analyses, original draft preparation. TBM: resources, formal

analysis, validation, writing—original draft preparation. TP: validation, writing—original draft preparation CT: resources, validation, formal analysis, validation. writing—original draft preparation.

**Competing interests:** The authors declare that they have no conflict of interest.

**Acknowledgements**

This study was funded by the Czech Science Foundation (Junior Grant 19-15096Y to DAP). We are grateful to Alexandra A.
Philips and two anonymous reviewers for constructive criticisms and suggestions that improved an early (preprint) version of this manuscript. The editorial input of Denise Akob is also thanked. We sincerely thank Jiří Jan and Jakub Borovec (BC-CAS) for technical support while sampling the bottom waters and anoxic sediments of Lake Medard and for support during analyses. We are thankful to K. Umbria-Salinas for wet lab assistance. S.V. Lalonde (European Institute for Marine Studies, Brest) is thanked for HR-ICP-MS of the water samples and V. Chrastný (Czech University of Life Sciences, Prague) for ICP-MS of
iron and manganese in the sediment reactive fractions.

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
