# Peer review of "Aqueous system-level processes and prokaryote assemblages in the ferruginous and sulfate-rich bottom waters of a post-mining lake"

_Biogeosciences, 2021_

## Referee Comment (RC2)

**Aqueous system-level processes and prokaryote assemblages in the ferruginous and sulfate-rich bottom waters of a post-mining lake**

Daniel A. Petrash1,2, Ingrid M. Steenbergen1,3, Astolfo Valero1,3, Travis B. Meador1,3, Tomáš Pačes2, Christophe Thomazo4,5

1Biology Centre of the Czech Academy of Sciences, České Budějovice, 370 05, Czechia
 2Czech Geological Survey, Prague, 152 00, Czechia
 3University of South Bohemia, České Budějovice, 370 05, Czechia
 4University of Burgundy, Dijon, 21000, France
 5Institut Universitaire de France, Paris, 75000, France

10 Correspondence to: Daniel A. Petrash (petrash@ualberta.ca)

[revised manuscript text omitted]

---

## Author Comment (AC1)

**Reviewer 1, Comment 1 (RC1.1).** This is an interesting system in that it has overlapping S, N, Fe, and C cycles all in the anoxic portion of the water column. This is unusual, b/c in many systems at least one of these is a functionally absent, a minor component, or undergoes minimal redox transformations because of the dominance of (an)other component(s).

**DP1.1:** The reviewer has kindly provided us with a relevant synthesis of the importance of our observations in this lake system, and it has been incorporated in the abstract and as introductory statement to a new section 4.7, titled "The unbalanced aqueous redox system in Lake Medard: synthesis". This introductory paragraph reads:

"The newly formed Lake Medard has overlapping S, N, Fe, and C cycles occurring in the anoxic portion of the water column. This is unusual in natural redox stabilized meromictic lakes where at least one of these cycles is functionally diminished or undergoes minimal redox transformations because of the dominance of (an)other component(s). Such stable condition favour alternation of two bistable states that are driven by feedback reactions in turn determined by the organic carbon content of the system. Accordingly, ferruginous conditions shall occur in low productivity, organic poor systems; whilst sulfide-rich, methanogenic conditions dominate in high productivity systems (Antler et al., 2019; van de Velde et al., 2021). […]"

**RC1.2.** The paper is mostly observational, but I think that's fine, because the authors have developed a nice model for the processes occurring that's shown in Figure 8. The paper presents a lot of results. I almost wish the authors could tie everything together a little more succinctly.

**DP1.2.** We have streamlined few sections of the manuscript while keeping our aim, which is to provide a complete account of current biogeochemical conditions in a transitional redox system. We have in all possible instances shortened and synthesized our results. But, please note that we had to include the Methods, as per reviewer 3 suggestion, in the main text. This adds to the overall length of the revised MS.

**RC1.3.** I think Figure 8 gives a nice summary of the processes involved. Maybe a little more time in the discussion focused on this model and a little less on the "paleo" implications would help the reader synthesize all of the observations into the process model.

**DP1.3.** As per reviewer suggestion we now dedicated a section (4.7, see DP1. Above) to better explain our biogeochemical model, also providing a synthesized view of the system based and our observations and interpretations. Reviewer 3, however, commented (RC3.25) on the "paleo" implications being quite enjoyable a section. So, we have chosen to keep it as it is (preprint version).

**RC1. 4.** Below, I provide some specific comments and suggestions that I hope can help improve the manuscript.

**DP1.4.** We sincerely thank the reviewer for her/his attention to detail while reviewing our draft/preprint; the comments and suggestion kindly provided were addressed as described below.

**RC1. 5.** Ln. 11. Do the authors mean "reductive Fe(III) dissolution"?

**DP1.5.** Yes, missing "Ï" is now added (Ln. 11)

**RC1. 6.** Ln. 16. "sustained" how?

**DP1.6.** Here we referred to a continuous genetic potential for anoxic sulfide oxidation. For clarity, the wording in the abstract has been modified as follow (Lns. 21-3): "[…] Yet, the planktonic microbial succession across the nitrogenous and ferruginous zones also indicates genetic potential for chemolithotrophic sulfur oxidation in the anoxic portion of the bottom water column […]".

**RC1. 7.** Ln. 17-18. What is the electron acceptor for sulfide oxidation? And if sulfide is oxidized all the way to sulfate, at which point does the sulfur disproportionation happen?

**DP1.7.** We considered that this information was adequately developed (within the length-limitations of the abstract section) in Lns. 19-21 (now 25-26): "Near and at the anoxic sediment-water interface, vigorous sulfur cycling, can be fuelled by ferric and manganic particulate matter and redeposited siderite stocks." There are no further changes regarding this reviewer's comment.

**RC1. 8.** Ln. 104. Change to "DNA extraction and MiSeq"

**DP1.8.** This has changed as per the Methods section now being part of the main text. Yet, please note that Ln. 120 now reads: "[…] environmental DNA (eDNA) extraction followed by MiSeq Illumina 16S rRNA gene amplicon sequencing […]"

**RC1. 9.** Ln. 105. Do you mean ICM-MS? Does mass spec work for specific ions? I don't think this is an adequate description of the analytical methods.

**DP1.9.** We thank the reviewer for noticing an omission/ and error in our description of analytical methods. Ion concentrations were determined using HP-LC. The following lines (Lns. 121-22) were added to address this flaw kindly noticed by the reviewer: "[…] (ii) mass determinations of cations (iron, manganese, potassium, sodium, magnesium and calcium); (iii) high pressure liquid chromatography for concentrations of chlorine, sulfate, nitrate, ammonium and phosphate anions, and VFA abundances; […]". Further details on how the measurements were carried out are in Sect. 3.1.5, subtitled "Ions, ammonia, and VFAs concentration analyses".

**RC1. 10.** I'm also al little bit leery of putting the entire description of nucleic acid-based microbial community analysis in the supplement.

**DP1.10.** The methods, including the specific set of analyses indicated by the reviewer, are now part of the main text.

**RC1. 11.** Ln. 106. Change to "measurement of dissolved"

**DP1.11.** Done as per reviewer suggestion (Ln. 124).

**RC1. 12. Figure 2[b]:** Can the authors make the axes the same on panels above and below the redoxcline? My first impression was that there was no difference between any of the organic acids, but there are actually rather dramatic differences.

**DP1.12.** Fig. 2b has been modified as per reviewer suggestion to better highlight the change in VFAs concentration recorded above and below the redox aqueous interface.

**RC1. 13.** Ln. 150. VFAs were a minor fraction of the total DOC. What is likely the rest? How labile might it be, and how does that inform the biogeochemical model?

**DP1.13.** The main components supporting microbial growth are simple mono- and oligomers that are only present in nM concentrations. We determined VFAs concentrations as they act as electron donor and or C source to heterotrophic microorganism that could influence the observed hydrochemical processes. As per this request of the reviewer, the composition of the DOC pool is inferred as follow (Lns. 357-60): "[…] DOC is generally comprised of relatively high molecular weight organic compounds (not quantified here), such as cellular exudates from alive and senescent planktonic microorganisms (e.g., algae, protists, bacteria) and their degradation products. Probably present in solution were also soluble humic substances derived from the biological breakdown of refractory organic matter (i.e., lignite particles) in the sediment (Petrash et al., 2018). […]"

**RC1. 14.** Ln. 167-168. Wouldn't sulfate reduction induce increase in pH? You're producing carbonate alkalinity and reducing a strong acid (sulfate) to a weak acid (sulfide).

**DP1.14.** Usually it is the expected effect. But, MSR has been experimentally shown to decrease pH when lactate is the electron donor. Our annotation speculates on such an effect potentially occurring here as exhaustion of lactate is linked to the increased number of OTUs functionally associated with $SO_4^{2-}$reduction in the monimolimnion. The following line address

further the speculative assertion (Lns. 378-79): "Therefore, the complete (oxidation to CO2) and incomplete (to acetate) oxidation of lactate by MSR could be a factor contributing to the slight decrease in pH in the monimolimnion (see Gallagher et al., 2012). […]"

Gallagher K.L. Kading T. Braissant O. Dupraz C. Visscher P.T.: Inside the alkalinity engine: The role of electron donors in the organomineralization potential of sulfate-reducing bacteria: Geobiology 10, 518–530, 2012.

**RC1. 15.** Ln. 221. Please be consistent in including the charge for nitrate

**DP1.15.** Charge of nitrate is now constantly stated in all instances where it appears on the text.

**RC1. 16.** Ln. 223. Please change "sequenced" to "detected"

**DP1.16.** Done.

**RC1. 17.** Ln. 232. By "abundance peak" do you mean maximal relative abundance?

**DP1.17.** Yes, as in revised Ln. 456: "The maximal relative abundance of an *Azospira*-like microorganism (95 % similarity) coincides with the peak of relative maximal abundance of members of the Gallionellacea family at 49 to 50 m depth (Fig. 5a, Supplement 1). […]"

**RC1. 18.** Ln. 210-256. Did the authors try to quantify nitrite? If there are nitrogen transformations occurring in this system, I would expect it to be important, and perhaps the ultimate oxidant for $Fe^{2+}$ in reactions 1 and 2.

**DP1.17.** The reviewer is right, nitrite could be expected to increase concentrations towards the anoxic part of the water column and exert an important control in reactions leading to Fe oxidation. Nitrite role, however, remains to be further tested experimentally or by specialized sampling/analytical protocols that can resolve reactive N availability and transformations occurring in the natural lab under examination. In our case, increasing Cl- concentrations with increasing depth hindered an accurate profiling of nitrite.

The revised text now informs the reader what anions were considered (please see DP1.9).Also, the following text was added as preamble to presenting reactions 1 to 3: "These cycles in the aqueous system under consideration are likely interlinked throughout microbial mediation in the generalized Reactions (1-3), but nitrite may as well be a relevant oxidant : […]"

**RC1. 19.** Ln. 282. Are the authors referring to Fe(II) oxidation by Mn(III/IV)? Please clarify.

**DP1.19.** Yes. As per reaction 4, iron is oxidized and Mn is reduced. It can also be seen as Fe(II) as reductant of Mn(III,IV). To clarify on this note, we modified the text to read "[…] Divalent iron […]" (Ln. 497).

**RC1. 20.** Ln. 293. Please change "[Fe]" to "Fe concentration"; also here and throughout, please check tense agreement.

**DP1.20.** Changed as per reviewer suggestions. Sentence tense agreement also revised thoroughly though the text.

**RC1. 21.** Ln. 301. I don't know about this. Attributing metabolism when you only have 91% similarity is tough.

**DP1.21.** True. The offending sentence is now removed.

**RC1. 22.** Ln. 329. Please remove "significantly"

**DP1.22.** Done.

**RC1. 23.** Ln. 336 and throughout this section. Why does diversity matter. Wouldn't relative abundance be more informative with respect to S transformations? There could be a whole lot of diversity of sulfate reducers, but they're only a minor fraction of the community.

**DP1.23.** We have clarified what we referred here as diversity as follow (Ln. 555): "[…] low number of taxonomic groups, and Ln. 576: "[…] a more diverse sulfur-respiring bacterial

population (Fig. 5c). This was dominated by many relatively rare taxa and a few abundant lineages […]"

The issues of considering relative abundance alone as a control of relevant hydrochemical transformations include potential biases induced by sampling processing. The information provided can therefore only inform us on changes in conditions that allow, for instance rare taxa to better thrive, or additional MSRs to be detected with the protocols implemented. We have added the following text to add additional context to this matter (Ln. 59-61): "Despite limitations linked to quantitative biases introduced, for example, during sample DNA extraction, PCR amplification, and uneven coverage of universal primers across phylogenetic groups, the sequencing of amplified fragments of prokaryote rRNA genes can provide insights useful for ecological deductions (see Piwosz et al., 2020) .[…]"

Piwosz, K., Shabarova, T., Pernthaler, J., Posch, T., Šimek, K., Porcal, P., and Salcher, M. M.: Bacterial and Eukaryotic Small-Subunit Amplicon Data Do Not Provide a Quantitative Picture of Microbial Communities, but They Are Reliable in the Context of Ecological Interpretations, 5, 2020.

**RC1. 24.** A later use of the term "diversity" leads me to believe the authors are referring to diversity of S metabolisms (e.g., oxidation, disproportionation, reduction of different redox states, etc.), but I'm not sure. Please revisit the use of this term and clarify.

**DP1.24.** The use of the term has been revisited and clarified as indicated above.

**RC1. 25. Ln. 372-374:** If there's evidence of S metabolizing organisms and some aqueous chemical evidence of S transformations, why no change in del34S-sulfate?

**DP1.25.** This question of the reviewer is answered in Lns. 615-19 of the revised text: "The intracellular isotope exchange of sulfite with anoxic ambient waters has been proven to produce an oxidized $SO_4^{2-}$ product that is enriched in $^{18}O$ relative to precursory thiosulfate and/or sulfite. This enrichment displays only a minor change, if any, in its corresponding S isotope composition (e.g., Böttcher et al., 2005; Johnston et al., 2014; Bertran et al., 2020; see Table 1)." (Lns. 398-401 of the preprint). But please note that the following lines were added as a closing statement of the section 4.4.1 (Lns. 619-24) to account for an increase observed at depth 48: "In line with this assertion, at the monimolimnion we observed in dissolved sulfate a negligible sulfur isotope fractionation accompanying the recorded fractionation of oxygen isotope. Yet, we registered a small, but significant reverse sulfur isotope effect (+2.2 ‰) at the upper hypolimnion (Fig. 6a: 48 m depth). This isotope effect could be ascribed either to abiotic or biotic oxidation processes of intermediate S species occurring at that level of the water column (see Zerkle et al., 2016, their table 1). […]"

**RC1. 26.** Reactions 4-6. Is Mn-dependent S disproportionation from the Böttcher et al. 2001 paper? What about the siderite-dependent disproportionation? I am unsure how this reaction might occur.

**DP1.26.** Yes. $FeCO_3$: i.e., dissolution of siderite by excess $H^+$ increases the availability of Fe(II) that scavenges by-product $H_2S$ to sustain disproportioning bacterial growth (Thamdrup et al., 1993). The missing relevant references describing these disproportionation mechanisms were added (Ln. 613) to provide further context (i.e., half reactions).

**RC1. 27.** Ln. 520, 534, 559. Why are these minerals italicized?

**DP1.27.** These are now subsections comprising section 4.6.

**DP1.28 [Final Statement, acknowledge]:** We sincerely thank the reviewer for a throughfall revision of our preprint that has contributed to its improvement aimed at final publication.

---

## Author Comment (AC2)

**Reviewer 2**

**Reviewer 2, Comment 1(RC2.1). [Article]:** is well-written and summarizes all principal aspect of the pit lake, as well as the importance of the study and how in general was conducted. There were few specific comments that I would the authors to take into consideration.

**DP2.1**. We sincerely thank the reviewer for her valuable time, the many suggestions generously given that—together with her constructive criticism—have significantly improved the MS for final publication.

**RC2.2. [Methods section]:** is nicely organized in subsections as supplementary information (S1), but there were specific details that would enhance the reproducibility of the applied methods.

**DP2.2.** Missing details that were missing are now added. For example, on sub-sampling tasks; HP-LC, etc. Also, please note that the methods section is now part of the main text as per the R3 request: RC3.11.

**RC2.3. [Results and Discussion]:** the authors did a good job discussing all the results and its significance. This section requires more work specially with respect to enhancing the clarity of the figures and their description in the text.

**DP2.3.** The figures have been further processed for clarity, for example we added in all profiles the hypo and monimolimnion, etc, we improve colour palettes used for genomics figure, revised captions, etc.

**RC2.4a.** It was not clear to me when was the lake flooded, 2005 according to figure 1a? If so, please state this in line 43 where it is written: "This newly formed,…". Do you have supporting info of when did the lake become meromictic, or how long has it been meromictic?

**DP2.4a.** Information and a reference to relevant published work that fill this gap kindly highlighted by the reviewer, have been added (Lns. 88-9): "[…] The filing of the former mine open-cast mine with river water started in 2010 and was reportedly completed in 2016 (Kovar et al., 2016). […]"

Also, Lns. 103-04): "[…] Water column stratification has been observed since 2009, when environmental monitoring of the hydrological system begun (e.g., Medová et al., 2015) […]"

Medová, H., Přikryl, I., Zapomnělová, E., and Pechar, L.: Effect of Postmining Waters on Cyanobacterial Photosynthesis, Water Environment Research 87, 180–190, 2015.

**RC2. 3b**. Do the meromictic conditions of the lake vary seasonally? Please present supporting information about this too.

**DP2.4b.** Yes. To address the reviewer request we revised the MS, Lns. 350-53, to now read: "Short-lived changes in redox potential of about 150 mV in the bottom water column were recently considered by Umbría-Salinas et al. (2021). These changes have effects on water column speciation (Fig. B1, Appendix B), and affect the partitioning of several redox sensitive metals that bind to reactive iron phases in the upper sediments (Umbría-Salinas et al. 2021, for details)."

Also, the revised Fig. B1 now shows the Eh-pH variability that we evaluated in the HAZMAT paper.

**RC2. 5. Line 95**: the authors talk about the present oligotrophic stratified conditions of the lake as a topic sentence. First, I would like to see physico-chemical profiles at this point to ease the understanding of such conditions for the reader. I also would like the authors to describe in more detail such conditions in all stratified layers. Finally, the last sentence of the paragraph, starting in line 100, deserved more written description too.

**DP2.5.** A call to Fig 2a, describing details on stratification of the bottom water column, is now added to the lines indicated by the reviewer (now Ln. 96-97). The reference to the oligotrophic state was deleted for being extemporary at this point, as kindly noticed by the reviewer.

Additional description on the Pourbaix diagrams in Fig. B1 was added (Ln. 108-09).: "[…]. The stability diagrams show that under the current aqueous physicochemical conditions, colloidal Fe oxyhydroxides form in the bottom anoxic waters (Fig. B1)."

**RC2. 6a.** Methods In line 13 [S1], when the authors refer to ~11 mL water samples, is 11 mL the aliquot amount referred to in line 11?

**DP2.6a.** The description provided is clear on this sampling step (now Lns. 188-121): Aliquots of the lake water collected were immediately transferred from the sampler to pre-cleaned (i.e., rinsed with $ddH_2O$ and oven-burned 550 ºC), 12 mL exetainer septum capped vials (Labco) pre-filled with He(g) and 1mL NaCl oversaturated solution (40%) for $CH_4$, or 1 mL 85% phosphoric acid for $\Sigma CO_2$. On board, the vials were filled with ~11 mL water samples […]". No other changes were implemented regarding this query, but please note that Methods is now moved to the main text, its extended section 3.

**RC2. 6b.** In addition, how many samples were collected? Were they collected along depth? where exactly?

**DP2.6b.** Lns. 115-19 now clarify on this important matter: "[…] Lake Medard (from 47 to 55 m depth) in its central location (Fig. 1a, star). The probing resolution was 1 m above and below the $O_2$ minimum zone and 0.5 m at the redoxcline. Water column samples (n = 8 and 4 replicates) were collected in November 2019 using a Ruttner sampler with a capacity of 1.7 L. Flushing/rinsing of the sampling device with distilled water ($dH_2O$) was performed between samples. A total of eighth samples were taken at 47, 48, 48.5, 49, 50, 52, 54 and 55 m depth. Replicate samples were taken at 47, 48.5, 50 and 54 m depth."

Also, Ln 135: "[…] A total of 11 water samples (i.e., 47 to 54 m depth and replicates), each consisting of 1 L water, […]"

**RC2.6c.** In section SM1.1.3, please clarify the number of samples taken. The same applies for SM1.1.4 and include information about samples from which depths (or layers) were considered for the cation and ion concentration analyses.

**DP2.6c.** Please refer to DP2.6b, above.

**RC2.6d.** In section1.1.5, please clarify the following questions: were the 11 water samples (line 29), the same as described in section SM1.1.3? If so? why eleven? does this number include biological replicates? are these only from two depths? it is important to clarify, where these samples were taken along depth. More details of the PCR and sequencing protocol would benefit future researchers and reproducibility of the methods.

**DP2.6d.** Please refer to DP2.6b.

**RC2.6d.** Were the mineralogical (SM1.2.2), isotopic (SM1.2.3), and SEM (SM1.2.4) analyses applied to all sliced sediments?

**DP2.6e.** Yes. The intro paragraph to section 3.2 (Methods applied to solid phases) reads:

**"3.2 Sediment samples**

We also sampled the upper anoxic sediment column to a depth of ~8 cm. The mineralogy of these sediments was qualitatively and semi-quantitatively assessed via X-ray diffraction (XRD). The $\delta^{34}S$ and $\delta^{18}O$ of gypsum ($CaSO_4 \cdot 2H_2O$), $\delta^{13}C$ of siderite ($FeCO_3$), and $\delta^{34}S$ isotope values of pyrite ($FeS_2$) from these sediments were also measured. Scanning electron microscopy aided by electron dispersive spectrometry (SEM-EDS) was used for textural analyses focused on the S- and/or Fe-bearing phases. In addition, a sequential extraction scheme (after Poulton et al., 2004; Goldberg et al., 2012) was conducted to characterize the sedimentary partitioning of reactive Fe and Mn fractions. Further details follow […]"

**RC2.7 Results and Discussion, Subsection 4.1:** I am little confused about what is shown in Figure 2a. What is happening above 48 m? To what depth are you referring to? depth of the lake water, or depth of the whole lake? Based on what is presented in Figure 2a, I interpret

that the depth of the water column is only ~10 m? I am sorry if it is not that obvious to me, but it might be worth to clarify.

**DP2.7.** To address the lack of clarity kindly pointed out by the reviewer, the caption has been also modified as follow: "[…] Lake Medard in its central sampling location, which has a maximum depth of 56 m (a) [… ].". Also, all figures portraying a depth profile now read, in their vertical axis, "Water column depth"

**RC2.8 Ln 121:** the authors refer to several previous profiles of the lake. Do you have previous profiles? Are they published somewhere? or included in the supp info?

**DP2.8.** Yes. Ln. 311, reference added: "[…] Petrash et al. (2018)".

**RC2.9 Ln. 135:** "The hydrochemically different monimolimnion persists in the deepest depressions of the lakebed throughout the year; although with slight variations in the monitored Eh range that could be accompanied by minor (±1 m) shifts in the vertical position of the redoxcline." Can you show profiles of this on the supp info?

**DP2.9.** Short-lived Eh-Ph variations are now shown in Fig. B1 as per this request of the reviewer. Please note that a publication dedicated to evaluating this observation and their implications is given as reference: Umbria-Salinas et al. (2021).

**RC2.10** [caption of figure 2], authors refer to dysoxic (n=4) and anoxic (n=3): at which depth were these samples collected, respectively?

**DP2.10.** Caption to Fig 2b was modified as follow: "[…] quantified in the dysoxic (n= 4, 48 m depth) and anoxic (n= 3; 54 depth) waters of Lake Medard […]"

**RC2.11:** Authors included a separating line referring to the redoxcline in Fig2b. Does this mean that the upper part of Fig 2b corresponds to the myxo-hypolimnion and the lower part to the monimolimnion? If so, please clarify it in the figure too. In addition, what are the red crosses? Could you also include an explanation in the caption?

**DP2.11.** Fig. 2b has been re-produced for clarity as per this important request of the reviewer.

**RC2.12 [Section 4.2]:** In line 150, the given DOC concentrations correspond to an average value of the 7 samples refer in figure 2b?

**DP2.12.** The average of all of the samples (and replicates) collected in the bottom waters. The text now reads (Ln 356): "The average of measured DOC concentrations in the bottom waters sampled is 1,050 ± 500 µM."

**RC2.13.** "A six- to ten-fold increase in concentrations of acetate, oxalate, and formate occurred towards the increasingly saline and O2-depleted bottom waters." This might be better to visualize in a profile. Could you please include one in the supp info?

**DP2.13.** No. A depth-profile is, unfortunately, not available for VFAs concentrations. This was the undesirable consequence of sample/replicate losses while fine-tunning/adapting the measuring conditions to the concentrations of multiple ions present, notably the increasing Cl⁻ concentrations towards the bottom.

**RC2.14.** In line 163, when authors referred to "[ΣCO2] were inversely correlated with the δ13C values", are they referring to figure 3d.

**DP2.14.** Referred to the profiles showed in Figs 3a-b (now clarified in Ln. 375). in

**RC2.14.** Paragraph starting in line 169 should have included a reference to Table 1 somewhere.

**DP2.14.** The edited section, now starting in Ln. 380, includes few references to Table 1, where appropriate.

**RC2.15.** About Figure 3d referred in line 189, I thought this figure was referring to the water samples. Please, clarify or correct accordingly.

**DP2.14.** The flux we referred in Fig. 3d and text is from the sediments to the water column. To clarify, Ln. 393 now reads: "[…] The range of estimated isotopic C values of the $CO_2$ flux from the sediments to the water column is between -3.0 and -4.2 ‰ (Fig. 3d). […]"

**RC2.15 Section 4.3:** Colours in Fig 5 must be changed. In 5a, there are two yellows, two greens, two light blues corresponding to different organisms, making it hard to interpret the figure and correlated with the written text. Fig 5b is even harder to differentiate colors and organisms.

**DP2.15.** Colour palettes used in the revised Fig. 5 were updated as per this relevant request of the reviewer.

**RC2.16.** While describing the microbial community, authors should be more quantitative (avoid low or high and refer to percentage). How much does "increases significantly" or "the abundance peak" mean? In addition, please be specific if what is shown in Fig 5. corresponds to normalized abundance in percentage with respect to the whole community or only among each microbial group shown separately in a b and c.

**DP2.16.** We did not implement quantitative PCR, in consequence the values are described for ecological interpretations only, e.g., *there is an increase in relative proportions {for a given OTU} from z1 to z2*. The value of amplicon reads do not constitute a definite measure of the *real* composition of the community, and limitations of 16S rDNA analyses has been discussed intensively in the microbial ecology literature, and we now cite a recent work that nicely clarifies the general current view on this matter (Ln. 59-61 at the introduction section): "Despite limitations linked to quantitative biases introduced, for example, during sample DNA extraction, PCR amplification, and uneven coverage of universal primers across phylogenetic groups, the sequencing of amplified fragments of prokaryote rRNA genes can provide insights useful for ecological deductions (see Piwosz et al., 2020)."

**RC2.17.** In section 4.3.2, the subtitle refers to dissolved Mn and Fe, are they total concentrations? otherwise please be specific and in accordance with what is shown in Fig 4b: MnII and FeII.

**DP2.17.** The subtitle now reads (Ln. 493): "4.3.2 Dissolved divalent manganese and iron, and the iron-utilizing prokaryotes"

**RC2.18.** In addition, do you have concentrations of Mn(IV)? How do authors know Mn(IV) is settling down from the upper layer? Or Fe(II) is difussing upwards?

**DP2.18.** These conflicting assertions are, in our opinion, valid educated guesses based on the well-known geochemical behaviour of Mn and Fe in redox interfaces and across concentration gradients, such as depicted in Figure 4 and numerically described in Table 1 (please see classic references cited: Davidson 1993; Loveley and Phillips, 1988).

No particulate metal concentrations profiles were produced as part of this study.

**RC2.19.** In line 291, do authors have mineralogical evidence of this fact" "Dissolved phosphate is re-complexed back onto nanocrystalline and amorphous ferrihydrite-like phases precipitating at the redoxcline." The same comment for mackinawite mentioned in line 295.

**DP2.19.** Phosphate: We have presented (in Table 1) geochemical evidence for phosphate solubilization occurring in the monimolimnion (i.e., increasing dissolved phosphate concentrations towards the lakebed), with a decreased concentration trend towards the redoxcline that is, on the other hand, indicative of complexation of the oxyanion via biotic and/or abiotic amorphous iron oxides formation at the oxycline (this is portrayed in Fig. 8).

Mackinawite: this is the prevalent metastable precursor of pyrite, and it is rather difficult to quantify in most practical cases, particularly when such cases involve field sampling. However, to comply with the query by the reviewer—regarding inferred monosulfide precipitation, we have added the following text (Ln.554-55): "[…] circumstantial evidence for FeS precipitation, with another being $\delta^{56}Fe$ values that increased across the redoxcline and towards the SWI

(Petrash et al., 2022)". On this matter, further information was provided (Ln 851-54): "[…] results from an undergoing $\delta^{56}$Fe systematics study (Petrash et al., 2022) firmly support co-recycling of Fe and S. An increase in the relative proportion of dissolved $^{56}$Fe towards the lakebed ($\delta^{56}$Fe = +0.12 ± 0.05 ‰;) is then ascribed to precipitation of monosulfides, whilst precipitation of oxyhydroxides at the redoxcline leads to depletion of the residual dissolved Fe(II) in heavy isotopes at the redoxcline ($\delta^{56}$Fe = -1.77 ± 0.03) (e.g., Busigny et al., 2014).

**RC2.20.** In line 303, when referring to *Pseudomonas* spp., do authors have any control showing that *Pseudomonas* was not part of the extraction kits, or sampling material?

**DP2.20.** Yes, controls had not enough extractable DNA for PCR. However, should the flaw suggested by the reviewer be the case, we would not expect to have contamination by this spp. affecting only one, intermediate sample and its replicate, but not the neighbouring samples. *Pseudomonas* was also identified in Petrash et al. (2018) in other sampling location, but using different isolation kit. In consequence, we are confident in *Pseudomonas* spp. being a relevant player in the metal respiring community near the redoxcline.

**RC2.21.** In 310, include a reference for "anoxygenic phototrophic and nitrate reducing species (*Magnetospirillum* and *Ferrigenium*; Fig. 5b, Supplement 2)", and "*Azospira*-like species."

**DP2.21.** Done. Ln 533. Also please note that the Supplement referred is now 1.

**RC2.22.** In line 323, when referring to the peak of *Geobacter,* include where specifically and how much?

**DP2.22.** The text indicated by the reviewer has been modified as follow (now Ln. 543): "[…] within the monimolimnion, at about 54 m depth." Again (as per DP2.16, above), we have no qPCR data, and we used universal primers that are not specific for Geobacteracea. The universal primers, however, did amplify a relatively higher abundance of *Geobacter* that iincidentally coincides with the peak of iron reduction. Importantly, if of relevance for the reader, please note that the Krona chart in [now] Supplement 1, interactively produces the # reads for any of the bacteria or archaea present in each sample/ or replicate that we evaluated.

**RC2.23.** There are some names of organisms in Fig 5b that are not mentioned in the text. Should you better remove them from the figure and include them, as other less abundant taxa, or mentioned them in the text.

**DP2.23.** Non-mentioned organisms in Fig 5b are now removed from Fig. 5b, as per this relevant suggestion of the reviewer.

**RC2.24.** In line 345, in Fig 5c is only as *Thioalkalivibrio...*should you add the species name too as you did in the text?

**DP2.24.** "*[…] paradoxus*" added (Ln. 566).

**RC2.25.** In line 349, authors mentioned "The abundance of *S. hydrogenivorans* increases in parallel to a decrease in the *T. paradoxus*-like bacterium, which suggests that the latter may be at a disadvantage and limited by organic C fixation under the specific hydrochemical conditions prevailing at the redoxcline" With the current colors in Figure 5, it is difficult to see what you are showing in the text.

**DP2.25.** True. The palette was updated. Also, please note that numerical values are in the Krona chart provided as supplement.

**RC2.26 Ln. 360:** which bar corresponds to *D. acetoxidans* in Figure 5c. The same comment for *Desulfaticacillum* in line 365 and *Sulfitobacter* in line 366.

**DP2.26.** Please see DP2.25.

**RC2.27.** In general, with the current colors in Fig 5, it is difficult to agree with the conclusions stated by the authors in section 4.3

**DP2.27.** Please see DP2.25.

**RC2.28 Section 4.4 Ln 374:** do authors have values to support the "weak correlation"?

**DP2.28.** The supporting values are listed in Table 1. The $R^2$ (~0.16) from an attempt for linear correlation of such values (shown below only), is now provided in the text (Ln. 595).

[Figure]

**RC2.29:** A reference is needed for the following statement: "It is also within the range observed in studies of S disproportionation reactions generally proceeding under anoxic conditions" in line 383.

**DP2.29.** True. Relevant references now added (Ln. 604): "[…] observed in studies of S disproportionation reactions generally proceeding under anoxic conditions (e.g., Böttcher et al., 2001, 2005)"

**RC2.30:** Reference needed for the examples given in line 409.

**DP2.30.** There are no "examples" given in line 409 of the preprint, but stable isotope measurements conducted in the acidic drainage shown in Fig. 1. These are plotted as well in Fig. 6, which now with colours better allow distinguishing them.

**RC2.31. Section 4.5:** Be more quantitative with respect to sentences like in line 445: "…. increase slightly towards the bottom of our 8 cm depth core but their abundance, relative to total iron, decreases downwards" or 451: "a significant increase…"

**DP2.31.** Quantities are listed/detailed in Table 2, but we have deleted qualificatives such a 'significant' or 'slightly', while referencing changes listed by depth (cm) in Table 2.

**RC2.32. Ln. 454:** a Sect. 4.6.3 is referred but not found in the current version of the manuscript.

**DP2.32.** Section 4.6.1 is now referred (there was a typo), Ln. 425.

**RC2.33.** Section. 4.7, Ln. 595, name which "scarce examples" authors are referring, as well as in line 596: add reference and name which lakes

**DP2.33.** This request of the reviewer has been addressed as follow (now Lns. 873-76): "[…] the scarce examples of redox stratified marine basins existing today (i.e., Black Sea, Cariaco Basin; Lyons et al., 2009), nor in the few natural mesotrophic to eutrophic meromictic lakes that are presumedly analogues to redox stratified ancient oceans (see Koeksoy et al., 2015)."

**RC2.34.** About figure 8: Nice figure but there are some improvements to be done: 1) a legend is required to understand symbols and colors in the figure. 2) add a depth profile and names of each layer. 3) why is it necessary the venn diagrams for the microbes, what each color of the circles mean? Add the biogeochemical role of each microbial group included in the figure.

**DP2.34a.** The caption of the figure has been simplified, and to address further the chosen grouping representing fuctionalities we added further information. The caption now reads:

[Figure]

"Figure 8. Scheme summarizing the speciation and stable isotopes ranges of sulfur-bearing phases (pyrite, $S^0$, CRS; gypsum, GY) and siderite (SID) and the biogeochemical cycling mechanisms likely operating in the redox stratified Lake Medard and its SWI. (Background colours as in Fig. 2) The prokaryote groups depicted represent nitrate, iron and sulfur utilizing species identified via 16S gene amplicon sequencing (see text for details)."

Note also that, as suggested by the reviewer, additional information was added to the edited version of the figure.

**DP2.36.** The reviewer kindly pointed out also a list of typos, repeated text, etc. These minor issues indicated by the reviewer in the preprint, were all addressed. Also, sentence tense agreement also revised thoroughly through the text.

---

## Author Comment (AC3)

**Reviewer 3 (Dr. Alexandra A. Phillips)**

**RC3.1** Cutting jargon: this paper is strongest when integrating results across their interdisciplinary dataset. However, the paper is in many places (namely the abstract, introduction, and final discussion section) unnecessarily complex and loses the non-expert reader. The authors should revisit these parts of the manuscript with an eye for unnecessary jargon - places they will lose interested geobiologists in a complicated explanation of geology, for example. In places where jargon is unavoidable, offering a few more definitions to the reader will help broaden the readership of this really interesting interdisciplinary study.

**DP3.1.** As per this relevant reviewer's suggestion, we have revisited and edited the text for the sake of simplicity, explained limnological-specific jargon, and simplified the text whenever possible. For example (Lns. 48-50): "[…] Natural lakes that display permanent stagnation and marked redox gradients in their water column are termed meromictic.[…]"

Also, Ln. 55: "[…] This newly formed, lacustrine system features low nutrient contents (i.e., it is oligotrophic), […]"

We also simplified our study site description, which now has no jargon related to the Miocene rift stage, fault mineralization, and paleolake development.

**RC3.2** Improved figures: The figures should be reworked with a goal of consistency. Much of the results/discussion center around different zones of the lake's water column, but it is difficult to orient yourself across the many figures. Figure 2a panel 4 does a nice job showing the mixolimnion, hypolimnion, and monimolimnion - I think it would be helpful to see these zones in all the figures - either as the different shades of gray like in the Eh figure or with dashed lines and labels. A few of the figures also appear low resolution - namely, figure 5, 6, and 7. Other comments specific to each figure can be seen below.

**DP3.2.** We have carefully implemented this relevant suggestion to all the figures where a water column profile is shown. These now consistently display the redox stratification. All figures are 300 dpi, portable network graphics.

**RC3.3.** Separating results and discussion: It was difficult as a reader to follow the results and discussion section - I wanted to already be acquainted with much of the data before seeing it synthesized. I found myself jumping back and forth across the sections often. My suggestion to the authors is to split the results and discussion and focus the results to be a succinct section, with extraneous details moved to the SI.

**DP3.3.** We tried implementing this suggestion prior to submission, and again now—to comply with the reviewer suggestion, but our feeling is that it detrimentally affects the intended integration of microbial ecology and geochemistry, also the importance of certain geological (e.g., mineralogical) observations for geochemical/ ecological interpretations. Then, the MS ends being even more disseminated and sections largerly unbridged. Therefore, we prefer keeping the manuscript as a Results and Discussion-type report and did not fulfil this suggestion of the reviewer.

**RC3.4.** Some parts felt too long and should be more succinct, while other parts begged for more detailed discussion

**DP3.4.** We have shortened some sections (e.g., Sects. 2 and 4.2.2), while adding additional discussion of results, where presumed desirable (e.g., Lns. 915-20): "Our observation could make the case for niche differentiation linked to high loads of dissolved metal concentrations conferring a competitive advantage to these archaea under suboxic conditions (e.g., Gwak et al., 2019). Alternatively, the NH$_3$-oxidizing archaea sequenced predominantly in the suboxic waters possess a yet to be explored tolerance to anoxia (see Mußmann et al., 2011). For instance, *Ca.* Nitrosocaldus encodes a pyruvate:ferredoxin oxidoreductase that is rather uncommon among aerobic ammonia oxidizers (Daebeler et al., 2018), but which is used by most anaerobes to catalyze the decarboxylation of pyruvate to form acetyl-coenzyme A (Chabrière et. al., 1999)."

**RC3.5. More methods:** Currently, [in the main text] there are not enough details for someone to replicate any of the work or think critically about advantages or disadvantages for any method. Much of the SI methods section should be moved to the main text.

**DP3.5.** Methods are now moved to the main text as per reviewer's request. Also, missing information on HP-LC was added (see DP3.11f, below), and all additional editions suggested by the reviewer in RC3.11 were incorporated.

**RC3.6. [Abstract]:** I would recommend moving the last few lines on the importance of the study more broadly to earlier in the abstract, perhaps as the second sentence, and then expanding/clarifying the point in the current first sentence that this geochemical situation is an unusual but exciting case study

**DP3.6.** We implemented the abstract editions kindly offered by the reviewer.

**RC3.7 [Introduction] Line 32:** It would be helpful for those less familiar with limnology terms to define meromictic briefly, perhaps simply as "indefinitely stratified" or something similar

**DP3.7.** Done, please see DP3.1.

**RC3.8 Line 33:** "common sulfate deficiency" feels unnecessarily complicated, do you mean low in sulfate or absent of sulfate?

**DP3.8.** Yes. The revised text (Ln. 44) now reads: "but low sulfate concentrations"

**RC3.9 Line 39:** If possible, I think it would be helpful to add one more sentence about the importance/relevance to paleo-studies - the connection to me right now is a little weak so would be great to strengthen that point a little more - how exactly do they better inform Precambrian Ocean redox stratification models?

**DP3.9.** Done. Lns. 51-53 now reads: "[...]. They constitute natural labs that permit constraining master aqueous geochemical variables pertaining to iron mineralization and, thus, are relevant to disentangle key aspects on the deposition of ancient iron formations"

**RC3.10 Line 47:** Would be helpful to include a mention of the lake's pH as well (anywhere in this introduction)

DP3.10. Done, Lns. 54-56 now read, "[…] Ferruginous water columns that also contain elevated dissolved sulfate concentrations are not uncommon in acidic shallow pit lakes (e.g., Denimal et al., 2005; Trettin et al., 2007), and have also been reported from the pH neutralized post-mining Lake Medard in NW Czechia (Petrash et al., 2018; Fig. 1a)."

**RC3.11a [Methods]:** I think a majority of the SI details should be moved to the main text - because of this I also have line edit suggestions for the SI methods.

**DP3.11a.** Please see 'DP3.5' above. We thank the reviewer for this relevant suggestion.

**RC3.11b.** Water sampling: what depths were sampled? Did those change based on the physiological parameters prior to water sampling?

**DP3.11b.** Methods now clearly state the water depths sampled (Ln. 224-229): "[…] The probing resolution was 1 m above and below the $O_2$ minimum zone and 0.5 m at the redoxcline. Water column samples (n = 8 and 4 replicates) were collected in November 2019 using a Ruttner sampler with a capacity of 1.7 L. Flushing/rinsing of the sampling device with distilled water ($dH_2O$) was performed between samples. A total of eight samples were taken at 47, 48, 48.5, 49, 50, 52, 54 and 55 m depth. Replicate samples were taken at 47, 48.5, 50 and 54 m depth."

**RC3.11c. Line 11:** were the exetainers cleaned prior to sampling?

**DP3.11c.** Yes, this is stated in Ln. 301-02 of the methods sections that now read: "[…] from the sampler to pre-cleaned (i.e., three-times rinsed with $ddH_2O$ and oven-dried at 550 °C), 12 mL exetainer septum capped vials (Labco), […]".

**RC3.11d.** Line 13: define PES

**DP3.11d.** PES acronym definition is now in Ln. 276 (its first appearance): "[…] using sterile high flow, 28 mm diameter, polyethersulfone (PES) filters to remove particles >0.45 μm […].

**RC3.11e.** Line 22: please define which anions and cations

**DP3.11e.** Defined as per this important suggestion kindly provided by the reviewer (Lns. 230-34): "[…] (ii) mass determinations of cations (iron, manganese, potassium, sodium, magnesium and calcium); (iii) high pressure liquid chromatography for concentrations of chlorine, sulfate, nitrate, ammonium and phosphate anions, and VFA abundances in the bottom water column; […]".

**RC3.11f.** Line 70: more details are needed on the IC method used for the VFA analysis - what is the run time, column used, etc?

**DP3.11f.** More details added to Methods, Ln 288-299 now reads:

**"3.1.5 Ions, ammonia, and VFAs concentration analyses**

Ions and ammonia concentrations were measured in filtered, unacidified water sample aliquots using high pressure liquid chromatography (HP-LC) on a Dionex IC25 IC + Eluent Generator EG40 instrument at the Biology Centre of the Czech Academy of Sciences, Ceske Budejovice. We used a Dionex IonPac AS11-HC ion exchange column (2x250 mm) that permits resolving analytes in our complex sample matrices in a single run (45 min) by using separation of inorganic anions via a large-loop injection on a microbore (2 mm) isocratic pump. Ammonium ion detection/quantification was achieved via a fluorescence detector after the post-column derivatization. A combined stock calibration standard solution featuring environmentally relevant anions ratios was used for determining their concentrations and was prepared from corresponding analytical-reagent grade sodium salts. VFAs (Volatile fatty acids) were measured on the same instrument. To optimize and calibrate the method for VFA analyses, and determine the limits of detections, we used stock standard mixtures of IC grade formate, oxalate, acetate, lactate, pyruvate, and butyrate standards for preparing working saline stocks solutions. Detection limits were better than 60 ppb for lactate and oxalate, and 200 ppb for byturate, formate, and acetate."

**RC3.11g.** Line 77: How much 5M NaOH was added?

**DP3.11g.** Information missing was added (Ln. 329): "[…] then 50 mL of 5 M NaOH were added. […]"

**RC3.11h. Line 87:** should be "quantified gravimetrically" - also can you clarify what you mean here? Just weighed?

**DP3.11h.** Edited to "weighted" (Ln 340).

RC3.11i. Line 147: Sort of unusual to report 3 sigma, maybe just report 2 sigma as you did earlier to be consistent

**DP3.11i.** Typo deleted, it corresponded to preliminary results considered in an earlier draft.

**RC3.12.** Supplement figure 3: this appears very pixelated on my download; can you make sure the figure has high enough resolution?

**DP3.12.** A higher resolution printout of the PREEQC results for mineral's SI is now provided in Supplement 2.

**RC3.13.** Line 119: awkward to start the sentence with "Figure 2a"

**DP3.13.** Sentence corrected as per this request of the reviewer. The offending line now reads (Lns. 423-29): "Physicochemical parameters measured in the dysoxic to anoxic bottom waters at the time of sampling (November 2019) are shown in Fig. 2a […]"

**RC3.14.** Line 150: can you elaborate on the DOC concentrations? What does that tell you about the system - is it typical or unusual? A little more discussion would be great, especially because you mention in the abstract that SR may be limited by low amounts of metabolizable OC

**DP3.14.** Done, the section now states the following in its first paragraph:

"The average of measured DOC concentrations in the bottom waters sampled is 1,050 ± 500 µM. This range of values was higher than observed in the bottom waters of meromictic lakes such as or Matano (< 100 µM; Crowe et al., 2008), or Pavin (300 ± 100 µM; Viollier et al., 1995). DOC is generally comprised of relatively high molecular weight organic compounds (not quantified here), such as cellular exudates from alive and senescent planktonic microorganisms (e.g., algae, protists, bacteria) and their degradation products. Probably present in solution were also soluble humic substances derived from the biological breakdown of refractory organic matter (i.e., lignite particles) in the sediment (Petrash et al., 2018). VFAs are linear short-chain aliphatic mono-carboxylate compounds produced during anaerobic degradation of the organic compounds referred above. They serve as C sources and electron donors for the planktonic microbial heterotrophy and were therefore quantified here. VFAs in the bottom waters […]"

**RC3.15. Line 155:** what is your hypothesis for the change in VFA concentration in the different layers of the lake? Can you relate this more explicitly to your 16S data at all?

**DP3.15.** We have related changes in VFA with the microbial ecology for example in Lns. 701-04:

"Although speculative, it is possible that microbial sulfate reduction (MSR) is responsible for the observed lactate depletion in the bottom waters. Therefore, the complete (oxidation to $CO_2$) and incomplete (to acetate) oxidation of lactate by MSR could be a factor contributing to the slight decrease in pH in the monimolimnion (see Gallagher et al., 2012). […]"

Also, Lns. 1075-79, i.e., with regard to *Desulfobulbus* more likely disproportionating S0, instead of thriving organotrophically: "Pyruvate, as lactate, was found below our detection limits across the bottom water column where sequences distantly related to *D. propionicus* (91 % similarity in 428bp) appeared to be particularly abundant (Fig. 5c; Supplement 1).

**RC3.16. Line 159:** I think a mention of pH should come much earlier, at least very early in results, if not hinted at in introduction - my initial assumption from hearing about a post-mining lake would be to expect really acidic conditions, so saying that the pH is closer to 7-8 earlier would be helpful.

**DP3.16.** As per reviewer suggestion, pH is mentioned in then introduction (DP3.10). Also, early in the results (Sect. 4.1, 1st paragraph, now Lns. 430-37 reads): "The pH in the hypolimnion was ~8.2 and decreased moderately downwards, reaching 7.4 ± 0.2 units near the anoxic SWI."

**RC3.17** Line 163: Please put some of the d13C values in the text, such as an average or range

**DP3..17** Done. Lns 702-03: […]. The $\delta^{13}C$ values are in the range +0.2 to -4.1, and were directly correlated with the dissolved sulfate concentrations [$SO_4^{2-}$] (Table 1), […]

**RC3.18.** This discussion on 4.2.2 on total DIC seems very lengthy compared to the other results sections and could be shortened for readability and to better emphasize the main points.

**DP3.18.** The discussion on total DIC has been now streamlined, shortened some 20 % (it begins in Ln 697).

**RC3.19a.** Line 171: Please change "d13C signatures" here and elsewhere to d13C values.

**DP3.19a.** Done. We now referred to values regarding this isotope system.

**RC3.19b Line 189:** Instead of "estimated isotopic C signature of the CO2" say either estimated C isotope composition or estimated d13C value

**DP3.19b.** Edited as per reviewer suggestions (Ln. 733).

**RC3.20 Line 370:** the title of 4.4.1 is awkward, maybe "Isotopic evidence"

**DP3.20.** The sub-section titles were edited as per reviewer's suggestion. The specific title now reads: "4.4.1 A proxy for disproportionation" (Ln. 1118)

**RC3.21.** Line 373: I would suggest avoiding "heavier" and just stick with "enriched in 18O"

**DP3.21.** Done, Ln. 1121.

**RC3.22 Line 375:** a number itself can't be narrow, so maybe change to something like "the bottom waters had a narrow range of d18O values: X to Y"

**DP3.22.** Text edited as per reviewer suggestion, Ln. 1124: "The ambient bottom waters had a narrow $\delta^{18}O_{H2O}$ range of values: -6.1 to 6.7 ‰ […]"

**RC3.23 Line 410:** you say that for the initial sulfate composition it is reasonable to assume its similar to the nearby acidic drainage and the pit lake before flooding - the second seems more reasonable to me but you dont report those values in the text? What are those?

**DP3.23.** Those values were reported in the compilation made in Fig B2 (Appendix B).

**RC3.24. Line 574:** extra parenthesis dangling - sentence is also not grammatically correct, so should be fixed

**DP3.24.** Corrected Lns. 1484-85: "[…] could be ascribed to incomplete microbial sulfate reduction, with an additional open system oxidative sulfur cycling also being probably active"

**RC3.25 Line 595:** more references needed here, overall really enjoyed this section on the paleo implications!

**DP3.25.** The reference provided, Lyons et al., 2009, is a review. But the text now reads (Ln. 1638): "[…] existing today (i.e., Black Sea, Cariaco Basin; Lyons et al., 2009, and refences therein), […]"

**RC3.26 Figure 1:** I really enjoyed this figure! Especially nice to see the lake from 2005 to 2020. The figure caption has a call out to panels b and c, but not to panel a, so that should be added.

**DP3.26.** The figure's caption is now corrected with regard to missing information.

**RC3.27 Figure 2:** In part b I think the main idea is to compare the VFA concentrations above and below the redoxcline - its currently hard to see that difference and the relative differences between other VFAs - it would be more clear to show these on all the same plot - so instead of 6 separate figures just one figure

**DP3.27.** The figure's caption is now corrected as per DP3.26. The revised Fig. 2b now shows, in the background, the stratification defined by the Eh gradients.

**RC3.28 Table 1:** I'm a little concerned by the errors in the ammonium measurements - especially that surface sample, where the error is larger than the measured value - is there also a reason why phosphate doesn't have concentration brackets - maybe just an error?

**DP3.28.** We thank the reviewer for her attention to details that allowed pointing out our mistake in data transcription to Table 1. Based on our replicates, errors in HC-LC data for ammonia, and in other anions simultaneously measured, are now correctly stated and do coincide to what was shown in Fig. 4a.

**RC3.29 Figure 3:** For panel A can you change it to mM so that the range is not to 12000 in the axis? Also needs more tic marks to see values in between, panel c also needs further tic marks for sulfate concentration. In the figure caption there is a CO2 that needs the 2 to be subscripted and "value" needs to be added after d13C.

**DP3.29.** The figure edition requested in this query were implemented.

**RC3.30 Figure 5**: this figure is pretty hard to read with the colors as they are - I would think about the main point you are trying to make with this figure - instead of having the arrow for the redoxycline. I would make the edit I suggested earlier for all figures, having different shades of grey boxes in the background - its hard to compare the abundance of different microbes against each other because the scale changes across panels as well.

**DP3.30.** Figure 5 was modified as per reviewer's request. We conserved, however, the variable scales as they do directly convey information on the relative abundances of the functional groups that we considered.

**RC3.31 Figure 6**: there is one data point for d34S of sulfate that is much higher value - at ~13.5 permil? Is this an outlier?

**DP3.31.** Concerning such value, effectively it is an outlier (see boxplot below), the revised text (Ln. 1193-96) now reads: "In line with this assertion, at the monimolimnion we observed in dissolved sulfate a negligible sulfur isotope fractionation accompanying the recorded fractionation of oxygen isotope. Yet, we registered a small, but significant reverse sulfur isotope effect (+2.2 ‰) at the upper hypolimnion (Fig. 6a: 48 m depth). This isotope effect could be ascribed either to abiotic or biotic oxidation processes of intermediate S species accruing at this level of the water column (see Zerkle et al., 2016, their table 1)."

[Figure]

**RC3.32.** Should be mentioned in text - 6c would be a bit easier to see if the symbols were also colored if possible

**DP3.32.** Figure 6c now has coloured markers as per reviewer's suggestion.

**RC3.33.** I hope these comments were helpful and they assist in improving what is already a really interesting manuscript.

**DP3.33 Final Statement, acknowledgement:** We sincerely thank Dr. Phillips for suggestions, her time and dedication to this review, and for rising concerns that helped us improving the revised manuscript. Dr. Alexandra Phillips provided a clear and detailed review. We very much appreciate her attention to details, sound suggestions, and constructive criticism. Above we described how we addressed her comments, and in a single case where we disagreed (splitting results and discussion section, i.e., RC3.3), we explained why.

---

## Author Comment (AC4)

[revised manuscript text omitted]

0.036 wt. %) was precipitated as Ag2S by using a 0.3 M AgNO3 solution. Subsequently, CRS was liberated using a hot and acidic 1.0 M CrCl2 solution (Canfield et al., 1986). The resulting H2S was trapped as Ag2S. Mass balance after gravimetric quantification was used to calculate the amount of AVS and CRS. Concentration analyses of Fe and Mn dissolved in each of

165 these extracts were conducted via ICP-MS measurements (Xseries II, Thermo Scientific) at the Department of Environmental Geosciences, Czech University of Life Sciences, Prague.

**3.2.3 Sedimentary geochemistry and stable S and C and O isotope analyses**

Aliquots of the sediment samples were analyzed for total S (Stot) concentration using a CS analyzer (ELTRA GmbH). The detection limit was 0.01 wt. % for Stot. The relative errors using the reference material (CRM 7001) was  $\pm 2$  % for Stot.

- Total S for δ34S determination was extracted in the form of BaSO4 from the sediments. To evaluate the S isotope ratio of gypsum (δ34Sgy), first the heavy mineral fraction of the samples, which includes pyrite, was excluded by using 1,1,2,2-tetrabromethane (ρ= 2.95). The gypsum was then dissolved in ddH2O to extract sulfate. The free sulfate obtained was precipitated as BaSO4 as described above (Sect. 3.1.7). The BaSO4 was then converted to SO2 by direct decomposition mixed with V2O5 and SiO2 powder and combusted at 1000 °C under vacuum (10-2-10-3 mbar); mass spectroscopic measurements of the evolved SO2 were conducted on a Finnigan MAT 251 IR-MS dedicated to S isotope determinations.
- The results are expressed in delta notation and reported against the V-CDT and V-SMOW standards. The accuracy of the measurements was checked by also measuring international standards; reproducibility was better than 0.2 ‰.

The same IR-MS used to evaluate the  $\delta^{34}$ S isotope ratios of dissolved sulfate in the waters at the Biogéosciences Laboratory, Dijon, France, was used to evaluate the  $\delta^{34}$ S of the pyrite in the upper anoxic sediments. Prior to analyses, an AVS/CRS wet chemical extraction scheme alike the one described above was applied. The resulting H2S was trapped as Ag2S. Mass balance after gravimetric quantification was used to calculate the amount of AVS and CRS. After centrifugation, the Ag2S precipitate was washed several times with ddH2O and oven-dried at 50 °C for 48 h. The pyrite  $\delta^{34}$ S measurements were performed on SO2 molecules via combustion of ~500 mg of silver sulfide homogeneously mixed with an equal amount of WO3 using a Vario PYRO cube (Elementar GmbH) connected online via an open split device to the IR-MS. International standards (IAEA-S-1, IAEA-S-2, IAEA-S-3) were used for calibration. Isotope results are reported in the standard delta

185 standards (IAEA-S-1, IAEA-S-2, IAEA-S-3) were used for calibration. Isotope results are reported in the standard delta notation against the V-CDT standard. Analytical reproducibility was better than 0.8 ‰ based on replicates for standard materials and samples.

The isotope ratios of carbonate in the sediment fraction were evaluated—after removal of organic carbon with  $H_2O_2$ , by implementing the method described by Rosenbaum and Sheppard (1986). These were measured using a mass spectrometer

- 190 (Delta V, Thermo Fisher Scientific) coupled with a Fisons EA-1108 elemental analyzer at the Czech Geological Survey, Prague. The same instrument was used for measuring the sediment  $\delta^{13}C_{org}$ . For this purpose, the samples where finely milled, place in tin (Sn) capsules, and oxidized to CO2 at 1040°C in the elemental analyzer. The reproducibility of the isotope measurements for organic C was better than ±0.12 ‰, and better than ± 0.1 ‰ for both carbon and oxygen isotopes of siderite. For siderite, the accuracy of the measurement was monitored by analyses of the IAEA NBS-18 ( $\delta^{13}C = -5.014$  ‰,
- 195  $\delta^{18}O = -23.2$  ‰) and two in-house standards; the long-term reproducibility is better than 0.05 ‰ for  $\delta^{13}C$  and 0.1 ‰ for  $\delta^{18}O$ .

**3.2.4 Textural features**

For SEM of the sediments, we either used a TESCAN Mira 3GMU scanning electron microscope combined with a NordlysNano electron back-scattering diffraction (EBSD) system for semi-quantitative chemical petrography, or a FEI Magellan 400 for higher resolution imaging in secondary electron mode.

200

**REFERENCES CITED ARE PROVIDED IN MAITEXT**

---

## Author Response (AR1)

Dr Denise Akob,

Associate Editor, Biogeosciences

Dear Dr Denise Akob:

As you fairly mentioned, the reviewers did an excellent job providing constructive criticism, and their comments were carefully taken into consideration while editing our revised manuscript version.

In the enclosed rebuttal text, we described how each of the reviewers' comments and suggestions were addressed, and in a single case where we had to respectfully disagree (i.e., splitting results and discussion section, i.e., RC3.3), we do explain why.

We look forward to your editorial decision or to address, in due time, any follow up opinion or editorial requests you may have. Our goal is publication of our work (bg-2021-253-R1) at BG before the next EGU General Meeting in Vienna. We hope you agree that this now could be possible.

With kind regards,

Dr Daniel Petrash

Czech Budweis, 31.12.22

**Reviewer 1**

**Reviewer 1, Comment 1 (RC1.1).** This is an interesting system in that it has overlapping S, N, Fe, and C cycles all in the anoxic portion of the water column. This is unusual, b/c in many systems at least one of these is a functionally absent, a minor component, or undergoes minimal redox transformations because of the dominance of (an)other component(s).

**DP1.1:** The reviewer has kindly provided us with a relevant synthesis of the importance of our observations in this lake system, and it has been incorporated in the abstract and as introductory statement to a new section 4.7, titled "The imbalanced aqueous redox system in Lake Medard: synthesis". This introductory paragraph (Lns. 828-34) reads:

"The newly formed Lake Medard has overlapping S, N, Fe, and C cycles occurring in the anoxic portion of the water column. This is unusual in natural, redox stabilized meromictic lakes where at least one of these cycles is functionally diminished or undergoes minimal redox transformations because of the dominance of (an)other component(s). Alternation of two bistable states could be the case in natural lake systems that can be rendered ferruginous, and this alternation is largerly controlled by shifting trophic states. Accordingly, ferruginous conditions may occur in low productivity, organic-poor systems; whilst euxinic conditions would dominate in high productivity, organic-rich systems where production of sulfide depletes dissolved sulfate and may out titrate dissolved iron (van de Velde et al., 2021; Antler et al., 2019). […]"

**RC1.2.** The paper is mostly observational, but I think that's fine, because the authors have developed a nice model for the processes occurring that's shown in Figure 8. The paper presents a lot of results. I almost wish the authors could tie everything together a little more succinctly.

**DP1.2.** We have streamlined few sections of the manuscript while keeping our aim, which is to provide a meaningful account of current biogeochemical conditions in a transitional redox system. We have in all possible instances shortened and synthesized our results.

**RC1.3.** I think Figure 8 gives a nice summary of the processes involved. Maybe a little more time in the discussion focused on this model and a little less on the "paleo" implications would help the reader synthesize all of the observations into the process model.

**DP1.3.** As per reviewer suggestion we now dedicated a section (4.7, see DP1. Above) to better explain our biogeochemical model, also providing a synthesized view of the system based and our observations and interpretations. Reviewer 3, however, commented (RC3.25) on the "paleo" implications being quite enjoyable a section. So, we have chosen to keep it mostly as it as in the preprint version.

**RC1. 4.** Below, I provide some specific comments and suggestions that I hope can help improve the manuscript.

**DP1.4.** We sincerely thank the reviewer for her/his attention to detail while reviewing our draft/preprint; the comments and suggestion kindly provided were addressed as described below.

**RC1. 5.** Ln. 11. Do the authors mean "reductive Fe(III) dissolution"?

**DP1.5.** Yes, missing "Ï" is now added (Ln. 11)

**RC1. 6.** Ln. 16. "sustained" how?

**DP1.6.** Here we referred to a continuous genetic potential for anoxic sulfide oxidation. For clarity, the wording in the abstract has been modified as follow (Lns. 21-3): "[…] Yet, the planktonic microbial succession across the nitrogenous and ferruginous zones also indicates genetic potential for chemolithotrophic sulfur oxidation in the anoxic portion of the bottom water column […]".

**RC1. 7.** Ln. 17-18. What is the electron acceptor for sulfide oxidation? And if sulfide is oxidized all the way to sulfate, at which point does the sulfur disproportionation happen?

**DP1.7.** We considered that this information was adequately developed (within the length-limitations of the abstract section) in Lns. 19-21 (now 24-25): "Near and at the anoxic sediment-water interface, vigorous sulfur cycling, can be fuelled by ferric and manganic particulate matter and redeposited siderite stocks." There are no further changes regarding this reviewer's comment.

**RC1. 8.** Ln. 104. Change to "DNA extraction and MiSeq"

**DP1.8.** This has changed as per the Methods section now being part of the main text. Yet, please note that Ln. 120 now reads: "[…] environmental DNA (eDNA) extraction followed by MiSeq Illumina 16S rRNA gene amplicon sequencing […]"

**RC1. 9.** Ln. 105. Do you mean ICM-MS? Does mass spec work for specific ions? I don't think this is an adequate description of the analytical methods.

**DP1.9.** We thank the reviewer for noticing an omission/ and error in our description of analytical methods. Ion concentrations were determined using HP-LC. The following lines (Lns. 121-22) were added to address this flaw kindly noticed by the reviewer: "[…] (ii) mass determinations of cations (iron, manganese, potassium, sodium, magnesium and calcium); (iii) high pressure liquid chromatography for concentrations of chlorine, sulfate, nitrate, ammonium and phosphate anions, and VFA abundances; […]". Further details on how the measurements were carried out are in Sect. 3.1.5, subtitled "Ions, ammonia, and VFAs concentration analyses".

**RC1. 10.** I'm also al little bit leery of putting the entire description of nucleic acid-based microbial community analysis in the supplement.

**DP1.10.** The methods, including the specific set of analyses indicated by the reviewer, are now part of the main text, Sect. 3.

**RC1. 11.** Ln. 106. Change to "measurement of dissolved"

**DP1.11.** Done as per reviewer suggestion (Ln. 123).

**RC1. 12.** **Figure 2[b]:** Can the authors make the axes the same on panels above and below the redoxcline? My first impression was that there was no difference between any of the organic acids, but there are actually rather dramatic differences.

**DP1.12.** Fig. 2b has been modified as per reviewer suggestion. Now it better highlights the change in VFAs concentration recorded above and below the redox aqueous interface.

**RC1. 13.** Ln. 150. VFAs were a minor fraction of the total DOC. What is likely the rest? How labile might it be, and how does that inform the biogeochemical model?

**DP1.13.** The main components supporting microbial growth are simple mono- and oligomers that are only present in nM concentrations. We determined VFAs concentrations as they act as electron donor and or C source to heterotrophic microorganism that could influence the observed hydrochemical processes. As per this request of the reviewer, the composition of the DOC pool is inferred as follow (Lns. 355-58): "[…] DOC is generally comprised of relatively high molecular weight organic compounds (not quantified here), such as cellular exudates from alive and senescent planktonic microorganisms (e.g., algae, protists, bacteria) and their degradation products. Probably present in solution were also soluble humic substances (HSs) derived from the biological breakdown of refractory organic matter (e.g., lignite particles) in the sediment (Petrash et al., 2018). […]"

Regarding "how does that inform the biogeochemical model", for instance Ln. 540 speculates what might a member of the Fe-utilizing community be doing (Ln 540): "[…]The HSs derived from lignite degradation contain abundant aromatic compounds (Wang et al., 2017). […]" Other research being conducted by out team also involves electron transfer between solubilized, re-oxidable HSs and iron (III) particles, and with Geobacteracea as mediators.

**RC1. 14.** Ln. 167-168. Wouldn't sulfate reduction induce increase in pH? You're producing carbonate alkalinity and reducing a strong acid (sulfate) to a weak acid (sulfide).

**DP1.14.** Usually, that is the expected effect. But, MSR has also been experimentally shown to decrease pH when lactate is the electron donor. Our annotation speculates on such an effect potentially occurring here as exhaustion of lactate is linked to the increased number of OTUs functionally associated with $SO_4^{2-}$ reduction in the monimolimnion. The following line address further the speculative assertion (Lns. 378-79): "Therefore, the complete (to $CO_2$) and incomplete (to acetate) oxidation of lactate by MSR could be a factor contributing to the slight decrease in pH in the monimolimnion (see Gallagher et al., 2012). […]"

Gallagher K.L. Kading T. Braissant O. Dupraz C. Visscher P.T.: Inside the alkalinity engine: The role of electron donors in the organomineralization potential of sulfate-reducing bacteria: Geobiology 10, 518–530, 2012.

**RC1. 15.** Ln. 221. Please be consistent in including the charge for nitrate

**DP1.15.** Charge of nitrate is now constantly stated in all instances where it appears on the text.

**RC1. 16.** Ln. 223. Please change "sequenced" to "detected"

**DP1.16.** Done.

**RC1. 17.** Ln. 232. By "abundance peak" do you mean maximal relative abundance?

**DP1.17.** Yes, as in revised Ln. 452: "The maximal relative abundance of an *Azospira*-like microorganism (95 % similarity) coincides […]"

**RC1. 18.** Ln. 210-256. Did the authors try to quantify nitrite? If there are nitrogen transformations occurring in this system, I would expect it to be important, and perhaps the ultimate oxidant for $Fe^{2+}$ in reactions 1 and 2.

**DP1.17.** The reviewer is right, nitrite could be expected to increase concentrations towards the anoxic part of the water column and exert an important control in reactions leading to Fe oxidation. Nitrite role, however, remains to be further tested experimentally or by specialized sampling/analytical protocols that can resolve reactive N availability and transformations occurring in the natural lab under examination. In our case, increasing Cl- concentrations with increasing depth hindered an accurate profiling of nitrite.

The revised text now informs the reader what anions were considered (please see DP1.9). Also, the following text was added as preamble to presenting reactions 1 to 3: Nitrite ($NO_2^-$), an intermediate between $NO_3^-$ and $NH_4^+$, can also accumulate. Yet concentration profiles of such intermediate remain to be accurately resolved in the increasingly saline (high chlorine) bottom water column of Lake Medard.

While Lns. 46-63 now read:

These cycles in the aqueous system under consideration are likely interlinked throughout microbial mediation in the generalized Reactions (1–3), but note that intermediate $NO_2^-$ may as well act as a relevant Fe(II) oxidant in this $O_2$-depleted system (Klueglein et al., 2014):

**RC1. 19.** Ln. 282. Are the authors referring to Fe(II) oxidation by Mn(III/IV)? Please clarify.

**DP1.19.** Yes. As per reaction 4, iron is oxidized and Mn is reduced. It can also be seen as Fe(II) as reductant of Mn(III,IV). To clarify on this note, we modified the text to read "[…] Divalent iron […]" (Ln. 496).

**RC1. 20.** Ln. 293. Please change "[Fe]" to "Fe concentration"; also here and throughout, please check tense agreement.

**DP1.20.** Changed as per reviewer suggestion. Sentence tense agreement also revised thoroughly.

**RC1. 21.** Ln. 301. I don't know about this. Attributing metabolism when you only have 91% similarity is tough.

**DP1.21.** True. The offending sentence is now removed.

**RC1. 22.** Ln. 329. Please remove "significantly"

**DP1.22.** Done.

**RC1. 23.** Ln. 336 and throughout this section. Why does diversity matter. Wouldn't relative abundance be more informative with respect to S transformations? There could be a whole lot of diversity of sulfate reducers, but they're only a minor fraction of the community.

**DP1.23.** We have clarified what we referred here as diversity as follow (Ln. 558): "[…] low number of taxonomic groups, and Ln. 579: "[…] a more diverse sulfur-respiring bacterial population (Fig. 5c). This was dominated by many relatively rare taxa and a few abundant lineages […]"

The issues of considering relative abundance alone as a control of relevant hydrochemical transformations include potential biases induced by sampling processing. The information provided can therefore only inform us on changes in conditions that allow, for instance rare taxa to better thrive, or additional MSRs to be detected with the protocols implemented. We have added the following text to add additional context to this matter (Ln. 59-61): "[…] Amplicon gene sequencing informed our ecological interpretations despite quantitative biases that are inherent in this type of data (see Piwosz et al., 2020) .[…]"

Piwosz, K., Shabarova, T., Pernthaler, J., Posch, T., Šimek, K., Porcal, P., and Salcher, M. M.: Bacterial and Eukaryotic Small-Subunit Amplicon Data Do Not Provide a Quantitative Picture of Microbial Communities, but They Are Reliable in the Context of Ecological Interpretations, 5, 2020.

**RC1. 24.** A later use of the term "diversity" leads me to believe the authors are referring to diversity of S metabolisms (e.g., oxidation, disproportionation, reduction of different redox states, etc.), but I'm not sure. Please revisit the use of this term and clarify.

**DP1.24.** The use of the term has been revisited and clarified as indicated above (DP1.23.).

**RC1. 25.** **Ln. 372-374:** If there's evidence of S metabolizing organisms and some aqueous chemical evidence of S transformations, why no change in del34S-sulfate?

**DP1.25.** This question of the reviewer is answered in Lns. 620-23of the revised text: "The intracellular isotope exchange of sulfite with anoxic ambient waters has been proven to produce an oxidized $SO_4^{2-}$ product that is enriched in $^{18}O$ relative to precursory thiosulfate and/or sulfite. This enrichment displays only a minor change, if any, in its corresponding S isotope composition (e.g., Böttcher et al., 2005; Johnston et al., 2014; Bertran et al., 2020; see Table 1)." (text was Lns. 398-401 of the preprint).

But please note that the following lines are now added as a closing statement of the section 4.4.1 (Lns. 624-27) to account for an increase observed at depth 48: "In line with this assertion, at the monimolimnion there is negligible sulfur isotope fractionation accompanying the recorded fractionation of oxygen isotope. Yet, our data recorded a small, but significant reverse sulfur isotope effect (+2.2 ‰) at the upper hypolimnion (Fig. 6a: 48 m depth). This isotope effect could be ascribed either to abiotic or biotic oxidation processes of intermediate S species occurring at that level of the water column (see Zerkle et al., 2016, their table 1). […]"

**RC1. 26.** Reactions 4-6. Is Mn-dependent S disproportionation from the Böttcher et al. 2001 paper? What about the siderite-dependent disproportionation? I am unsure how this reaction might occur.

**DP1.26.** Yes. The missing relevant references describing these disproportionation mechanisms were added (Ln. 613) to provide further context (for example to the reader interested in how the half reactions proceed).

$FeCO_3$: dissolution of siderite increases the availability of Fe(II) that scavenges excess $H_2S$ by-product, which sustains disproportioning bacterial growth (Thamdrup et al., 1993).

**RC1. 27.** Ln. 520, 534, 559. Why are these minerals italicized?

**DP1.27.** These are now subsections comprising section 4.6.

**Reviewer 2, Comment 1(RC2.1). [Article]:** is well-written and summarizes all principal aspect of the pit lake, as well as the importance of the study and how in general was conducted. There were few specific comments that I would the authors to take into consideration.

**DP2.1**. We sincerely thank the reviewer for her valuable time, the many suggestions generously given that—together with her constructive criticism—have significantly improved the MS for final publication.

**RC2.2. [Methods section]:** is nicely organized in subsections as supplementary information (S1), but there were specific details that would enhance the reproducibility of the applied methods.

**DP2.2.** Missing details that were missing are now added. For example, on sub-sampling tasks; HP-LC, etc. Also, please note that the methods section is now part of the main text.

**RC2.3. [Results and Discussion]:** the authors did a good job discussing all the results and its significance. This section requires more work specially with respect to enhancing the clarity of the figures and their description in the text.

**DP2.3.** The figures have been further processed for clarity, for example we added in all profiles the hypo and monimolimnion, etc, we improve colour palettes used for genomics figure, revised captions, etc.

**RC2.4a.** It was not clear to me when was the lake flooded, 2005 according to figure 1a? If so, please state this in line 43 where it is written: "This newly formed,…". Do you have supporting info of when did the lake become meromictic, or how long has it been meromictic?

**DP2.4a.** Information and a reference to relevant published work that fill this gap kindly highlighted by the reviewer, have been added (Lns. 86-8): "[…] The filling of the former open-cast mine pit with river water started in 2010 and was reportedly completed by 2016 (Kovar et al., 2016). […]"

Also, Lns. 101-02): "[…] Water column stratification has been observed since 2009, when environmental monitoring of the hydrological system begun (e.g., Medová et al., 2015) […]"

Medová, H., Přikryl, I., Zapomnělová, E., and Pechar, L.: Effect of Postmining Waters on Cyanobacterial Photosynthesis, Water Environment Research 87, 180–190, 2015.

**RC2. 3b**. Do the meromictic conditions of the lake vary seasonally? Please present supporting information about this too.

**DP2.4b.** Yes. To address the reviewer request we revised the MS, Lns. 350-53, to now read: "Short-lived changes in redox potential of about 150 mV in the bottom water column were recently considered by Umbría-Salinas et al. (2021). These changes have effects on water column speciation (Fig. B1, Appendix B), and affect the partitioning of several redox sensitive metals that bind to reactive iron phases in the upper sediments (Umbría-Salinas et al. 2021, for details)."

Also, the revised Fig. B1 now shows the Eh-pH variability that we evaluated in that published contribution.

**RC2. 5. Line 95**: the authors talk about the present oligotrophic stratified conditions of the lake as a topic sentence. First, I would like to see physico-chemical profiles at this point to ease the understanding of such conditions for the reader. I also would like the authors to describe in more detail such conditions in all stratified layers. Finally, the last sentence of the paragraph, starting in line 100, deserved more written description too.

**DP2.5.** A call to Fig 2a, describing details on stratification of the bottom water column, is now added earlier: Ln. 50. The reference to the oligotrophic state that the reviewer referred to was deleted for being extempory at that point, as kindly noticed by the reviewer. Additional description on the Pourbaix diagrams in Fig. B1 was added (Ln. 108-09).: "[…]. The stability diagrams show that the current physicochemical conditions of the bottom sulfatic waters favour colloidal Fe(III)-oxyhydroxides formation."

**RC2. 6a.** Methods In line 13 [S1], when the authors refer to ~11 mL water samples, is 11 mL the aliquot amount referred to in line 11?

**DP2.6a.** The description provided is clear on this sampling step (now Lns. 188-121): Aliquots of the lake water collected were immediately transferred from the sampler to pre-cleaned—i.e., three-times rinsed with $ddH_2O$ and oven-dried at 550 ºC, 12 mL glass exetainer septum capped vials (Labco), pre-filled with He(g) and 1mL NaCl oversaturated solution (40%) for $CH_4$, or 1 mL 85% phosphoric acid for $\Sigma CO_2$.

On board, the vials were filled with ~11 mL water samples […]". No changes were implemented regarding this query, but please note that Methods is now moved to the main text, Sect. 3.

**RC2. 6b.** In addition, how many samples were collected? Were they collected along depth? where exactly?

**DP2.6b.** Lns. 114-19 now clarify on this important matter: "[…] in the stratified portion of the water column of Lake Medard (from 47 to 55 m depth) in its central location (Fig. 1a, star). The probing resolution was 1 m above and below the $O_2$ minimum zone and 0.5 m at the redoxcline. Based on the profiles, water column samples (n = 8 and 4 replicates) were collected (in November 2019) using a Ruttner sampler with a capacity of 1.7 L. Flushing/rinsing of the sampling device with distilled water ($dH_2O$) was performed between samples. A total of eight samples were taken at depths 47, 48, 48.5, 49, 50, 52, 54 and 55 m. Replicate samples were taken at depths 47, 48.5, 50 and 54 m below the lake water surface. […]"

Also, Ln 137: "[…] A total of 11 water replicates (i.e., 47 to 54 m depth and replicates) were evaluated. […]"

**RC2.6c.** In section SM1.1.3, please clarify the number of samples taken. The same applies for SM1.1.4 and include information about samples from which depths (or layers) were considered for the cation and ion concentration analyses.

**DP2.6c.** Please refer to DP2.6b, above.

**RC2.6d.** In section1.1.5, please clarify the following questions: were the 11 water samples (line 29), the same as described in section SM1.1.3? If so? why eleven? does this number include biological replicates? are these only from two depths? it is important to clarify, where these samples were taken along depth. More details of the PCR and sequencing protocol would benefit future researchers and reproducibility of the methods.

**DP2.6d.** Please refer to DP2.6b.

**RC2.6d.** Were the mineralogical (SM1.2.2), isotopic (SM1.2.3), and SEM (SM1.2.4) analyses applied to all sliced sediments?

**DP2.6e.** Yes. The intro paragraph to section 3.2 (Methods applied to solid phases) reads:

**"3.2 Sediment samples**

We also sampled the upper anoxic sediment column to a depth of ~8 cm. The mineralogy of these fine-grained sediments (silt to clay in size) was qualitatively and semi-quantitatively assessed via X-ray diffraction (XRD). The $\delta^{34}S$ and $\delta^{18}O$ of gypsum ($CaSO_4 \cdot 2H_2O$), $\delta^{13}C$ of siderite ($FeCO_3$), and $\delta^{34}S$ isotope values of pyrite ($FeS_2$) from these sediments were also measured and reported as described above using the delta notation, $\delta = R_{sample}/R_{standard} - 1$, where R is the mole ratio. Scanning electron microscopy aided by electron dispersive spectrometry (SEM-EDS) was used for textural analyses focused on the S- and/or Fe-bearing phases. In addition, a sequential extraction scheme (after Poulton et al., 2004; Goldberg et al., 2012) was conducted to characterize the sedimentary partitioning of reactive Fe and Mn fractions. Details on these analyses follow. […]"

**RC2.7 Results and Discussion, Subsection 4.1:** I am little confused about what is shown in Figure 2a. What is happening above 48 m? To what depth are you referring to? depth of the lake water, or depth of the whole lake? Based on what is presented in Figure 2a, I interpret that the depth of the water column is only ~10 m? I am sorry if it is not that obvious to me, but it might be worth to clarify.

**DP2.7.** To address the lack of clarity kindly pointed out by the reviewer, the caption has been also modified as follow: "[…] Lake Medard in its central sampling location, which has a maximum depth of 56 m (a) [… ].". Also, note that all figures portraying a depth profile now read, in their vertical axis, "Water column depth (m)"

**RC2.8 Ln 121:** the authors refer to several previous profiles of the lake. Do you have previous profiles? Are they published somewhere? or included in the supp info?

**DP2.8.** Yes. Ln. 311, reference added: "[…] Petrash et al. (2018)".

**RC2.9 Ln. 135:** "The hydrochemically different monimolimnion persists in the deepest depressions of the lakebed throughout the year; although with slight variations in the monitored Eh range that could be

accompanied by minor (±1 m) shifts in the vertical position of the redoxcline." Can you show profiles of this on the supp info?

**DP2.9.** Short-lived Eh-Ph variations are now shown in Fig. B1 as per this request of the reviewer. Please note that a publication dedicated to evaluating this observation and their implications is also given: Umbria-Salinas et al. (2021).

**RC2.10** [caption of figure 2], authors refer to dysoxic (n=4) and anoxic (n=3): at which depth were these samples collected, respectively?

**DP2.10.** Caption to Fig 2b was modified as follow: "[…] quantified in the dysoxic (n= 4, 48-48.5 m depth) and anoxic (n= 3; >52 m depth) waters of Lake Medard […]"

**RC2.11:** Authors included a separating line referring to the redoxcline in Fig2b. Does this mean that the upper part of Fig 2b corresponds to the myxo-hypolimnion and the lower part to the monimolimnion? If so, please clarify it in the figure too. In addition, what are the red crosses? Could you also include an explanation in the caption?

**DP2.11.** Fig. 2b has been re-produced for clarity as per this important request of the reviewer.

**RC2.12 [Section 4.2]:** In line 150, the given DOC concentrations correspond to an average value of the 7 samples refer in figure 2b?

**DP2.12.** The average of all of the samples (and replicates) collected in the bottom waters. The text now reads (Ln 353): "The average of measured DOC concentrations in the bottom waters sampled is 1,050 ± 500 µM."

**RC2.13.** "A six- to ten-fold increase in concentrations of acetate, oxalate, and formate occurred towards the increasingly saline and O2-depleted bottom waters." This might be better to visualize in a profile. Could you please include one in the supp info?

**DP2.13.** No. A depth-profile is, unfortunately, not available for VFAs concentrations. This was the undesirable consequence of increasing $Cl^-$ concentrations towards the bottom, only sample with error below 20% were reported/considered.

**RC2.14.** In line 163, when authors referred to "[$\Sigma CO_2$] were inversely correlated with the $\delta^{13}C$ values", are they referring to figure 3d.

**DP2.14.** Referred to the profiles showed in Figs 3a-b (now clarified in Ln. 371).

**RC2.14.** Paragraph starting in line 169 should have included a reference to Table 1 somewhere.

**DP2.14.** The edited section, now starting in Ln. 365, includes few references to Table 1, where appropriate.

**RC2.15.** About Figure 3d referred in line 189, I thought this figure was referring to the water samples. Please, clarify or correct accordingly.

**DP2.14.** The flux we referred in Fig. 3d and text is from the sediments to the water column. To clarify, Ln. 393-4 now read: "[…] The range of estimated isotopic C values of the $CO_2$ flux from the sediments to the water column is between -3.0 and -4.2 ‰ (Fig. 3d). […]"

**RC2.15 Section 4.3:** Colours in Fig 5 must be changed. In 5a, there are two yellows, two greens, two light blues corresponding to different organisms, making it hard to interpret the figure and correlated with the written text. Fig 5b is even harder to differentiate colors and organisms.

**DP2.15.** Colour palettes used in the revised Fig. 5 were updated as per this relevant request of the reviewer.

**RC2.16.** While describing the microbial community, authors should be more quantitative (avoid low or high and refer to percentage). How much does "increases significantly" or "the abundance peak" mean? In addition, please be specific if what is shown in Fig 5. corresponds to normalized abundance in percentage with respect to the whole community or only among each microbial group shown separately in a b and c.

**DP2.16.** We did not implement quantitative PCR, in consequence the values are described for ecological interpretations only, e.g., *there is an increase in relative proportions {for a given OTU} from z1 to z2*. The value of amplicon reads do not constitute a definite measure of the *real* composition of the community, and limitations of 16S rDNA analyses has been discussed intensively in the microbial

ecology literature, and we now cite a recent work that nicely clarifies the general current view on this matter (Ln. 59-60 at the introduction section): "[…] *Amplicon gene sequencing informed our ecological interpretations despite* quantitative biases *that are inherent in this type of data* (*e.g.,* Piwosz et al., 2020). […]*"

*The biases are introduced, for example, during sample DNA extraction, PCR amplification, and could results from uneven coverage of universal primers across phylogenetic groups, the sequencing of amplified fragments of prokaryote rRNA genes can provide insights useful for ecological deductions (see Piwosz et al., 2020).<--- this text unrelated to our work, not added in submitted version.*

**RC2.17.** In section 4.3.2, the subtitle refers to dissolved Mn and Fe, are they total concentrations? otherwise please be specific and in accordance with what is shown in Fig 4b: MnII and FeII.

**DP2.17.** The subtitle now reads (Ln. 493): "4.3.2 Dissolved divalent manganese and iron, and the iron-utilizing prokaryotes"

Preamble Lns. 473-75 read: "Concentrations of these dissolved metals are operationally defined as the combined ionic and colloidal fractions that passed the 0.22 µm cut-off of membrane filters."

**RC2.18.** In addition, do you have concentrations of Mn(IV)? How do authors know Mn(IV) is settling down from the upper layer? Or Fe(II) is difussing upwards?

**DP2.18.** No. No particulate metal concentrations profiles were produced as part of this study.

The assertions mentioned are valid educated guesses based on the geochemical behaviour of Mn and Fe in redox interfaces and across concentration gradients, such as depicted in Figure 4 and numerically described in Table 1 (please referrer to classical refs. Davidson 1993; Loveley and Phillips, 1988).

**RC2.19.** In line 291, do authors have mineralogical evidence of this fact" "Dissolved phosphate is re-complexed back onto nanocrystalline and amorphous ferrihydrite-like phases precipitating at the redoxcline." The same comment for mackinawite mentioned in line 295.

**DP2.19.** Phosphate: We have presented (in Table 1) geochemical evidence for phosphate solubilization occurring in the monimolimnion (i.e., increasing dissolved phosphate concentrations towards the lakebed), with a decreased concentration trend towards the redoxcline that is indicative of complexation of the oxyanion via biotic and/or abiotic amorphous iron oxides formation at the oxycline (this is portrayed in Fig. 8).

Mackinawite: this is the prevalent metastable precursor of pyrite, and it is rather difficult to quantify in most practical cases, particularly when such cases involve field sampling. However, to comply with the query by the reviewer—regarding inferred monosulfide precipitation, we have added the following text (Ln.554-55): "[…] circumstantial evidence for FeS precipitation, with another being $\delta^{56}$Fe values that increased across the redoxcline and towards the SWI (Petrash et al., 2022)".

On this matter, further information was provided (Ln 848-53): "[…] We are furthering the interplay between Fe and S cycles in the O2-depleted water column by bridging our $\delta^{34}$S data with $\delta^{56}$Fe measurements. The combined results support an active vigorous co-recycling of these elements below the redoxcline (Petrash et al., 2022). Accordingly, an increase in the relative proportion of dissolved $^{56}$Fe towards the lakebed ($\delta^{56}$Fe = +0.12 ± 0.05 ‰) can be ascribed to precipitation of monosulfides, whilst precipitation of oxyhydroxides at the redoxcline leads to 56Fe depletion ($\delta^{56}$Fe = −1.77 ± 0.03) of the residual Fe(II) (cf. Busigny et al., 2014). "

Busigny, V., Planavsky, N. J., Jézéquel, D., Crowe, S., Louvat, P., Moureau, J., Viollier, E., and Lyons, T. W.: Iron isotopes in an Archean ocean analogue, Geochim. Cosmochim. Acta, 133, 443–462, 2014.

**RC2.20.** In line 303, when referring to *Pseudomonas* spp., do authors have any control showing that *Pseudomonas* was not part of the extraction kits, or sampling material?

**DP2.20.** Yes, controls had not *Pseudomonas*. Importantly, should the flaw suggested by the reviewer be the case, we would not expect to have contamination by this spp. only in one, intermediate sample—also its replicate, but not the neighbouring samples. *Pseudomonas* was also identified in similar redox interface by Petrash et al. (2018) in other sampling locations of Medard yet using different isolation kit. In consequence, we are confident in *Pseudomonas* spp. being relevant players in the metal (Mn) respiring community near the redoxcline.

**RC2.21.** In 310, include a reference for "anoxygenic phototrophic and nitrate reducing species (*Magnetospirillum* and *Ferrigenium*; Fig. 5b, Supplement 2)", and "*Azospira*-like species."

**DP2.21.** Done. Ln 533.

**RC2.22.** In line 323, when referring to the peak of *Geobacter,* include where specifically and how much?

**DP2.22.** The text indicated by the reviewer has been modified as follow (now Ln. 543): "[…] within the monimolimnion, at about 54 m depth." Again (as per DP2.16, above), we have no qPCR data, and we used universal primers that are not specific for Geobacteracea. The universal primers, however, did amplify a relatively higher abundance of *Geobacter* that, incidentally, coincides with the peak of iron reduction. Importantly, please note that the Krona chart in Supplement 2, interactively produces the # reads for any of the bacteria or archaea present in each sample/ or replicate that might be interest to the reader and permit evaluating its variation with depth.

**RC2.23.** There are some names of organisms in Fig 5b that are not mentioned in the text. Should you better remove them from the figure and include them, as other less abundant taxa, or mentioned them in the text.

**DP2.23.** Non-mentioned organisms in Fig 5b are now removed as per this relevant suggestion of the reviewer.

**RC2.24.** In line 345, in Fig 5c is only as *Thioalkalivibrio...*should you add the species name too as you did in the text?

**DP2.24.** "*[…] paradoxus"* added (Ln. 568).

**RC2.25.** In line 349, authors mentioned "The abundance of *S. hydrogenivorans* increases in parallel to a decrease in the *T. paradoxus*-like bacterium, which suggests that the latter may be at a disadvantage and limited by organic C fixation under the specific hydrochemical conditions prevailing at the redoxcline" With the current colors in Figure 5, it is difficult to see what you are showing in the text.

**DP2.25.** True. The palette was updated. Also, please note that numerical values are in the Krona chart provided as supplement.

**RC2.26 Ln. 360:** which bar corresponds to *D. acetoxidans* in Figure 5c. The same comment for *Desulfaticacillum* in line 365 and *Sulfitobacter* in line 366.

**DP2.26.** Please see DP2.25.

**RC2.27.** In general, with the current colors in Fig 5, it is difficult to agree with the conclusions stated by the authors in section 4.3

**DP2.27.** Please see DP2.25.

**RC2.28 Section 4.4 Ln 374:** do authors have values to support the "weak correlation"?

**DP2.28.** The supporting values are listed in Table 1. The $R^2$ (~0.16) from an attempt for linear correlation of such values (shown below only), is now provided in the text (Ln. 596).

[Figure]

**RC2.29:** A reference is needed for the following statement: "It is also within the range observed in studies of S disproportionation reactions generally proceeding under anoxic conditions" in line 383.

**DP2.29.** True. Relevant references now added (Ln. 604): "[…] observed in studies of S disproportionation reactions generally proceeding under anoxic conditions (e.g., Böttcher et al., 2001, 2005)"

**RC2.30:** Reference needed for the examples given in line 409.

**DP2.30.** There are no "examples" given in line 409 of the preprint, but stable isotope measurements conducted in the acidic drainage shown in Fig. 1. These are plotted as well in Fig. 6, which now with colours better allow distinguishing them.

**RC2.31. Section 4.5:** Be more quantitative with respect to sentences like in line 445: ".... increase slightly towards the bottom of our 8 cm depth core but their abundance, relative to total iron, decreases downwards" or 451: "a significant increase…"

**DP2.31.** Quantities are listed/detailed in Table 2, but we have deleted qualificatives such a 'significant' or 'slightly', while referencing changes listed by depth (cm) in Table 2.

**RC2.32. Ln. 454:** a Sect. 4.6.3 is referred but not found in the current version of the manuscript.

**DP2.32.** Section 4.6.1 is now referred (there was a typo), Ln. 422.

**RC2.33.** Section. 4.7, Ln. 595, name which "scarce examples" authors are referring, as well as in line 596: add reference and name which lakes

**DP2.33.** This request of the reviewer has been addressed as follow (now Lns. 872-73): "[…] the scarce examples of redox stratified euxinic marine basins existing today (i.e., Black Sea, Cariaco Basin; Meyer and Kump, 2008) nor in the few natural mesotrophic to eutrophic ferruginous lakes presumedly analogues to ancient redox stratified oceans (see Koeksoy et al., 2015)"

**RC2.34.** About figure 8: Nice figure but there are some improvements to be done: 1) a legend is required to understand symbols and colors in the figure. 2) add a depth profile and names of each layer. 3) why is it necessary the venn diagrams for the microbes, what each color of the circles mean? Add the biogeochemical role of each microbial group included in the figure.

**DP2.34a.** The caption of the figure has been simplified, and to address the chosen grouping, representing functionalities, we added further information. The caption now reads:

[Figure]

"Figure 8. Scheme summarizing the speciation and stable isotopes ranges of sulfur-bearing phases (pyrite, $S^0$, CRS; gypsum, GY) and siderite (SID) and the biogeochemical cycling mechanisms likely operating in the redox stratified Lake Medard and its SWI. (Background colours as in Fig. 2) The prokaryote groups depicted represent nitrate, iron and sulfur utilizing species identified via 16S gene amplicon sequencing (see text for details)."

Also note that, as suggested by the reviewer, additional information was added to the edited version of the figure.

**DP2.36.** The reviewer kindly pointed out also a list of typos, repeated text, etc. These minor issues indicated by the reviewer in the preprint, were all addressed. Also, sentence tense agreement also revised thoroughly through the text.

**Reviewer 3 (Dr. Alexandra A. Phillips)**

**RC3.1** Cutting jargon: this paper is strongest when integrating results across their interdisciplinary dataset. However, the paper is in many places (namely the abstract, introduction, and final discussion section) unnecessarily complex and loses the non-expert reader. The authors should revisit these parts of the manuscript with an eye for unnecessary jargon - places they will lose interested geobiologists in a complicated explanation of geology, for example. In places where jargon is unavoidable, offering a few more definitions to the reader will help broaden the readership of this really interesting interdisciplinary study.

**DP3.1.** As per this relevant reviewer's suggestion, we have revisited and edited the text for the sake of simplicity, explained limnological-specific jargon, and simplified the text whenever possible. For example (Ln. 40): "[…] Natural lakes that display permanent stagnation and marked redox gradients in their water column are termed meromictic.[…]"

Also, Ln. 49: "[…] This newly formed, lacustrine system features low nutrient contents (i.e., it is oligotrophic), […]"

We also simplified our study site description, which now has no jargon related to the Miocene rift stage, fault mineralization, and paleolake development.

**RC3.2** Improved figures: The figures should be reworked with a goal of consistency. Much of the results/discussion center around different zones of the lake's water column, but it is difficult to orient yourself across the many figures. Figure 2a panel 4 does a nice job showing the mixolimnion, hypolimnion, and monimolimnion - I think it would be helpful to see these zones in all the figures - either as the different shades of gray like in the Eh figure or with dashed lines and labels. A few of the figures also appear low resolution - namely, figure 5, 6, and 7. Other comments specific to each figure can be seen below.

**DP3.2.** We have carefully implemented this relevant suggestion to all the figures where a water column profile is shown. These now consistently display the redox stratification. All figures are 300 dpi, portable network graphics.

**RC3.3.** Separating results and discussion: It was difficult as a reader to follow the results and discussion section - I wanted to already be acquainted with much of the data before seeing it synthesized. I found myself jumping back and forth across the sections often. My suggestion to the authors is to split the results and discussion and focus the results to be a succinct section, with extraneous details moved to the SI.

**DP3.3.** We tried implementing this suggestion prior to submission, and again now—to comply with the reviewer suggestion, but our feeling is that it detrimentally affects the intended integration of microbial ecology and geochemistry, also the importance of certain geological (e.g., mineralogical) observations for geochemical/ ecological interpretations. Then, the MS ends being even more disseminated and sections largerly unbridged. Therefore, we have kept the manuscript as a Results and Discussion-type report.

**RC3.4.** Some parts felt too long and should be more succinct, while other parts begged for more detailed discussion

**DP3.4.** We have shortened some sections (e.g., Sects. 2 and 4.2.2), while adding additional discussion of results, where presumed desirable (e.g., Lns. 443-54; 456, 508-10, 522-26, 540, among others

**RC3.5. More methods:** Currently, [in the main text] there are not enough details for someone to replicate any of the work or think critically about advantages or disadvantages for any method. Much of the SI methods section should be moved to the main text.

**DP3.5.** Methods are now moved to the main text as per reviewer's request. Also, missing information on HP-LC was added (see DP3.11f, below), and all additional editions suggested by the reviewer in RC3.11 were incorporated.

**RC3.6. [Abstract]**: I would recommend moving the last few lines on the importance of the study more broadly to earlier in the abstract, perhaps as the second sentence, and then expanding/clarifying the point in the current first sentence that this geochemical situation is an unusual but exciting case study

**DP3.6.** We implemented the abstract editions kindly offered by the reviewer.

**RC3.7 [Introduction] Line 32:** It would be helpful for those less familiar with limnology terms to define meromictic briefly, perhaps simply as "indefinitely stratified" or something similar

**DP3.7.** Done, please see DP3.1.

**RC3.8 Line 33:** "common sulfate deficiency" feels unnecessarily complicated, do you mean low in sulfate or absent of sulfate?

**DP3.8.** Yes. The revised text (Ln. 35) now reads: "but low sulfate concentrations"

**RC3.9 Line 39:** If possible, I think it would be helpful to add one more sentence about the importance/relevance to paleo-studies - the connection to me right now is a little weak so would be great to strengthen that point a little more - how exactly do they better inform Precambrian Ocean redox stratification models?

**DP3.9.** Done. Lns. 38-49 now reads: "[...]. empirical framework to interpret modern iron biomineralization mechanisms and, by analogy, similar processes that would have allocated widespread, yet punctual deposition of ancient iron formations in the Precambrian rock record.

Lakes that display permanent stagnation and marked redox gradients in their water column are termed meromictic. Meromictic lakes featuring ferruginous conditions in their water columns (i.e., $[Fe^{2+}]$ > $[H_2S/HS^-]$ and $[Fe^{2+}]$ > $[NO_3^-/NO_2^-]$) are relevant to decipher the environmental significance of specific chemical and isotopic signals recorded in iron-rich deposits, and to advance paleoenvironmental interpretations of redox stratified oceans, such as those prevalent during the Precambrian (Canfield et al., 2018), or intermittently developed during the Phanerozoic. [...]"

**RC3.10 Line 47:** Would be helpful to include a mention of the lake's pH as well (anywhere in this introduction)

**DP3.10.** Done, Lns. 46-47 now read, "[...] Ferruginous water columns that also contain elevated dissolved sulfate concentrations are not uncommon in acidic shallow pit lakes (e.g., Denimal et al., 2005; Trettin et al., 2007), and have also been reported from the pH neutralized post-mining Lake Medard in NW Czechia (Petrash et al., 2018; Fig. 1a)."

**RC3.11a [Methods]:** I think a majority of the SI details should be moved to the main text - because of this I also have line edit suggestions for the SI methods.

**DP3.11a.** Please see 'DP3.5' above. We thank the reviewer for this relevant suggestion.

**RC3.11b.** Water sampling: what depths were sampled? Did those change based on the physiological parameters prior to water sampling?

**DP3.11b.** Methods now clearly state the water depths sampled (Ln. 115 -19): "[...] The probing resolution was 1 m above and below the $O_2$ minimum zone and 0.5 m at the redoxcline. Based on the profiles, water column samples (n = 8 and 4 replicates) were collected (in November 2019) using a Ruttner sampler with a capacity of 1.7 L. Flushing/rinsing of the sampling device with distilled water (dH_2O) was performed between samples. A total of eight samples were taken at depths 47, 48, 48.5, 49, 50, 52, 54 and 55 m. Replicate samples were taken at depths 47, 48.5, 50 and 54 m below the lake water surface."

**RC3.11c. Line 11:** were the exetainers cleaned prior to sampling?

**DP3.11c.** Yes, this is stated in Ln. 301-02 of the methods sections that now read: "[...] from the sampler to pre-cleaned—i.e., three-times rinsed with ddH_2O and oven-dried at 550 ºC, 12 mL glass exetainer septum capped vials (Labco), [...]".

**RC3.11d.** Line 13: define PES

**DP3.11d.** PES acronym definition is now in Ln. 165 (its first appearance): "[...] using sterile high flow, 28 mm diameter, polyethersulfone (PES) filters to remove particles >0.22 μm [...].

**RC3.11e.** Line 22: please define which anions and cations

**DP3.11e.** Defined as per this important suggestion kindly provided by the reviewer (Lns. 120-21): "[...] (ii) mass determinations of cations (iron, manganese, potassium, sodium, magnesium and calcium); (iii) high pressure liquid chromatography for concentrations of chlorine, sulfate, nitrate, ammonium and phosphate anions, and VFA abundances in the bottom water column; [...]".

**RC3.11f.** Line 70: more details are needed on the IC method used for the VFA analysis - what is the run time, column used, etc?

**DP3.11f.** More details added to Methods, Ln 181-91 now reads:

**"3.1.5 Ions, ammonia, and VFAs concentration analyses**

[…]

Ions, ammonia and VFAs concentrations were measured in filtered, unacidified water sample aliquots via high pressure liquid chromatography (HP-LC) at BC-CAS, České Budějovice. For these analyses we used an ICS5000 + Eluent Generator (Dionex), with conductivity detection application, and suppression. Analytes were separated using Dionex IonPac AS11-HC-4 µm (anions, VFAs) and IonPac CS16-4 µm (ammonium) columns (2x250 mm in size). The flow rate was 0.36 mL/min; run time was 65 min (anions, VFAs) and 17 min for ammonium. Potassium hydroxide was the eluent for inorganic anions and monovalent organic acids; methanesulfonic acid was the eluent for ammonium ion detection/quantification. A combined stock calibration standard solution featuring environmentally relevant anions ratios was used for determining concentrations and was prepared from corresponding analytical-reagent grade salts. . To optimize and calibrate the method for VFA analyses and determine the limits of detection, we used stock mixtures of IC grade formate, oxalate, acetate, lactate, pyruvate, and butyrate standards for preparing our working saline stocks solutions. Detection limits were better than 60 ppb for lactate and oxalate, and 200 ppb for pyruvate, formate, and acetate. Recoveries, based on standards, exceed 80 % for all analytes reported."

**RC3.11g.** Line 77: How much 5M NaOH was added?

**DP3.11g.** Information missing was added (Ln. 218): "[…] then 50 mL of 5 M NaOH were added. […]"

**RC3.11h. Line 87:** should be "quantified gravimetrically" - also can you clarify what you mean here? Just weighed?

**DP3.11h.** Edited to "weighted" (Ln 228).

RC3.11i. Line 147: Sort of unusual to report 3 sigma, maybe just report 2 sigma as you did earlier to be consistent

**DP3.11i.** Typo deleted, it corresponded to uncorrected text for preliminary results considered in an earlier draft.

**RC3.12.** Supplement figure 3: this appears very pixelated on my download; can you make sure the figure has high enough resolution?

**DP3.12.** A higher resolution printout of the PREEQC results for mineral's SI is now provided in Supplement 1.

**RC3.13.** Line 119: awkward to start the sentence with "Figure 2a"

**DP3.13.** Sentence corrected as per this request of the reviewer. The offending line now reads (Ln. 308): "Physicochemical parameters measured in the dysoxic to anoxic waters at the time of sampling are shown in Fig. 2a […]"

**RC3.14.** Line 150: can you elaborate on the DOC concentrations? What does that tell you about the system - is it typical or unusual? A little more discussion would be great, especially because you mention in the abstract that SR may be limited by low amounts of metabolizable OC

**DP3.14.** Done, the section now states the following in its first paragraph:

"The average of measured DOC concentrations in the bottom waters sampled is 1,050 ± 500 µM. This range of values was higher than observed in the bottom waters of meromictic lakes such as or Matano (< 100 µM; Crowe et al., 2008), or Pavin (300 ± 100 µM; Viollier et al., 1995). DOC is generally comprised of relatively high molecular weight organic compounds (not quantified here), such as cellular exudates from alive and senescent planktonic microorganisms (e.g., algae, protists, bacteria) and their degradation products. Probably present in solution were also soluble humic substances derived from the biological breakdown of refractory organic matter (i.e., lignite particles) in the sediment (Petrash et al., 2018). VFAs are linear short-chain aliphatic mono-carboxylate compounds produced during anaerobic degradation of the organic compounds referred above. They serve as C sources and electron donors for the planktonic microbial heterotrophy and were therefore quantified here. VFAs in the bottom waters […]"

**RC3.15. Line 155:** what is your hypothesis for the change in VFA concentration in the different layers of the lake? Can you relate this more explicitly to your 16S data at all?

**DP3.15.** We have related changes in VFA with the microbial ecology for example in 376:

" […]Therefore, the complete (to $CO_2$) and incomplete (to acetate) oxidation of lactate by MSR could […]"

Also, Lns. 564-66, i.e., with regard to *Desulfobulbus* more likely disproportionating $S^0$, instead of thriving organotrophically: "Pyruvate, as lactate, was found below our detection limits across the bottom water column where sequences distantly related to *D. propionicus* (91 % similarity in 428bp) appeared to be particularly abundant (Fig. 5c; Supplement 1).

**RC3.16. Line 159:** I think a mention of pH should come much earlier, at least very early in results, if not hinted at in introduction - my initial assumption from hearing about a post-mining lake would be to expect really acidic conditions, so saying that the pH is closer to 7-8 earlier would be helpful.

**DP3.16.** As per reviewer suggestion, pH is mentioned in then introduction (DP3.10). Also, early in the results (Sect. 4.1, 1st paragraph, now Lns. 312-13 reads): "The pH in the hypolimnion was ~8.2 and decreased moderately downwards, reaching 7.4 ± 0.2 units near the anoxic SWI."

**RC3.17** Line 163: Please put some of the d13C values in the text, such as an average or range

**DP3..17** Done. Lns 702-03: […]. The $\delta^{13}C$ values are in the range +0.2 to -4.1, and were directly correlated with the dissolved sulfate concentrations $[SO_4^{2-}]$ (Table 1), […]

**RC3.18.** This discussion on 4.2.2 on total DIC seems very lengthy compared to the other results sections and could be shortened for readability and to better emphasize the main points.

**DP3.18.** The discussion on total DIC has been now streamlined, shortened some 20 % (it begins in Ln 365).

**RC3.19a.** Line 171: Please change "d13C signatures" here and elsewhere to d13C values.

**DP3.19a.** Done. We now referred to values regarding this isotope system.

**RC3.19b Line 189:** Instead of "estimated isotopic C signature of the CO2" say either estimated C isotope composition or estimated d13C value

**DP3.19b.** Edited as per reviewer suggestions.

**RC3.20 Line 370:** the title of 4.4.1 is awkward, maybe "Isotopic evidence"

**DP3.20.** The sub-section titles were edited as per reviewer's suggestion. The specific title now reads: "4.4.1 A proxy for disproportionation" (Ln. 591)

**RC3.21.** Line 373: I would suggest avoiding "heavier" and just stick with "enriched in 18O"

DP3.21. Done, Ln. 594, also Ln 621.

**RC3.22 Line 375:** a number itself can't be narrow, so maybe change to something like "the bottom waters had a narrow range of d18O values: X to Y"

**DP3.22.** Text edited as per reviewer suggestion, Ln. 597: "The ambient bottom waters had a narrow $\delta^{18}O_{H2O}$ range of values: −6.1 to −6.7 ‰. […]"

**RC3.23 Line 410:** you say that for the initial sulfate composition it is reasonable to assume its similar to the nearby acidic drainage and the pit lake before flooding - the second seems more reasonable to me but you dont report those values in the text? What are those?

**DP3.23.** Values were/are reported in the compilation made in Fig B2 (Appendix B).

**RC3.24. Line 574:** extra parenthesis dangling  - sentence is also not grammatically correct, so should be fixed

**DP3.24.** Corrected Lns. 808-09: "[…] could be ascribed to incomplete microbial sulfate reduction, with an additional open system oxidative sulfur cycling […]"

**RC3.25 Line 595:** more references needed here, overall really enjoyed this section on the paleo implications!

**DP3.25.** The reference now provided, Meyer and Kump (2008), is a very nice review (Ln. 872).

**RC3.26 Figure 1:** I really enjoyed this figure! Especially nice to see the lake from 2005 to 2020. The figure caption has a call out to panels b and c, but not to panel a, so that should be added.

**DP3.26.** The figure's caption is now corrected and includes missing information.

**RC3.27 Figure 2:** In part b I think the main idea is to compare the VFA concentrations above and below the redoxcline - its currently hard to see that difference and the relative differences between other VFAs - it would be more clear to show these on all the same plot - so instead of 6 separate figures just one figure

**DP3.27.** The figure's caption is now corrected as suggested. The revised Fig. 2b now shows, in the background, the stratification defined by the Eh gradients.

**RC3.28 Table 1:** I'm a little concerned by the errors in the ammonium measurements - especially that surface sample, where the error is larger than the measured value - is there also a reason why phosphate doesn't have concentration brackets - maybe just an error?

**DP3.28.** We thank the reviewer for her attention to details that allowed pointing out our mistake in data transcription to Table 1. Based on our replicates, errors in HC-LC data for ammonia, and in other anions simultaneously measured, are now correctly stated and do coincide to what was shown in Fig. 4a (error bars).

**RC3.29 Figure 3:** For panel A can you change it to mM so that the range is not to 12000 in the axis? Also needs more tic marks to see values in between, panel c also needs further tic marks for sulfate concentration. In the figure caption there is a CO2 that needs the 2 to be subscripted and "value" needs to be added after d13C.

**DP3.29.** Done.

**RC3.30 Figure 5:** this figure is pretty hard to read with the colors as they are - I would think about the main point you are trying to make with this figure - instead of having the arrow for the redoxycline. I would make the edit I suggested earlier for all figures, having different shades of grey boxes in the background - its hard to compare the abundance of different microbes against each other because the scale changes across panels as well.

**DP3.30.** Figure 5 was modified as per reviewer's request. We conserved, however, the variable scales as they do directly convey information on the change in relative abundances of the functional groups that we considered.

**RC3.31 Figure 6:** there is one data point for d34S of sulfate that is much higher value - at ~13.5 permil? Is this an outlier?

**DP3.31.** Concerning such value—effectively an outlier (see boxplot below), the revised text (Ln. 623-27) now reads: "In line with this assertion, at the monimolimnion there is negligible sulfur isotope fractionation accompanying the recorded fractionation of oxygen isotope. Yet, our data recorded a small, but significant reverse sulfur isotope effect (+2.2 ‰) at the upper hypolimnion (Fig. 6a: 48 m depth). This isotope effect could be ascribed either to abiotic or biotic oxidation processes of intermediate S species occurring at that level of the water column (see Zerkle et al., 2016, their table 1)."

[Figure]

**RC3.32.** Should be mentioned in text - 6c would be a bit easier to see if the symbols were also colored if possible

**DP3.32.** Figure 6c now has coloured markers as per reviewer's suggestion.

**RC3.33.** I hope these comments were helpful and they assist in improving what is already a really interesting manuscript.

Thanks! They did.